# Genetic immune escape landscape in primary and metastatic cancer

Francisco Martínez-Jiménez ● [1,2,5,6] ✉, Peter Priestley[3,6], Charles Shale[3], Jonathan Baber[3], Erik Rozemuller[4] & Edwin Cuppen ● [1,2] ✉

Studies have characterized the immune escape landscape across primary tumors. However, whether late-stage metastatic tumors present differences in genetic immune escape (GIE) prevalence and dynamics remains unclear. We performed a pan-cancer characterization of GIE prevalence across six immune escape pathways in 6,319 uniformly processed tumor samples. To address the complexity of the HLA-I locus in the germline and in tumors, we developed LILAC, an open-source integrative framework. One in four tumors harbors GIE alterations, with high mechanistic and frequency variability across cancer types. GIE prevalence is generally consistent between primary and metastatic tumors. We reveal that GIE alterations are selected for in tumor evolution and focal loss of heterozygosity of *HLA-I* tends to eliminate the HLA allele, presenting the largest neoepitope repertoire. Finally, high mutational burden tumors showed a tendency toward focal loss of heterozygosity of *HLA-I* as the immune evasion mechanism, whereas, in hypermutated tumors, other immune evasion strategies prevail.

Cancer immune escape is the process whereby tumor cells prevent their elimination by the immune system[1,2]. Tumors acquire this capacity as a response to the accumulation of tumor-specific alterations, which may be presented—in the form of neoepitopes—by the major histocompatibility complex class I (MHC-I). Escape from immune system recognition often involves tumor-specific genomic alterations in immune-related pathways, a process named genetic immune escape (GIE).

GIE alterations operate through different mechanisms, including partial or complete abrogation of neoepitope presentation[3] or suppression of proapoptotic signals from the surrounding immune cells[4]. Therefore, identification of GIE events across human cancers is key to understanding the interplay between cancer cells and the immune system, as well as to enable effective precision medicine based on immunotherapy.

Previous studies have performed cancer type-specific molecular profiling of GIE events and their phenotypic implications in several cancer types, including non-small-cell lung cancer[5,6] (NSCLC) and colorectal carcinoma[7], among others[8,9]. Others have performed an extensive analysis of loss of heterozygosity (LOH) of *HLA-I* across thousands of tumor samples[10]. However, a pan-cancer analysis of the prevalence and impact of diverse GIE events is currently lacking. In addition, the focus of these studies was to portray GIE in early stage primary tumors, whereas the changes induced by exposure to treatment and by the metastatic bottleneck have not been comprehensively addressed.

One of the main challenges to perform such analyses lies in the extraordinary diversity of the *HLA-I* locus, with >15,000 different sequences of the *HLA-A*, *HLA-B* and *HLA-C* genes reported to date[11]. This extensive polymorphism hampers the identification of tumor-specific somatic alterations, prompting the development of tools that specifically identify LOH of *HLA-I* (ref. 12) or *HLA-I* somatic mutations[13] from whole-exome sequencing (WES) and whole-genome sequencing (WGS) data. However, none of these tools provides an integrative characterization of the *HLA-I* tumor status in both the germline and the tumor, which includes *HLA-I* typing, allelic imbalance, LOH of *HLA-I* and somatic mutation annotation.

[1]Center for Molecular Medicine and Oncode Institute, University Medical Center Utrecht, Utrecht, the Netherlands. [2]Hartwig Medical Foundation, Amsterdam, the Netherlands. [3]Hartwig Medical Foundation Australia, Sydney, New South Wales, Australia. [4]GenDx, Utrecht, the Netherlands. [5]Present address: Vall d'Hebron Institute of Oncology, Barcelona, Spain. [6]These authors contributed equally: Francisco Martínez-Jiménez, Peter Priestley. ✉e-mail: fmartinez@vhio.net; e.cuppen@hartwigmedicalfoundation.nl

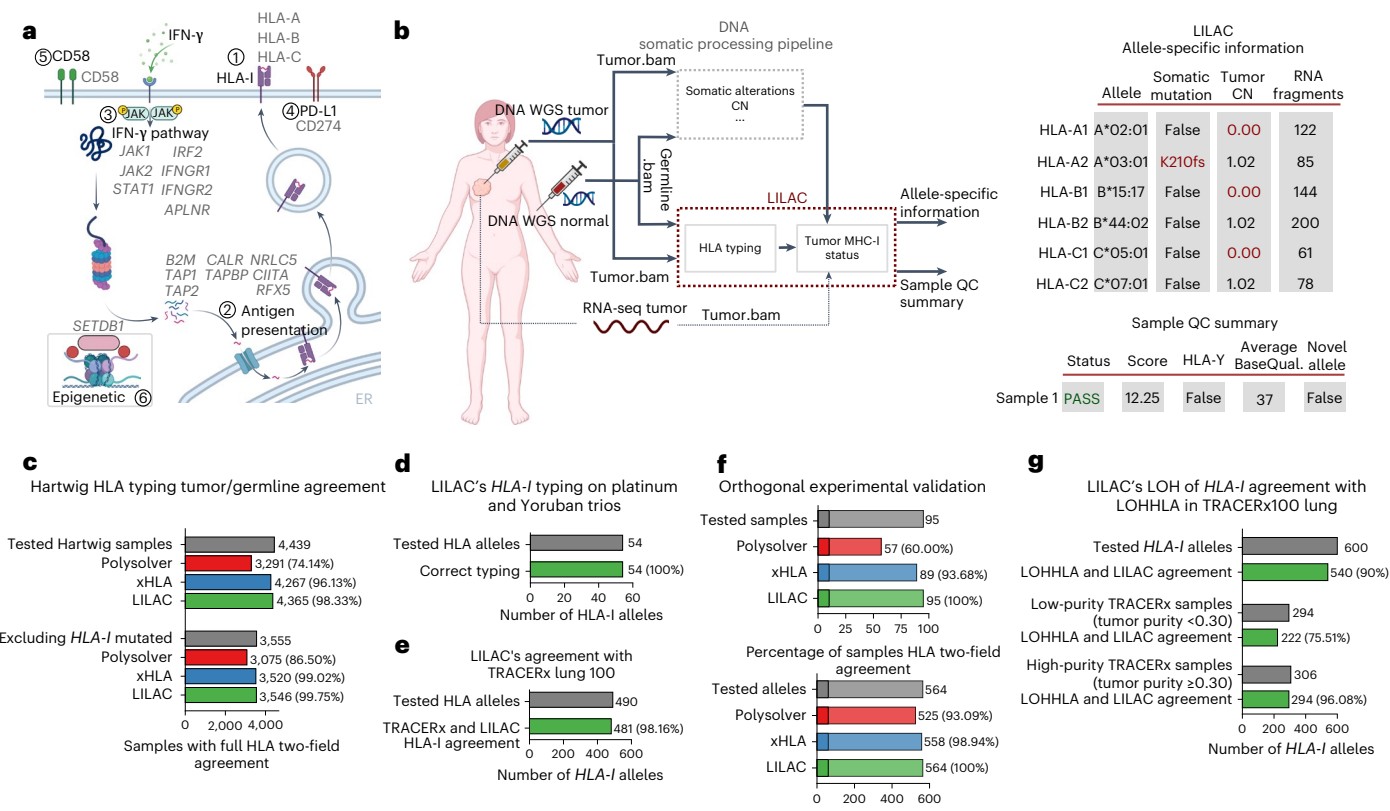

**Fig. 1 | Inference of *HLA-I* tumor status with LILAC. a**, Representation of the six immune escape pathways considered in the present study alongside their associated genes (adapted from 'MHC class I and II pathways', by BioRender. com). The genes considered for each immune escape pathway are depicted in gray. **b**, Left, workflow of the Hartwig tumor analytical pipeline integrating LILAC. LILAC's framework is highlighted with red text and a red border. Right, tables showing an illustrative example of LILAC's allele-specific and global patient

reports (partially created with BioRender.com). QC, quality control; BaseQual., basecalling quality score. **c**, *HLA-I* typing tumor and germline agreement in Hartwig cohort. **d**, LILAC's *HLA-I* typing validation using Platinum and Yoruban family trios. **e**, LILAC's agreement with the *HLA-I* types from the TRACERx lung cohort[12]. **f**, LILAC's *HLA-I* typing experimental validation. **g**, LILAC and LOH of *HLA-I* agreement in the TRACERx100 lung cancer cohort.

In the present study, we present a pan-cancer landscape of the GIE prevalence in primary (represented by the PCAWG (pan-cancer analysis of whole genomes) cohort) and unmatched metastatic patients (represented by the Hartwig cohort). Furthermore, to address the complexity of the *HLA-I* locus, we developed LILAC, an open-source integrative framework that characterizes the *HLA-I* locus, including its tumor status from WGS data. We applied LILAC and a universal tumor-processing pipeline to establish a comprehensive portrait of GIE events and their positive selection landscape across six different pathways associated with an immune evasion phenotype: the *HLA-I* locus, the antigen presentation machinery, interferon (IFN)-γ signaling pathway, the programmed cell death ligand 1 (PD-L1) immune checkpoint, the costimulatory signaling by the CD58 receptor and epigenetic immune escape driven by *SETDB1* (Fig. 1a and Supplementary Table 1). We also studied how the tumor mutational burden (TMB) and other genomic and environmental features influence the prevalence of GIE alterations, providing insights into tumorigenesis and its interplay with the immune system.

## Results

### Inference of *HLA-I* tumor status with LILAC
Inference of the correct *HLA-I* tumor status is fundamental to identifying GIE alterations (Fig. 1a), to estimate the neoepitope repertoire and burden and to predict the response to immune checkpoint inhibitors[14,15] (ICIs). We have developed LILAC, a framework that performs *HLA-I* typing for the germline of each patient, as well as determining the status of each of those alleles in the tumor using WGS data on tumor-normal

paired samples as input. LILAC also allows for detection of novel human leukocyte antigen (HLA) alleles and provides allele-specific and sample-level, quality control measurements (Fig. 1b and Supplementary Note 1).

We first assessed LILAC's *HLA-I* typing robustness by independently calculating the germline and tumor *HLA-I* two-field-calling agreement across 6,279 patients, including 4,439 patients from the Hartwig[16] dataset and 1,839 from the PCAWG[17] cohort. LILAC showed the highest agreement compared with two state-of-the-art *HLA* typing tools, Polysolver[13] and xHLA[18] (Fig. 1c, Extended Data Fig. 1a and Supplementary Data 1). The Hartwig dataset showed higher normal-tumor agreement for all tools, possibly due to the higher sequencing coverage and read quality of this dataset. In a three-way comparison, LILAC also displayed the highest overlap with the predictions from the other tools across both datasets (Extended Data Fig. 1b,c). Moreover, LILAC's *HLA-I* typing performance on three family trios with diverse genetic ancestries showed a perfect agreement with previously reported *HLA-I* types (Fig. 1d). Next, we demonstrated WES applicability by running LILAC on the TRACERx100 lung cohort, where it showed a 98.16% agreement with the *HLA-I* types originally reported in the publication[12] (Fig. 1e). Finally, we evaluated LILAC *HLA-I* typing sensitivity in a set of 95 samples with challenging *HLA-I* types—including 10 from tumor biopsies—with an independent orthogonal and clinically validated *HLA-I* typing approach (Supplementary Note 1). LILAC showed a perfect 100% two-field agreement across the 564 alleles, higher than Polysolver (93.09%) and xHLA (98.94%) agreements (Fig. 1f and Supplementary Data 1). To conclude, LILAC reported nine somatic mutations in seven

of the tumor biopsies evaluated. All of them were perfectly matched by the orthogonal approach (Supplementary Data 1).

HLA allele-specific, tumor copy number (CN) determination is key to identify LOH of *HLA-I* genes in tumors, a well-established mechanism of immune evasion[10,12]. LILAC annotates allele-specific ploidy levels of each *HLA-I* allele based on the purity-corrected local tumor CN estimations and the number of fragments assigned to each allele (Supplementary Note 1). WGS data provide adequate resolution to annotate purity-adjusted minor and major allele ploidy in the HLA-I locus (Extended Data Fig. 1d,e). Moreover, we quantified LILAC's agreement with LOHHLA[12] in the TRACERx100 lung WES cohort. LILAC and LOHHLA estimates displayed a global 90% agreement (Fig. 1g and Supplementary Data 1). Importantly, high tumor purity samples showed considerably better concordance than low-purity samples (96.08% in samples with tumor purity ≥0.3, 75.51% when tumor purity <0.3), reflecting increased challenges for genome-wide CN loss calling in low-purity WES samples. Finally, the three tumor samples harboring LOH of *HLA-I*, according to our framework and evaluated by the orthogonal approach, displayed a strong allelic imbalance in the experimental validation (Supplementary Data 1).

### GIE prevalence across cancer types

We then combined LILAC with the Hartwig tumor analytical cancer WGS pipeline[16,19] to annotate GIE events across 6 pathways strongly associated with immune escape (Fig. 1a and Supplementary Table 1) across 6,319 uniformly processed WGS samples[20], including 1,880 primary patients from PCAWG and 4,439 patients with metastases from Hartwig (Fig. 2a, Extended Data Fig. 2a, Supplementary Table 2 and Supplementary Note 1). In total, these patients were classified into 58 cancer types, which included 30 tumor types with sufficiently high representativeness (that is, number of patients ≥15) in the metastatic cohort, 27 in the primary dataset and 20 cancer types with sufficient representation in both datasets (Fig. 2b, Extended Data Fig. 2b,c and Supplementary Table 2).

GIE prevalence showed high mechanistic and frequency variability across primary and metastatic cancer types (Fig. 2c,d, top panels and Supplementary Data 2). The median proportion of patients harboring GIE alterations per cancer type was 0.27 for the metastatic cohort and 0.20 for primary tumors, both showing highly dispersed distributions (±0.15 s.d. and ±0.19 s.d. in metastatic and primary tumors, respectively). In certain cancer types, such as pancreatic neuroendocrine (PANET, metastatic), diffuse large B-cell lymphoma (DLBCL, metastatic) and kidney chromophobe cancer (KICH, primary), GIE was present in >50% of patient samples (65%, 55% and 74%, respectively) whereas in others, such as lung neuroendocrine (LUNET, metastatic), GIE was an extremely rare event. Overall, one in four patients (26% in metastatic and 24% in primary) presented GIE alterations based on the six investigated pathways (Fig. 2c,d, bottom panels).

The most frequent GIE alteration was partial loss of the *HLA-I* locus (including both LOH of *HLA-I* and homozygous deletions of *HLA-I* genes that were grouped as LOH of *HLA-I* for simplicity), which was present in 783 (18%) of metastatic and 319 (17%) of primary cancer patients, followed by IFN-γ inactivation (4% in metastatic and 3% in primary) and alterations in the antigen presentation pathway (4% in metastatic and 3% in primary). CD58 inactivation was the least frequent immune escape event present in only 16 metastatic and 8 primary patients. The high GIE rates of KICH and PANET were exclusively due to LOH of *HLA-I* (Fig. 2e), whereas other cancer types displayed a wider range of GIE mechanisms (Fig. 2f). Of note, we did not observe a significant mutual exclusivity between LOH of *HLA-I* and other GIE events in cancer types with sufficient representation of multiple GIE mechanisms (Supplementary Data 2). This suggests that certain tumors may require complementary GIE alterations, such as concurrent alterations that disrupt *HLA-I*-mediated neoepitope presentation and *CD58* loss[21], to effectively escape immune surveillance.

### High agreement between primary and metastatic GIE rates

We next sought to investigate whether there was a GIE prevalence difference between early stage primary and late-stage metastatic tumors. Comparison by tumor type across the 20 cancer types with sufficient representation showed a broad agreement between both stages (Fig. 3a). Although nine cancer types showed a certain degree of metastatic enrichment ($\log_2$(odds ratio) ($\log_2$(OR)) > 0.5; Fig. 3a,b), only in prostate carcinoma (PRAD) and thyroid cancer (THCA) was this difference statistically significant (Fisher's exact test corrected $P < 0.01$). The significant enrichment in these two cancer types might be connected to the substantial genome transformation at the metastatic transition[20].

Breaking down pathway-specific differences revealed that THCA metastatic enrichment is the result of increased LOH of *HLA-I* incidence, whereas the discrepancies in PRAD are the result of a widespread enrichment across several pathways (Fig. 3b). In general, LOH of *HLA-I* showed a nonsignificant trend toward metastatic enrichment across seven of the nine metastatic-enriched cancer types. None of the cancer types showed a significantly higher GIE incidence in primary tumors.

### Positive selection of *HLA-I* alterations

We next examined to what extent somatic alterations in *HLA-I* genes (that is, *HLA-A*, *HLA-B* and *HLA-C*) were positively selected during tumorigenesis.

First, a pan-cancer-grouped *HLA-I* analysis revealed a nonsynonymous:synonymous substitution (dN:dS) ratio >1 for nonsense, splice site and truncating variants in both the metastatic and the primary datasets (Fig. 4a), indicating that these genes are subject to positive selection. Next, pan-cancer and gene-specific dN:dS ratios showed that *HLA-A* and *HLA-B*, but not *HLC-C*, are positively selected and are mostly enriched in truncating variants but not in missense mutations (Fig. 4b,c). Finally, gene and cancer type-specific analysis showed that *HLA-A* and *HLA-B* were deemed as drivers across several cancer types, including metastatic colorectal, NSCLC and DLBCL as well as the pan-cancer cohorts (Fig. 4d,e and Supplementary Data 3).

Somatic point mutations and small indels (insertions and deletions) of *HLA-I* genes were evenly distributed along their sequences (Fig. 4f and Extended Data Fig. 3a). The main exception was the recurrent *HLA-A* Lys210 frameshift indel (chromosome 6 at position 29911899), which was observed in six mismatch repair-deficient (MMRd) metastatic tumors. This genomic region overlaps with a $(C)_7$ homopolymer repeat, which probably explains its susceptibility for the observed base indel. No enrichment for mutations in amino acids involved in the peptide binding was observed. Such uniform distribution was in agreement with previous observations[22] and with the expected profile in tumor-suppressor genes dominated by inactivating variants[23].

LOH of *HLA-I* trims the repertoire of *HLA-I*-presented epitopes in *HLA-I* heterozygous individuals. Therefore, to further shed light on the tumorigenic role of LOH of *HLA-I*, we developed a randomization strategy that pinpoints cancer types where the LOH of *HLA-I* rates were significantly higher than the expected, given their background LOH rates using three genomic resolutions (that is, nonfocal LOH including all LOH events spanning >75% of the chromosome arm length, focal LOH for those events <75% of the chromosome arm and highly focal LOH for LOH events <3 Mb). In spite of the global correlation with background genome-wide LOH rates (Extended Data Fig. 3b), our analyses revealed higher-than-expected rates of LOH of *HLA-I* across several cancer types in both the metastatic and the primary datasets (G-test goodness of fit $q$ value <0.1; Fig. 4d,e and Supplementary Data 3). PANET (Fig. 4g) and KICH (Extended Data Fig. 3c) showed nonfocal LOH of *HLA-I* enrichment. Others, such as metastatic cervix carcinoma (Fig. 4h), metastatic colorectal cancer (Fig. 4i) or primary DLBCL, showed focal or highly focal LOH of *HLA-I* patterns. Furthermore, 33 patients with nonsynonymous mutations of *HLA-I* genes (20% of the total 159 patients with mutations in *HLA-I* genes) displayed the

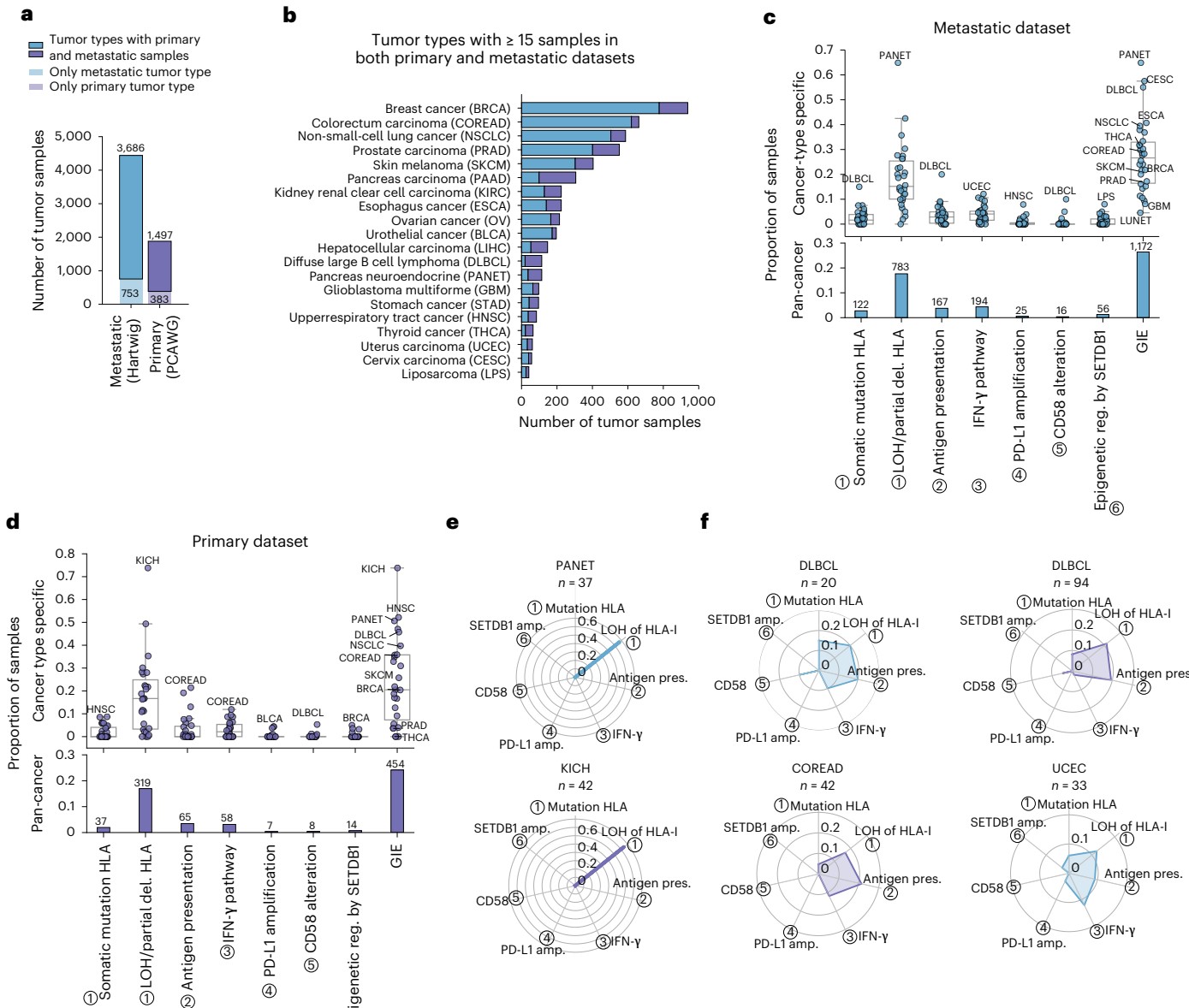

**Fig. 2 | GIE prevalence across cancer types. a,** Total number of uniformly processed WGS samples included in the study from the metastatic (Hartwig) and primary (PCAWG) datasets. **b,** Number of processed samples from each cohort across cancer types with at least 15 samples in both datasets. **c,** Top, cancer type-specific proportion of metastatic samples with GIE alterations across the six pathways and the combined group. Bottom, pan-cancer proportion and number of samples with GIE alterations in the metastatic group. **d,** Analogous for the primary dataset. Boxplots: the center line is the median, the box limits the first and third quartiles and the whiskers the lowest/highest datapoints at the first quartile ± 1.5 × the interquartile range (IQR). The ticks on the x axis label numbers representing the associated immune escape pathway, relative to Fig. 1a. **e,** Radar plots representing, using a shaded area, the cohort and cancer type-specific fraction of samples with GIE alterations in metastatic PANET (top) and primary kidney chromophobe (KICH, bottom) tumors across the six pathways from Fig. 1a. **f,** Analogous representation for, from top-left in a clockwise direction, metastatic DLBCL, primary DLBCL, primary COREAD and metastatic UCEC tumors. amp., amplification; del., deletion; pres., presentation; reg., regulation. The remaining cancer-type acronyms are displayed in **b.**

concurrent loss of the alternative allele by LOH, potentially leading to complete inactivation of the *HLA* gene.

Finally, we did not observe any biallelic deletion of the entire *HLA-I* locus (Supplementary Data 3), suggesting that homozygous deletions within the *HLA-I* might be constrained by purifying selection, featuring the importance of expressing a minimal amount of *HLA-I* molecules to avoid immune-alerter signals[24].

## Differences between focal and nonfocal LOH of HLA-I

Our results suggest that LOH of *HLA-I* is a positively selected genomic event in certain tumor types. However, it remains unclear whether these losses target a specific allele and whether both focal and nonfocal LOH of HLA events display similar selective patterns. To address these questions, we assessed whether LOH of *HLA-I* tends to involve the allele(s) with the highest neoepitope ratio (that is, higher number of predicted neoepitopes compared with the alternative allele; Fig. 5a).

We observed a positive association between the neoepitope ratio and the frequency of the allele with highest neoepitope repertoire to be lost in both the metastatic and the primary cohorts (Fig. 5b,c). This trend was significantly different from a neutral scenario where both alleles are equally likely to be lost independently of their neoepitope repertoire (Kolmogorov–Smirnov test metastatic *P* = 2.47 × 10⁻⁵

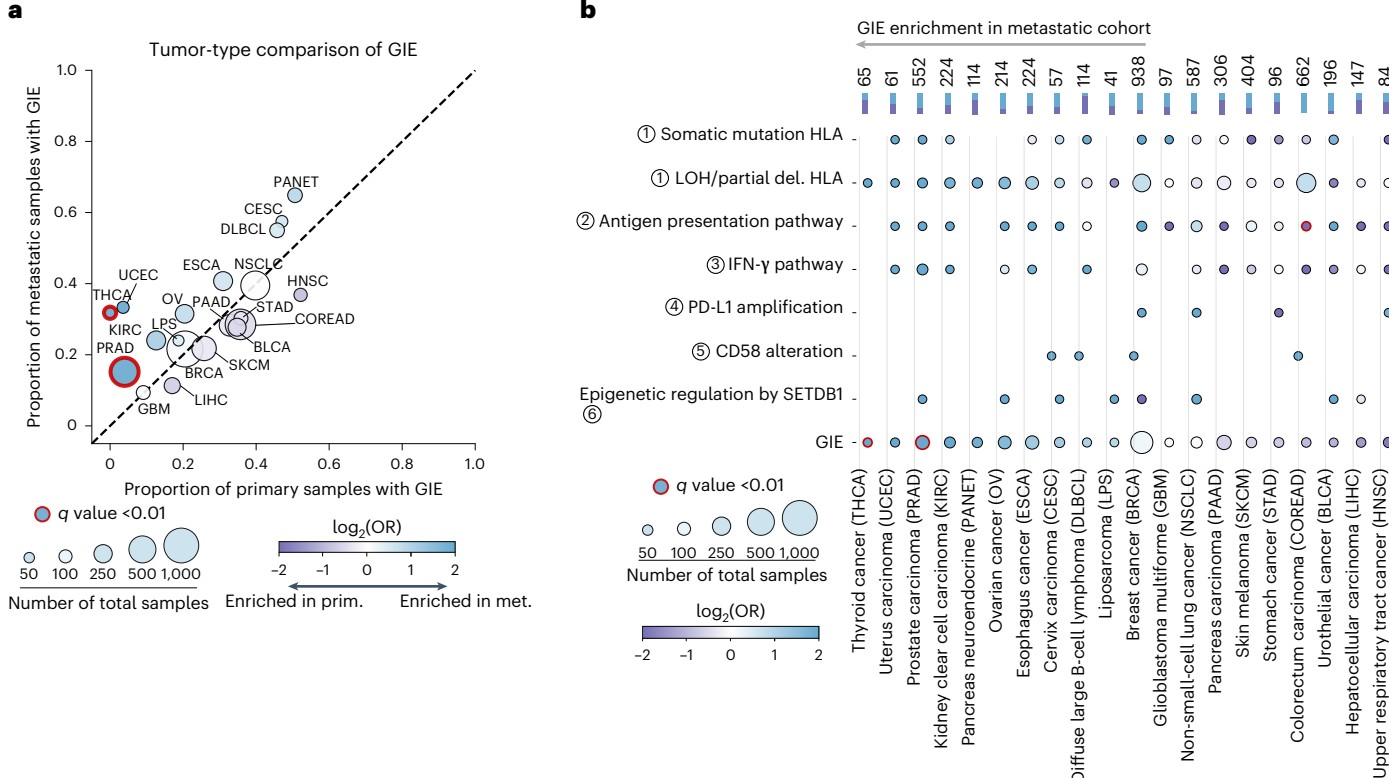

**Fig. 3 | GIE prevalence in primary and metastatic tumors. a**, Combined proportion of primary (PCAWG) and metastatic (Hartwig) samples affected by GIE alterations across 20 cancer types. The definitions of cancer type acronyms are displayed in **b. b**, Top, stacked bars, number and proportion of combined (metastatic (met.) and primary (prim.)) cancer-type samples; main, pathway-specific GIE frequency comparison alongside its statistical significance. In both panels the size of the dots is proportional to the number of total samples, dot colors are proportional to the $\log_2(OR)$ and the red edge lines represent a false discovery rate-adjusted, two-sided Fisher's exact test: $P < 0.01$.

and primary $P = 2.24 \times 10^{-7}$). Remarkably, the association between neoepitope ratio and the loss frequency became stronger when selecting for focal LOH of *HLA-I* events (Fig. 5d,e; $P = 1.71 \times 10^{-9}$ and $P = 1.17 \times 10^{-10}$ for metastatic and primary, respectively). However, it was indistinguishable from a neutral scenario for nonfocal LOH of *HLA-I* (Fig. 5f,g; $P = 0.32$ and $P = 0.99$ for metastatic and primary, respectively), showing that nonfocal LOH of *HLA-I* does not select for the allele with the highest neoepitope repertoire and that its high recurrency in several cancer types may be associated with other selective forces operating on chromosome 6.

Furthermore, the majority of focal LOH of *HLA-I* events were CN neutral (81% in metastatic tumors and 70% in primary), which was considerably higher than for nonfocal events (65% in metastatic and 35% in primary), providing further support for the notion that the loss of neoepitope repertoire, and not gene dosage, is the main driving force behind focal LOH of *HLA-I*.

**Positive selection of GIE alterations beyond *HLA-I***

Alterations in other pathways beyond the HLA-I locus may also lead to immune escape. Hence, we explored signals of positive selection across 18 genes associated with 5 immune escape pathways (pathways 2–6 in Fig. 1a).

Grouped pan-cancer analysis of the dN:dS ratio in these pathways (covering a total of 16 genes, excluding those with an oncogenic mechanism based on CN amplification; Methods) revealed a >1 ratio for nonsense, splice site and truncating variants in both the metastatic and the primary datasets (Fig. 6a), which was indicative of positive selection.

Refining the analysis for specific genes and cancer types revealed that two genes from the antigen presentation pathway

(that is, *B2M* and *CALR*) displayed recurrent patterns of inactivating mutations and focal biallelic deletions across several tumor types, as well as in the pan-cancer cohorts (Fig. 6b–d). Moreover, higher-than-expected frequencies of focal biallelic deletions for several IFN-γ pathway genes, including *JAK1*, *JAK2* and *IRF2*, were also observed. *CD58* also harbored a higher-than-expected number of nonsynonymous mutations and homozygous deletions in DLBCL and the pan-cancer primary cohort. Finally, the chromatin modifier *SETDB1* was recurrently focally amplified in multiple cancer types, including metastatic NSCLC (Fig. 6e) and primary breast cancer. Full results are available in Supplementary Data 3.

**GIE association with cancer genomic features**

We next investigated whether, aside from cancer-type intrinsic differences, there were other cancer genomic and environmental features associated with GIE prevalence. Thus, we performed a cancer type-specific univariate logistic regression of 99 tumor genomic features and 366 driver genes against the presence of GIE events (excluding nonfocal LOH of *HLA-I*) across 38 cancer types (Supplementary Note 3). Moreover, to control for associations that may be secondary to increased mutation and CN variant (CNV) background rates, we filtered out significant associations that were found in our GIE simulations (Supplementary Note 3).

Overall, 35 genomic features and 5 driver genes showed a statistically significant association with GIE in at least one cancer type (Fig. 7a and Extended Data Fig. 4a). Even after controlling for background mutation rates, TMB and patient's neoepitope load were strongly associated with GIE events in DLBCL, pancreas carcinoma and skin melanoma ($q$ value < 0.05, $\log_2(OR) > 0.0$ and simulated GIE prevalence ≤2%; Fig. 7a

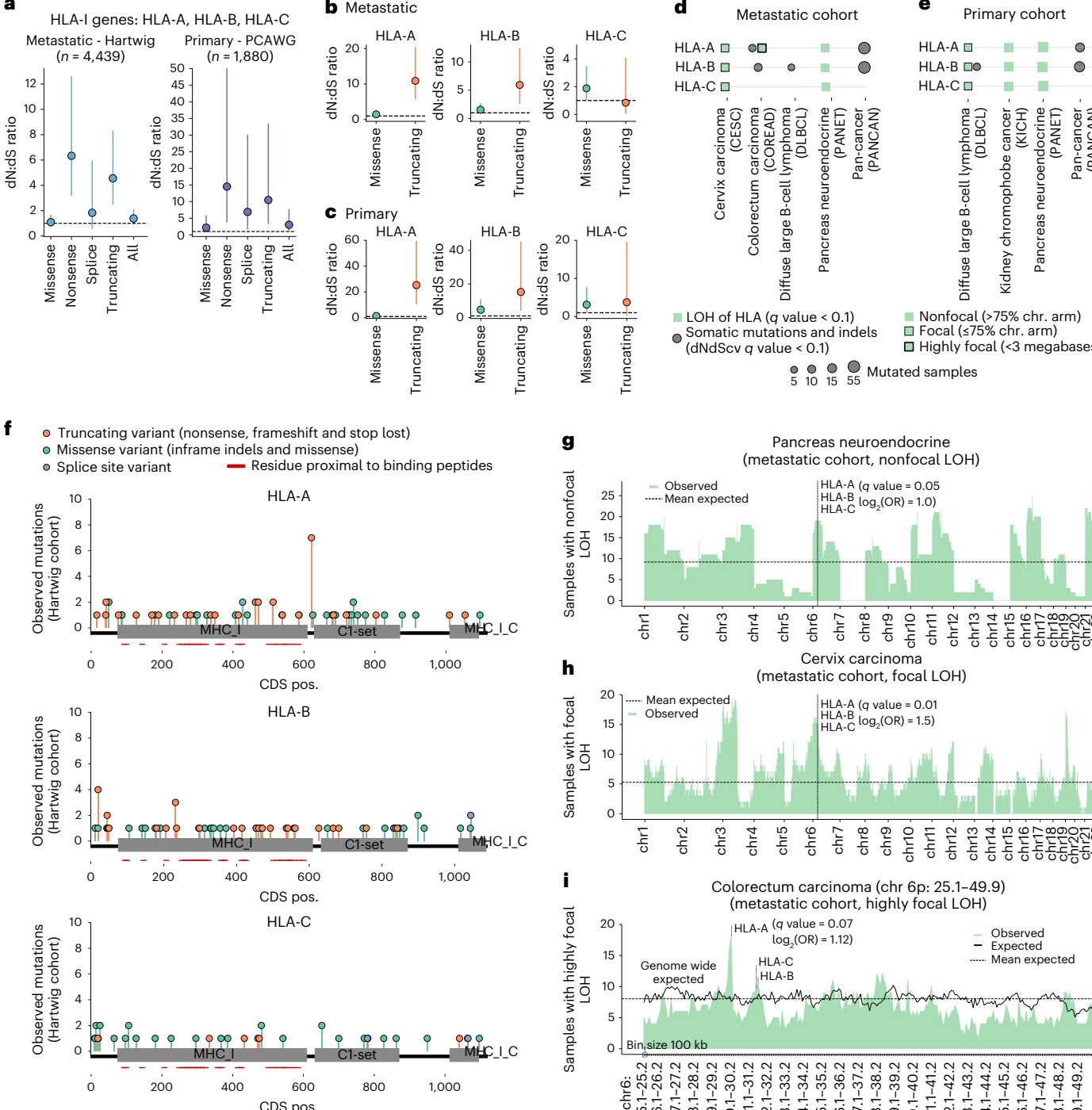

**Fig. 4 | Positive selection of *HLA-I* genes. a**, Pan-cancer dN:dS ratios of *HLA-I* genes in the metastatic (left) and primary (right) dataset. The vertical lines represent the 5% and 95% confidence intervals (CIs) after ten randomizations, dots the maximum likelihood estimates and *n* the number of samples. **b**, Metastatic pan-cancer and gene-specific dN:dS ratios of *HLA-I* genes. The vertical lines represent the 5% and 95% CIs after ten randomizations and the dots the maximum likelihood estimates. **c**, Similar to **b** for the primary dataset. **d,e**, Representation of gene and cancer type-specific positive selection of *HLA-I* in the metastatic (**d**) and primary cohorts (**e**). **f**, Needle plots representing the pan-cancer distribution of somatic mutations along the HLA-A, HLA-B and HLA-C protein sequences in the metastatic dataset. Mutations are colored

according to the inferred consequence type. Rectangles represent the Pfam[36] domains. **g**, Distribution of nonfocal LOH events along the autosomes in PANET tumors of the metastatic cohort (ticks on the *x* axis represent the chromosomal starting position). **h**, Distribution of focal LOH events along the autosomes in cervical cancer tumors of the metastatic cohort (ticks on the *x* axis represent the chromosomal starting position). **i**, Distribution of highly focal LOH events surrounding the HLA-I locus, spanning chromosome (chr) 6 from 25.1-Mb to 49.9-Mb genomic locations in the metastatic colorectal cancer cohort. Each bin represents 100 kb. Dashed horizontal lines represent the expected mean after randomization and vertical dashed lines highlight the *HLA-I* genomic locations. CDS pos., coding sequence position.

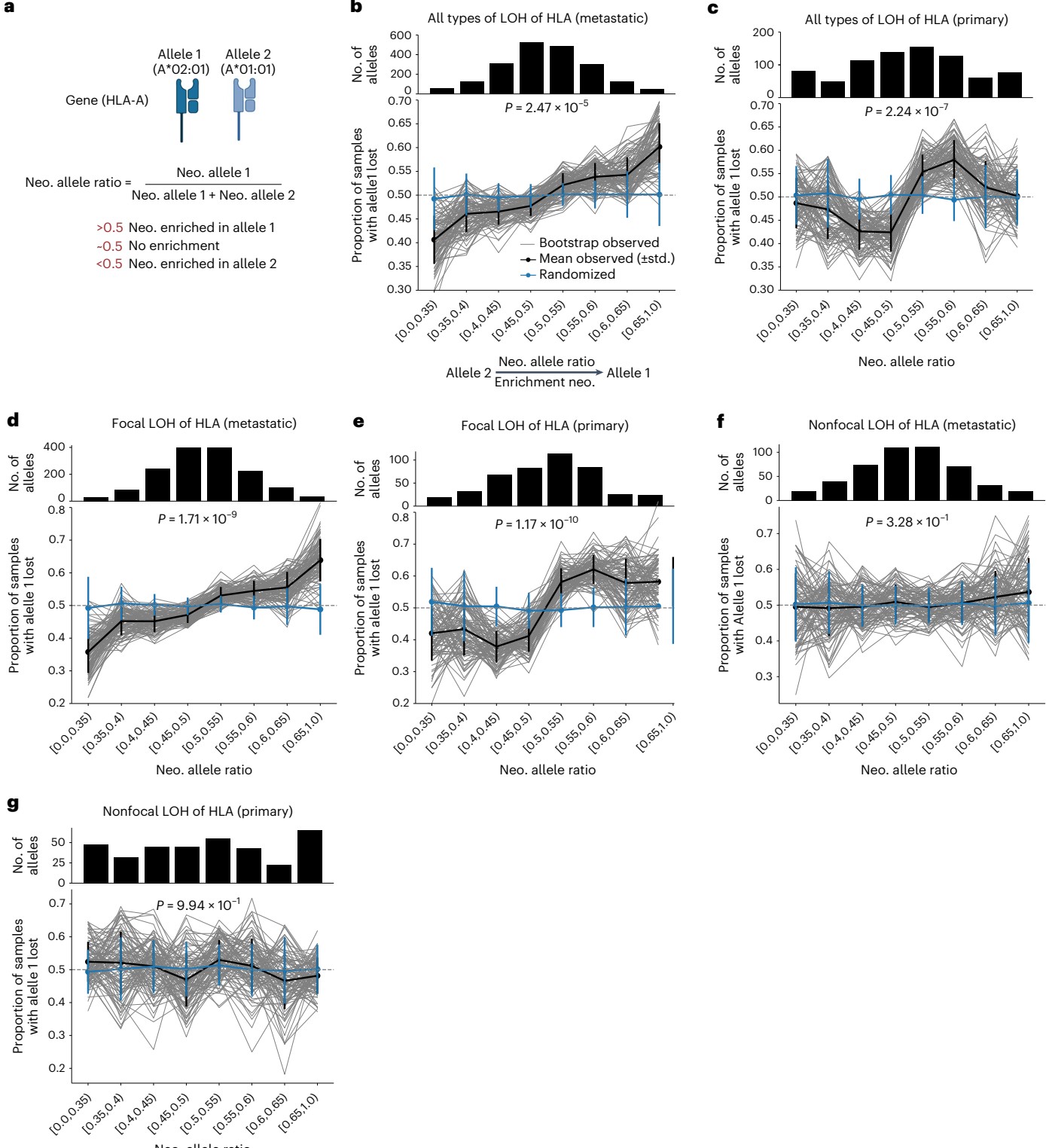

**Fig. 5 | LOH of *HLA-I* and neoepitope load. a**, Visual depiction of the neoepitope (neo.) allele ratio and its significance (partially created with BioRender.com). **b**, Top, number of allele pairs in each neoepitope allele ratio bucket in metastatic samples harboring LOH of *HLA-I*. Bottom, representation of the mean observed (black) and randomized (blue) neoepitope allele ratio across 100 bootstraps (thicker lines). The vertical error bars represent the s.d. of the neoepitope allele ratio. The narrow black lines represent the observed neoepitope allele ratio values across the 100 bootstraps. *P* values were calculated using the two-sample Kolmogorov–Smirnov test for goodness of fit. **c**, Analogous to **b** for the primary PCAWG dataset. **d**,**e**, Similar representation but subsampling for focal LOH of *HLA-I* in metastatic (Hartwig) (**d**) and primary (PCAWG) (**e**) datasets, respectively. **f**,**g**, Similar representation but subsampling for nonfocal LOH of *HLA-I*: metastatic (Hartwig) (**f**) and primary (PCAWG) (**g**) tumor datasets.

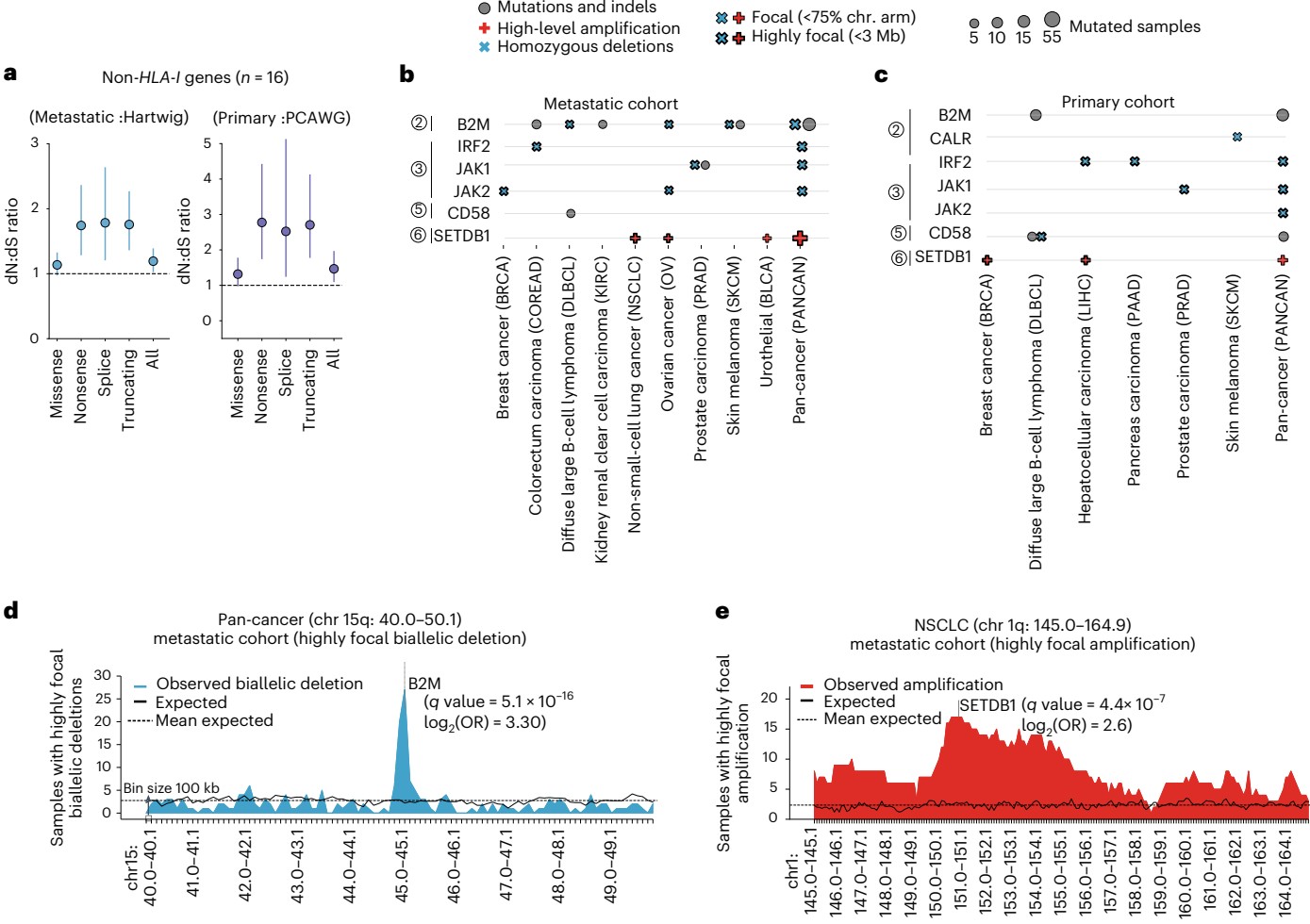

**Fig. 6 | Positive selection of GIE events beyond the *HLA-I*. a**, Pan-cancer dN:dS ratios of non-*HLA-I* genes in the metastatic (left) and primary (right) datasets. The vertical lines represent the 5% and 95% CIs after ten randomizations, and the dots the maximum likelihood estimates. **b,c**, Representation of gene and cancer type-specific, positive selection of non-*HLA-I* GIE-associated genes in the metastatic (Hartwig) (**b**) and primary (PCAWG) (**c**) cohorts. The pathway number attributed to each gene is displayed next to the gene name, relative to Fig. 1a.

**d**, Distribution of highly focal biallelic deletions surrounding the *B2M* gene, spanning chromosome (chr.) 15 from 40.0-Mb to 50.1-Mb genomic location in the pan-cancer metastatic cohort. **e**, Distribution of highly focal CN amplification surrounding the *SETDB1* gene, spanning chr. 1 from 145.0-Mb to 164.9-Mb genomic location in the metastatic NSCLC cohort. Each bin represents 100 kb. The dashed horizontal lines represent the expected mean after randomization and the vertical dashed lines highlight the gene genomic location.

and Extended Data Fig. 4a–c). It is interesting that clonal TMB and clonal neoepitope load showed a strong positive association with GIE, whereas subclonal TMB and neoepitope load showed a modest correlation (Fig. 7a–c), highlighting the relevance of mutation cellularity in triggering immune responses. Finally, fusion-derived neoepitopes were significantly associated with GIE in DLBCL and NSCLC (Fig. 7a), which emphasizes the importance of considering noncanonical sources of neoepitopes beyond small nonsynonymous variants in coding regions.

Exposure to certain endogenous and exogenous mutational processes have been correlated with increased immunogenicity[25] and response to ICIs[26,27]. After controlling for molecular age and excluding non-GIE exclusive associations (that is, associations also observed in the GIE simulations) several mutational processes showed significant association with GIE incidence (Fig. 7a and Extended Data Fig. 4a). First, MMRd mutational processes were broadly associated with increased GIE incidence. Similarly, exposure to the APOBEC family of cytidine deaminases was strongly associated with GIE in multiple cancer types, including breast carcinomas (Fig. 7d and Extended Data Fig. 4d). Last, the mutation burden associated with several exogenous mutational processes, such as ultraviolet light in skin melanoma (Fig. 7e and Extended Data Fig. 4e) and platinum treatment in NSCLC

(Fig. 7f and Extended Data Fig. 4f), was also significantly linked to an increased incidence of GIE events in these cancer types.

We also identified other tumor genomic features that were correlated with GIE. For instance, in colorectal cancers, which also include some patients with anal cancer, human papillomavirus DNA integration was positively associated with GIE incidence (Fig. 7a). Moreover, high-immune infiltration, as determined by several RNA-sequencing-based deconvolution measurements (Supplementary Note 3), was significantly linked with higher GIE incidence in this cancer type (Fig. 7g and Extended Data Fig. 4g), which is in agreement with previous reports[7].

Certain driver alterations, beyond the GIE pathways considered in the present study, also showed a strong association with GIE events. Specifically, *CASP8*, *KMT2D*, *RPL22* and *TGFBR2* alterations tended to co-occur with GIE in patients with colorectal cancer. Of note, *CASP8* (ref. [13]) and *TGFBR2* (ref. [28]) alterations have previously been linked to immune surveillance escape.

Finally, other factors, such as the *HLA-I* supertype, the germline *HLA-I* divergence, patient chronological age or exposure to previous treatments, including immunotherapy, failed to attain significant association with GIE (or the association was also observed in the

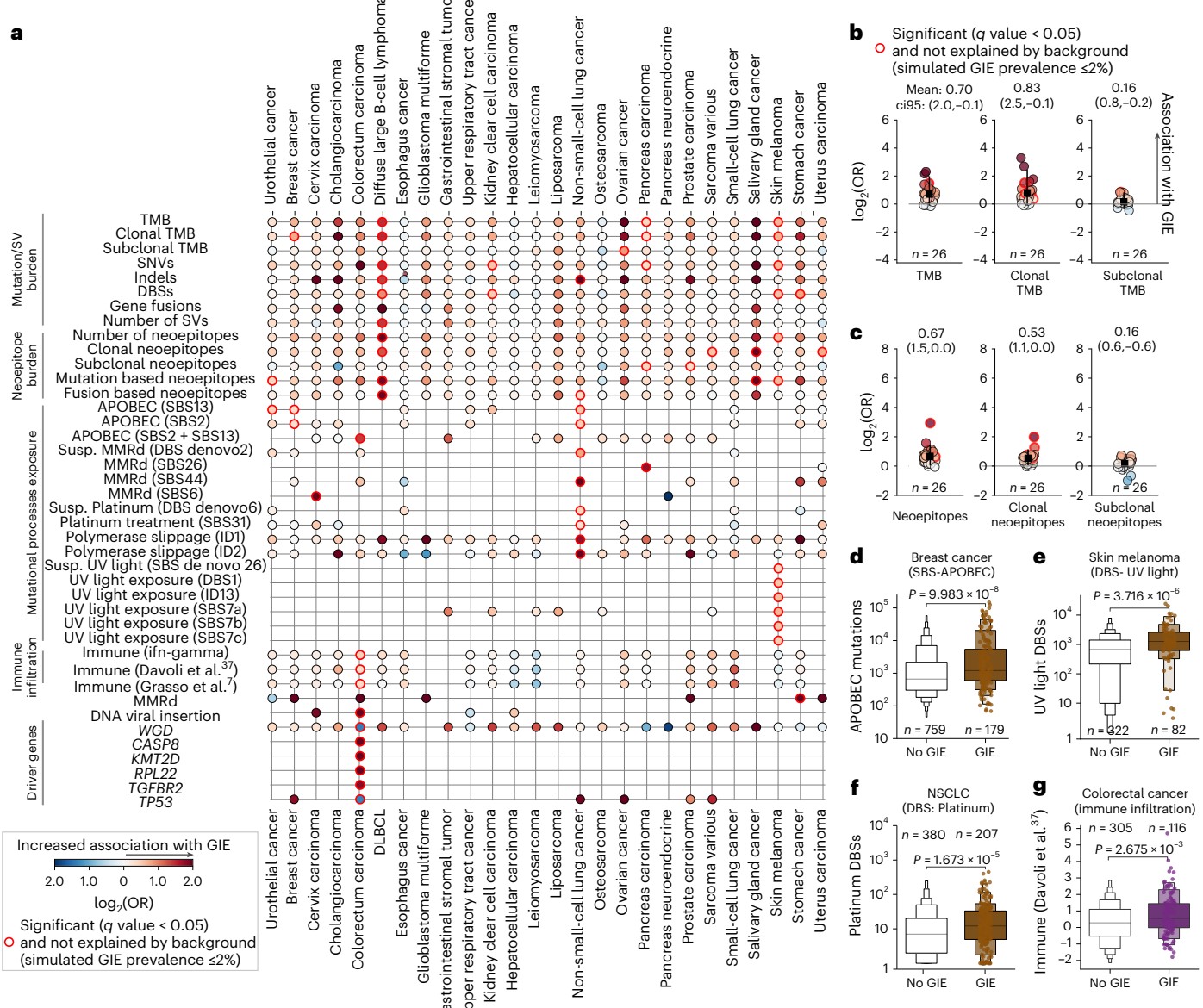

**Fig. 7 | GIE association with cancer genomic features. a**, Heatmap displaying the association of 40 genomic features with GIE frequency across 26 cancer types. The features displayed have, at least, one significant cancer-type association with GIE alterations. Significant associations that cannot be explained by higher background mutation rate are highlighted by a red border. Dot colors are colored according to the $\log_2(OR)$. UV, ultraviolet. **b**, Left to right, dot plot representations of the TMB, clonal TMB and subclonal TMB $\log_2(OR)$ across the 26 cancer types. The black square represents the mean values and the error bar the 95% and 5% CIs across the 26 cancer types. The horizontal lines represent a neutral scenario with $\log_2(OR) = 0$ (*n* is the number of cancer types). **c**, Analogous representation for predicted neoepitopes, clonal neoepitopes

and subclonal neoepitopes. **d**, Comparison of the APOBEC mutational exposure between samples bearing GIE alterations (GIE) and wild-type (no GIE) in breast cancer. **e,f**, Analogous comparison for UV-induced double-base substitutions (DBSs) in skin melanoma (**e**) and for platinum treatment-attributed DBSs in NSCLC (**f**). **g**, Comparison of immune infiltration estimates from Davolit et al.[37] between samples bearing GIE alterations (GIE) and wild-type (no GIE) in colorectal cancer. Boxplots: the center line is the median and the first section out from the center line contains 50% of the data. The next sections contain half the remaining data until we are at the outlier level. Each level out is shaded lighter. *P* values of the boxplots are calculated using a two-sided Mann–Whitney *U*-test. SBS, single base substitution; Suspect., suspected.

simulated GIE). All the screened molecular features alongside their cancer type-specific significance coefficients are available in Supplementary Data 4.

**The selected immune evasion mechanisms depends on TMB**

An increase in mutational load leads to the generation of neoepitopes susceptible to recognition as neoantigens by the adaptive immune system. Therefore, we investigated the relationship between the frequency of GIE alterations (excluding nonfocal LOH of *HLA-I*) and

the TMB across 20 evenly distributed TMB buckets (Methods). We first observed that GIE frequency steadily increased with the TMB (Fig. 8a; observed GIE) and that this trend was not fully explained by an increased background mutation and CNV rate (Fig. 8a; simulated GIE). More specifically, as the TMB increases, the observed GIE frequency deviates from the expected frequency given by the GIE simulations. This is particularly noticeable for (ultra)hypermutated tumors, which showed a GIE incidence two- to threefold higher than the simulations. This trend was still consistent after controlling for the cancer

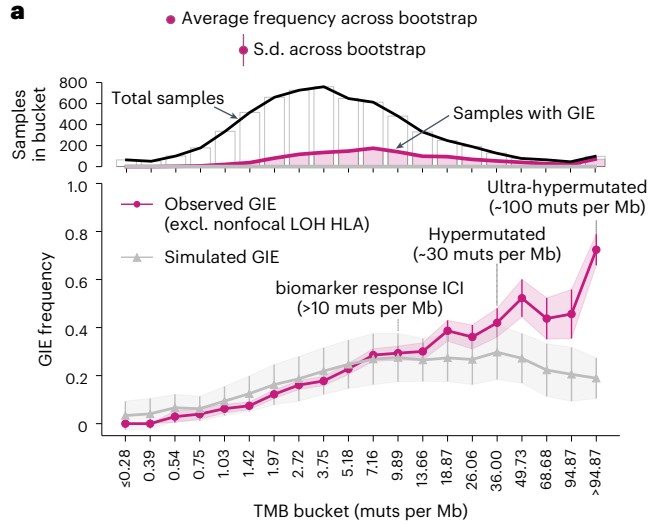

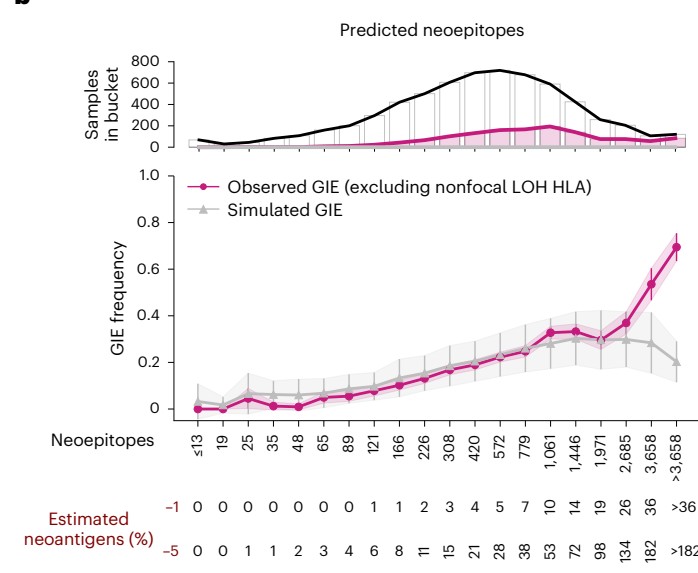

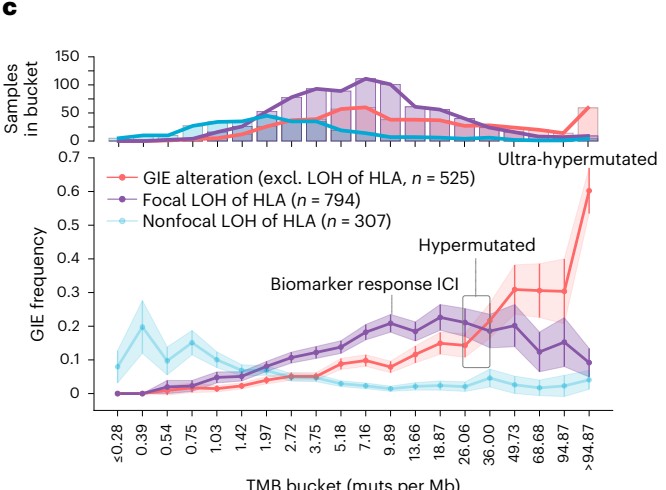

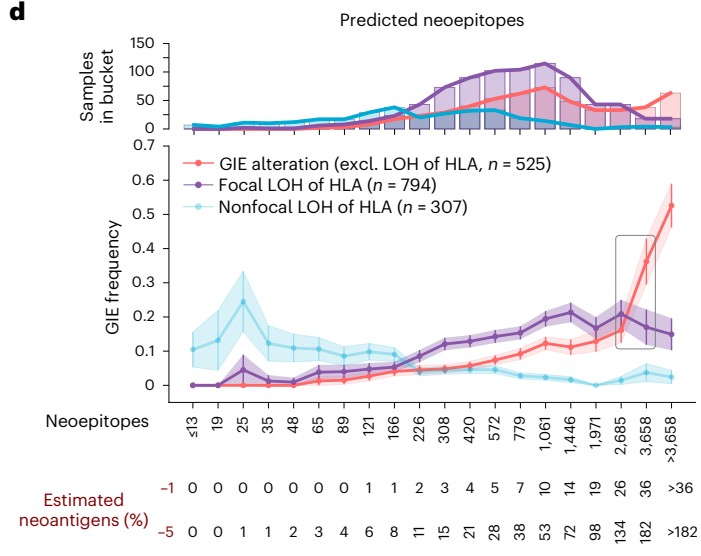

**Fig. 8 | Immune evasion mechanisms and TMB. a**, Top, number of bucket-assigned (white bars with a black contouring line) and GIE-positive (pink bars with pink contouring line) samples across 20 evenly distributed TMB buckets using the entire cohort (*n* = 6,319). Bottom, representation of observed (pink) and simulated (gray) GIE frequency across these buckets. For the observed GIE values, the average (represented as pink dots) and s.d. (vertical error bars and shaded pink area) values are computed using 1,000 bootstraps from the total number of samples classified into each bucket (from the top panel). For the simulated GIE values, average (gray triangle) and s.d. (vertical bars and shaded gray area) values are computed from 100 GIE simulations using the total number of samples assigned into each bucket. **b**, Analogous representation but using predicted neoepitopes as baseline for the buckets. Bottom, number of estimated neoantigens as a relative percentage (1% and 5%) of the number of predicted neoepitopes in the bucket. **c,d**, Related to **a** (**c**) and **b** (**d**), respectively, but splitting by type of *HLA-I* alteration. Each dot/line is colored according to the type of GIE event. The inner boxes highlight the bucket where non-LOH of *HLA-I* frequency (red) surpasses focal LOH of *HLA-I* (purple). Excl., excluding; muts/Mb, mutations per megabase.

type (Extended Data Fig. 5a) and mutation clonality (Extended Data Fig. 5b). Using the burden of predicted neoepitopes based on the germline *HLA-I* profile as baseline also revealed an almost uniformly increasing distribution across the neoepitope buckets, which becomes sharper and higher than expected after the 17th bucket (Fig. 8b).

It is interesting that, in the bucket grouping samples with ~10–13 mutations per Mb, which is the minimal threshold regularly used as a response to ICIs, we observed an average GIE frequency of 0.30 ± 0.03 s.d. Similarly, in the group of samples between 26 and 36 mutations per Mb, mostly including hypermutated tumors, the average frequency was 0.42 ± 0.06 s.d., whereas beyond ~95 mutations

per Mb (considered to be ultra-hypermutated tumors[29]) we identified GIE alterations in >70% of samples (0.72 ± 0.06 s.d.). Our results thus showed that an important fraction of patients eligible for ICIs harbored tumor alterations that may hinder recognition and/or elimination by the immune system.

We then analyzed the relationship between the TMB and the presence of specific GIE alterations across the six immune escape pathways included in the present study. Overall, the observed frequency distributions across these pathways were remarkably different (Fig. 8c and Extended Data Fig. 5c). In fact, different types of *HLA-I* alterations showed a distinctive frequency distribution along the

TMB buckets. Nonfocal LOH of *HLA-I* was primarily present in low-TMB tumors, whereas focal LOH of *HLA-I* showed a clear enrichment for mid and high TMB tumors, peaking around ~10–20 mutations per Mb (average frequency of 0.22 ± 0.04 s.d.) and displaying an inverted U-shaped distribution. Finally, mutations in *HLA-I* genes were more frequent in hypermutated tumors (that is, from ~26 mutations per Mb to 36 mutations per Mb). Similarly, alterations in the antigen presentation machinery and the IFN-γ pathway were predominantly found in hypermutated tumors (Extended Data Fig. 5c). The remaining pathways did not show any clear TMB preference, probably due the lower prevalence of these alterations in our dataset. Finally, using the number of predicted neoepitopes as baseline revealed consistent distributions (Fig. 8d and Extended Data Fig. 5d).

## Discussion

In the present study, we have characterized the prevalence and impact of GIE alterations involved in six major pathways across thousands of uniformly processed primary and metastatic tumors from fifty-eight cancer types. Moreover, we addressed the complexity of identifying tumor-specific *HLA-I* alterations by developing LILAC.

Our results revealed that, on average, one in four patients bears a GIE event, primarily as a result of LOH of *HLA-I*. However, GIE incidence and the targeted pathways showed high diversity across cancer types. Importantly, the fact that we did not observe mutual exclusivity between GIE alterations targeting different pathways suggests that multiple GIE alterations may concur to effectively avoid immune surveillance.

Remarkably, our analyses also showed that the frequency of GIE alterations in metastatic patients are comparable to their primary counterparts across most cancer types. This result is also supported by independent studies[6,30], denoting that early stages of tumorigenesis have already acquired the capacity to escape from immune system recognition.

Immune escape alterations were often positively selected during tumor evolution. Specifically, loss-of-function mutations in *HLA-A* and *HLA-B*, as well as multiple genes from other immune escape pathways, displayed higher-than-expected frequencies across several cancer types. Nevertheless, *HLA-C* did not show a significant enrichment in inactivating variants which may imply that its expression is needed to avoid natural killer-mediated immunity[31] and that the neoepitope repertoire of this gene is generally lower compared with *HLA-A* and *HLA-B*. Finally, we also observed higher-than-expected LOH of *HLA-I* rates across multiple cancer types.

Related to this, focal and nonfocal LOH of *HLA-I* undergo divergent mechanisms of selection. Focal LOH of *HLA-I* was primarily a CN-neutral event that tended to target the HLA allele with the largest neoepitope repertoire, indicating an active role in immune evasion. On the contrary, we did not observe such allelic preference for nonfocal LOH of *HLA-I*, suggesting that alternative selective forces, such as *DAXX* haploinsufficiency[32], are operating in these large-scale chromosome 6 events.

Multiple tumor intrinsic and extrinsic features displayed a significant association with increased GIE incidence. However, in our cohort, a patient's exposure to previous cancer therapies, including immunotherapies, did not attain a significant association with GIE frequency, indicating that the efficacy of GIE alterations may be compromised when dealing with the strong immune pressure released by ICIs.

The tumor mutation and neoepitope burden influenced both the GIE frequency and the targeted GIE pathway. Although focal LOH of *HLA-I* was the most frequent mechanism in mid and high TMB tumors, the loss of certain *HLA-I* alleles was apparently not sufficient to cope with the neoepitope load of (ultra)hypermutated tumors, where a nontargeted GIE mechanism, such as antigen presentation abrogation, is probably needed. However, we cannot rule out the fact that such differences may also be partially shaped by mutation and CNV rate differences across cancer types. It is important to mention that the GIE escalation as the TMB increases was not entirely attributed to the underlying increase in background mutation rate, particularly in hypermutated tumors. Although the modeling of background GIE rates could be sensitive to the selected randomization strategy, our results are supported by independent studies based on orthogonal analytical approaches[30], evidencing the robustness of our conclusions.

The present study considered a collection of highly confident GIE alterations across six well-characterized, immune-related pathways. However, in our dataset, three of four patients did not harbor GIE events, highlighting the need to characterize other mechanisms of immune evasion. These may involve not only alternative molecular pathways such as the *HLA-II* (ref. 33), but also other types of alterations such as germline variants[34] and epigenetic modifications[5,35]. Finally, tumor extrinsic factors such as clonal hematopoiesis, tumor-associated microbiome or the tissue architecture may also play an important role in tumor immune evasion. We expect that the combination of cancer genomics with high-resolution characterization of the tumor microenvironment will aid in further understanding of the interplay between tumor evolution and the immune system.

## Online content

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

## Methods

### Data collection and processing

The Hartwig Medical Foundation sequences and characterizes the genomic landscape for a large number of patients with metastases. A detailed description of the consortium and the whole patient cohort has been given in detail in Priestley et al.[16]. In the present study, the Hartwig cohort included 4,784 metastatic tumor samples from 4,468 patients.

The Hartwig patient samples have been processed using the Hartwig analytical pipeline5 (https://github.com/hartwigmedical/pipeline5) implemented in Platinum (v.1.0) (https://github.com/hartwigmedical/platinum). Briefly, Platinum is an open-source pipeline designed for analyzing WGS tumor data. It enables a comprehensive characterization of tumor WGS samples (for example, somatic point mutations and indels, structural variants, CN changes) in one single run.

Hartwig samples that failed to provide a successful pipeline output, potential nontumor samples, with purity <0.2, with TMB < 50 SNVs/indels, lacking sufficient informed consent for the present study or without enough read coverage to perform two-field HLA typing (Supplementary Note 1) were discarded. Similarly, for patients with multiple biopsies, we selected the tumor sample with the most recent biopsy date and, if this information did not exist, we selected the sample with the highest tumor purity. However, some Hartwig patients had biopsies from different primary tumor locations. In these cases, we kept at least one sample from each primary tumor location and, when there were multiple samples from the same primary tumor location, we applied the aforementioned biopsy date and tumor purity-filtering criteria. A total number of 4,439 Hartwig samples were whitelisted and used in the present study (Extended Data Fig. 2a and Supplementary Table 2).

Preprocessed RNA-sequencing (RNA-seq) data by ISOFOX (https://github.com/hartwigmedical/hmftools/tree/master/isofox) were available for 1,864 Hartwig samples and were consequently used in the immune infiltration deconvolution analysis.

Patient clinical data were obtained from the Hartwig database. Cancer-type labels were harmonized to maximize the number of samples that had tumor types comparable with the PCAWG dataset (Supplementary Table 2).

The PCAWG cohort consisted of 2,835 patient tumors and access for raw sequencing data for the PCAWG-US was approved by the National Institutes of Health (NIH) for the dataset General Research Use in The Cancer Genome Atlas (TCGA) and downloaded via the dbGAP download portal. Raw sequencing access to the non-US PCAWG samples was granted via the Data Access Compliance Office (DACO). A detailed description of the consortium and the whole patient cohort has been given in Campbell et al.[17].

The samples were fully processed using the same cancer analytical pipeline applied to the Hartwig cohort (BWA[38] v.0.7.17, GATK[39] v.3.8.0, SAGE[16] v.2.2, GRIDSS[40] v.2.9.3, PURPLE[16] v.2.53 and LINX[41] v.1.17). This enabled a harmonized analysis and eliminated the potential biases introduced by applying different methodological approaches. Samples that failed to provide a successful pipeline output, with a tumor purity <0.2, potential nontumor samples, blacklisted by the PCAWG original publication[17] or without enough read coverage to perform two-field HLA typing were discarded. Similarly, for patients with multiple samples, we selected the first according to the aliquot ID alphabetical order. A total number of 1,880 were whitelisted and used in the present study (Extended Data Fig. 2a and Supplementary Table 2). For more details about the re-processing of the PCAWG dataset and the technical validation see Martínez-Jiménez et al.[20].

Preprocessed gene level expression data were downloaded for 1,118 samples from the International Cancer Genome Consortium (ICGC) portal (https://dcc.icgc.org/releases/PCAWG/transcriptome/gene_expression/tophat_star_fpkm_uq.v2_aliquot_gl.tsv.gz). ENSEMBL identifiers were mapped to HUGO symbols. Of these samples, 930 belonged to biopsies selected for the present study and were therefore used for the RNA analysis in PCAWG samples.

The most recent clinical data were downloaded from the PCAWG release page (https://dcc.icgc.org/releases/PCAWG) on August 2021. Cancer-type labels were harmonized to maximize the number of samples that had tumor types comparable with the Hartwig dataset (Supplementary Table 2).

### LILAC

All information relative to LILAC's algorithm, implementation and validation is described in Supplementary Note 1.

### Definitions of GIE alterations

We searched in the literature for somatic genomic alterations that are robustly and recurrently associated with immune evasion. We stratified the reported alterations into six major pathways (Fig. 1a and Supplementary Table 1):

(1) The *HLA-I*: somatic alterations in the *HLA-A*, *HLA-B* and *HLA-C* genes have been extensively reported as a mechanism for immune evasion across several cancer types[9,10,12,22]. We considered LOH of *HLA-I*, homozygous deletions and somatic nonsynonymous mutations on these genes as immune evasion alterations. We defined LOH for *HLA-A*, *HLA-B* and *HLA-C* as those cases with a minor allele ploidy <0.3 and a major allele ploidy >0.7 according to LILAC annotation. We also relied on LILAC mapping of somatic mutations into *HLA-A*, *HLA-B* and *HLA-C* alleles to report samples with nonsynonymous alterations. Finally, we also used LILAC allele-specific tumor CN estimations to annotate samples with homozygous deletions of *HLA-A*, *HLA-B* and *HLA-C* genes. A gene was homozygous deleted in a sample if the estimated minimum tumor CN of the gene was <0.5.

(2) The antigen presentation pathway: several studies have reported the immunomodulatory effect of somatic inactivation of genes involved in the antigen presentation machinery (see Supplementary Table 1 for gene-specific references). The most recurrent alteration is *B2M* inactivation, but there are other genes involved in antigen presentation and antigen presentation activation, the inactivation of which has been linked to increased immune evasion, including *CALR*, *TAP1*, *TAP2*, *TAPBP*, *NLRC5*, *CIITA* and *RFX5*. We defined inactivation events as monoallelic and biallelic clonal loss-of-function mutations (frameshift variant, stop gained, stop lost, splice acceptor variant, splice donor variant, splice region variant and start lost), biallelic clonal nonsynonymous mutations not included in the former group (for example, missense mutations) and homozygous deletions. A gene was homozygous deleted in a sample if the estimated minimum tumor CN of the gene was <0.5.

(3) The IFN-γ pathway: IFN-γ is a cytokine with known proapototic and immune booster capacities. Hence, it has been reported that tumors frequently leverage somatic alterations targeting IFN-γ receptors and downstream effectors to evade immune system surveillance (see Supplementary Table 1 for gene-specific references). More specifically, we considered that inactivation events (see above for specifics of which type of alterations are included) in *JAK1*, *JAK2*, *IRF2*, *IFNGR1*, *IFNGR2*, *APLNR* and *STAT1* have been probed to have the ability to provide an immune evasion phenotype.

(4) The PD-L1 receptor: the PD-L1 receptor, encoded by the *CD274* gene, plays a major role in suppressing the adaptive immune system. It has been reported how overexpression of PD-L1 in tumor cells leads to impaired recruitment of immune effectors[42]. We therefore considered *CD274* CN amplification as a genetic mechanism of immune evasion. We defined a *CD274* CN amplification event as samples with *CD274* minimum tumor CN >3× the average sample ploidy.

(5) The CD58 receptor: the CD58 receptor, encoded by *CD58*, plays an essential role in T-cell recognition and stimulation. It has been extensively reported that *CD58* alterations in B-cell lymphomas lead to immune evasion[21]. Moreover, a recent study identified *CD58* loss as one of the major effectors of impaired T-cell recognition[43]. Consequently, we considered inactivation events (see above) in *CD58* as alterations able to provide an immune escape phenotype.

(6) Epigenetic driven immune escape: it has been recently reported how *SETDB1* amplification leads to epigenetic silencing of tumor intrinsic immunogenicity[44]. *SETDB1* amplification was recurrently found across several cancer types and was therefore considered in the present study as a mechanism of immune evasion. We defined a *SETDB1* CN amplification event as samples with SETDB1 minimum tumor CN >3× the mean sample ploidy.

A summary table with all 21 considered genes, their associated pathway, references and their type of somatic alterations is presented in Supplementary Table 1.

## GIE mutual exclusivity

To assess whether LOH of HLA-I events were mutually exclusive with other GIE events, we performed two statistical tests. First, we performed a left-sided Fisher's exact test comparing two groups of annotations (LOH of *HLA-I* and other GIE events) in a cancer type-specific manner. Second, for each cancer type, we compared the number of samples bearing both LOH of *HLA-I* and other GIE events with the expected given by 10,000 randomization, using the observed alteration frequency of both groups in the specific cancer type (LOH of *HLA-I* and other GIE alterations). The significance was computed using an empirical one-sided *P* value (that is, number of randomizations with co-occurring events lower than the real observed value divided by the total number of randomizations).

## Primary and metastatic GIE prevalence

The prevalence of a pathway alteration for a particular cohort was calculated as the number of samples with at least one alteration in the pathway divided by the total number of cohort samples. The presence of a genetic immune alteration in a given sample was annotated if there was at least one pathway with an alteration in that sample.

For the primary versus metastatic comparison, we performed a tumor type-specific Fisher's exact test comparing pathway-specific and global escaped status prevalence across the two cohorts. *P* values were adjusted with a multiple-testing correction using the Benjamini–Hochberg procedure ($\alpha = 0.05$).

## Positive selection: somatic point mutations and indels

Positive selection analysis based on somatic point mutation and small indels was performed using dNdScv and the hg19 reference genome. The analysis was performed in a cohort-specific, cancer-type and pan-cancer manner across the two datasets. The analysis was restricted to datasets with sufficient representativeness (that is, number of samples ≥15). Global grouped dN:dS ratios of the *HLA-I* (*HLA-A*, *HLA-B* and *HLA-C*) and the 16 non-*HLA-I* genes potentially targeted by mutations (that is, excluding *SETDB1* and *CD274* because their immune escape phenotype is associated with CN gains; Supplementary Table 1) were calculated in a pan-cancer manner using the gene_list attribute of the dndscv function.

We used the geneci() function of dNdScv to estimate the pan-cancer and gene-specific dN:dS ratios, which include confidence intervals (CIs), of the *HLA-I* genes.

## Positive selection: CNAs

We devised a statistical test to assess positive selection in LOH, homozygous deletion (HD) and CN amplification (AMP) events. LOH

was defined as those genomic regions where the minor allele ploidy of this gene was <0.3 and the major allele ploidy >0.7. HD was defined as those regions with estimated minimum CN < 0.5. Similarly, AMP events were defined as those genomic regions with the minimum tumor CN >3× the mean sample ploidy.

For a particular type of genomic event overlapping with a gene, this test compares the number of observed samples bearing the alteration with the expected number after whole-genome randomization. More specifically, these are the steps followed:

(1) Let us first denote *E* as the type of query alteration (that is, LOH, HD or AMP), *S* as a group of samples (usually samples from the same cancer type and same dataset) and $G_s$ as the genomic scale (that is, nonfocal for segment lengths >75% chromosome arm, focal for segments <75% of the chromosome arm and highly focal for segments <3 Mb).

(2) For every sample $S_i$ in $\{S_1, S_2, ... S_T\}$ we first gather the number and length of observed ($O_i$) segments targeted by *E* within that $G_s$. Only *E* events overlapping with autosomes are considered in the present study. Samples that do not harbor any event of type *E* within that $G_s$ are ignored.

(3) Next, for every sample $S_i$ we performed 10 independent randomizations ($R_{i1}, R_{i2}, ... R_{i10}$) of the $O_i$ events, by randomly shuffling these events *E* along the autosomes. For this, we used the shuffle function from pybedtools[45] with the following parameters (genome='hg19', noOverlapping=True, excl='sexual_chomosomes', allowBeyondChromEnd=False). In certain samples, with an extremely high segment load ($O_i > 10,000$) or with mean ploidy of -1 (that is, monoploid genome), the noOverlapping flag was set to False because the randomization would not converge.

(4) We then binned the autosomes into 28,824 bins of 100 kb and counted for each bin $k_j \{k_1, ... k_{28,842}\}$ the total number of observed events $O_{Tj}$ as the sum of observed events $O_{1k}, ... O_{TK}$ overlapping with that bin across all *S* samples.

(5) Similarly, for each $R_{ith}$ ($R_1, ... R_{10}$) randomization and each bin $k_j \{k_1, ... k_{28,842}\}$, we counted the total number of simulated events as the sum of events—in that *i*th randomization and overlapping with that bin across all samples in *S*.

(6) We then performed a bin-specific comparison of the $O_{Tk}$ with the average number of simulated events $R_{TK}$ across the ten simulations and performed a statistical test of significance using a G-test goodness of fit. As chromosome starting bins were highly depleted in the simulated group ($R_{TK}$), we also computed the global simulated mean across all bins $k_j \{k_1, ... k_{28,842}\}$, and used this as the expected number of events for the statistical significance assessment.

(7) The *P* values were adjusted (that is, converted to *q* values) with a multiple-testing correction using the Benjamini–Hochberg procedure ($\alpha = 0.05$).

(8) For each gene, overlapping with one or with multiple $k_j$ bins, we used the minimal adjusted *P*-value significance of the bin(s) overlapping with the genomic location of the specific gene-coding sequence. Therefore, by definition, two genes sharing the same bins would have a similar *q* value. We used ENSEMBL v.88 to perform the annotation of gene exonic regions to hg19 genomic coordinates.

We observed that LINE insertions near the HLA-I locus (LINE activation site at chr6:29,920,000) in some esophageal cancer samples had an incorrect CN estimation due to multiple insertions originating from almost the same site in the same sample. Consequently, these samples were not considered in the *HLA-I* homozygous deletion analysis.

## Distribution of mutations in *HLA-I* genes

LILAC mapped the *HLA-A*, *HLA-B* and *HLA-C* somatic mutations detected by SAGE into the inferred *HLA-I* alleles (see LILAC section). LILAC

provides the consequence type and coding sequence position of *HLA-I* alterations, which was used to display the distribution of mutations across the *HLA-I* coding sequence. The 34 amino acids involved in peptide presentation were gathered from our neoepitope prioritization pipeline (see below). Pfam *HLA-A*, *HLA-B* and *HLA-C* domains were manually downloaded from the Pfam[36] website.

## Tumor-specific neoepitopes

The methodology for the identification and prioritization of neoepitopes is extensively described in Supplementary Note 2.

## Calculation and randomization of neoepitope ratio

We wanted to evaluate whether LOH of *HLA-I* tends to select the *HLA-I* allele with highest neoepitope repertoire. Let us first introduce the neoepitope allele ratio (nr). Given an *HLA-I* gene, *G*, we defined nr as $G_{A1/2} = G_{A1}/(G_{A1} + G_{A2})$, where $G_{A1/2}$ is the number of predicted neoepitopes of allele 1 and allele 2, respectively. For each patient tumor sample, the assignment of allele number (that is, allele 1 or allele 2) was randomly performed. Then, we followed the next steps:

(1) For each patient sample with LOH of *HLA-I* we calculated the nr across the *HLA-I* genes targeted by the LOH. Homozygous *HLA-I* cases were not considered, because their nr is by definition 0.5.

(2) We then grouped the nr into eight buckets: (0.0–0.35), (0.35, 0.4), (0.4, 0.45), (0.45, 0.5), (0.5, 0.55), (0.55, 0.6), (0.6, 0.65) and (0.65, 1.0). Consequently, each bucket included *n* allele pairs with an nr within the bound limits.

(3) Next, we performed 100 bootstraps by randomly subsampling 75% of the total number of available allele pairs in the bucket.

(4) For each bootstrap iteration *i*th ($i \in 1, \dots 100$) and each bucket we estimated the frequency of allele 1 loss ($F_{A1\_loss}$) as the number of cases with allele 1 loss compared with the total number of cases in that bucket. Similarly, we computed the expected frequency ($F_{A1exp}$) by randomly assigning LOH events to the allele 1 (background probably of 0.5).

(5) We then computed the bucket-specific average and s.d. of $F_{A1loss}$ and $F_{A1exp}$ values across the 100 bootstraps.

(6) Finally, we performed a Kolmogorov–Smirnov test to compare the observed distribution with the expected given random distribution of the LOH events.

This test was applied to LOH of *HLA-I*, focal LOH of *HLA-I* and nonfocal LOH of *HLA-I* events across the metastatic (Hartwig) and primary (PCAWG) datasets.

## GIE and tumor genomic features

Check Supplementary Note 3 for a full description of the methods for this section.

## GIE and TMB association

We aggregated the two datasets, metastatic and primary, to increase the robustness of this analysis. We then defined 20 evenly arranged buckets (10 for the cancer type-specific analyses) of the $\log_{10}$(TMB) scale, starting from the 1st percentile and ending in the 99th percentile values. Next, each sample with a $\log_{10}$(TMB) = $S_{tmb}$, was allocated to the *i*th ($i \in 1, \dots 20$) bucket such as $\log_{10}(TMB)_{i-1} < S_{tmb} \le \log_{10}(TMB)_i$. Samples with an $S_{tmb}$ greater than the last bucket threshold (that is, $\log_{10}(TMB)_{20}$) were allocated into the last bucket. The number of mutations in each bucket was displayed as the number of mutations per megabase by dividing the total number of mutations by 3,000 (that is, approximated number of human genome megabases). Finally, the GIE frequency ($GIE_{freq}$) of the *i*th bucket was defined as the number of GIE samples in the *i*th bucket divided by the total number of available samples in that bucket.

To enable calculation of the uniformity in GIE frequency among samples in the same TMB bucket, we performed *n* (where *n* = 1,000) bootstraps of the 50% of samples allocated to each bucket. We then calculated the average and s.d. of the $GIE_{freq}$ across the bootstraps.

A similar approach was conducted to analyze the relationship between the predicted neoepitope load and GIE frequency. The number of neoantigens of each bucket was estimated as 1% (ref. 46) and 5% (ref. 47) of the total predicted neoepitopes assigned to that bucket threshold.

For the simulated GIE control, we estimated the average and s.d. across the 100 simulated GIE iterations for each TMB bucket.

## Statistics and reproducibility

Sample sizes were determined by the availability of samples with sufficient quality from the two datasets included in the present study (PCAWG and Hartwig). Sample-exclusion criteria are thoroughly described in Methods, Supplementary Note 1 and the original publication describing the harmonized cohort[20].

The statistical tests and randomization strategies used in each specific analysis are described in Methods and the figure legends. Unless otherwise specified, the scipy[48] (v.1.5.3) library from python v.3.6.9 was used to carry out the statistical tests.

All the code and data to reproduce the analyses presented in the present article have been deposited in public repositories as described in Data availability and Code availability.

## Reporting summary

Further information on research design is available in the Nature Portfolio Reporting Summary linked to this article.

## Data availability

The Hartwig dataset used in the present study is freely available for academic use from the Hartwig Medical Foundation through standardized procedures and request forms that can be found at https://www.hartwigmedicalfoundation.nl/en/applying-for-data. This includes raw sequencing data (.bam files and unmapped reads for hg19 reference genome) as well as the processed data through the latest version of the Hartwig tumor-processing pipeline. The re-processed PCAWG data using the Hartwig Medical Foundation pipeline (for hg19 reference genome) have also been made available for academic purposes. The ICGC part of the PCAWG dataset[17] can be accessed now through the ICGC platform (https://dcc.icgc.org/releases/PCAWG/Hartwig), following their standard access control mechanisms originally put in place. Similarly, users with authorized access can download the re-processed TCGA portion of the PCAWG dataset at https://icgc.bionimbus.org/files/5310a3ac-0344-458a-88ce-d55445540120. We refer to the accompanying publication[20], including the description of the entire primary and metastatic cohort, for further information about the technical aspects of the re-processing of the PCAWG dataset. Raw sequencing data of the high-resolution HLA typing performed by GenDx can also be downloaded via European Genome-phenome Archive (http://www.ebi.ac.uk/ega) under accession no. EGAD00001008643. *HLA-I* typing, sample-specific GIE events and processed data are now shared as Supplementary Data and Supplementary Tables.

## Code availability

The Hartwig analytical processing pipeline is available at https://github.com/hartwigmedical/pipeline5 and implemented in Platinum (https://github.com/hartwigmedical/platinum). LILAC's source code is available at https://github.com/hartwigmedical/hmftools/tree/master/lilac. The source code of the neoepitope prioritization pipeline is available at https://github.com/hartwigmedical/hmftools/tree/master/neo. The source code to reproduce the figures and analysis of the manuscript is available at https://github.com/UMCUGenetics/Genetic-Immune-Escape.

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

## Acknowledgements

This publication and the underlying study have been made possible partly on the basis of the data that the Hartwig Medical Foundation and the Center of Personalised Cancer Treatment have made available to the study. We thank A. Van Hoeck and A. G. Perez for critical reading of the manuscript and their valuable input. We also thank A. Van Hoeck for assistance in the dataset collection and L. Nyguen for help in the mutational signature extraction. We thank H. W. Lee for technical assistance in the TRACERx dataset processing. The present study makes use of data generated by the TRAcking Non-small Cell Lung Cancer Evolution Through Therapy (TRACERx) Consortium and provided by the UCL Cancer Institute and the Francis Crick Institute. The TRACERx study is sponsored by University College London (UCL), funded by Cancer Research UK and coordinated through the Cancer Research UK and UCL Cancer Trials Centre. We also thank N. McGranahan, O. Pich and C. Puttick for sharing the original *HLA-I* typing and LOHHLA calls used in their original publication. Finally, we thank W. Mulder, M. Penning and H. Merkens for discussions on LILAC's orthogonal validation.

## Author contributions

F.M.J., E.C. and P.P. conceived the study. F.M.J., P.P., J.B. and C.S. provided the methodology. P.P., J.B. and C.S. provided the software. F.M.J., P.P. and E.R. validated the study. F.M.J. and P.P. performed the formal analysis. F.M.J. and P.P. carried out the investigations. E.C. and P.P. provided the resources. F.M.J., P.P., C.S. and E.R. curated the data. F.M.J. wrote the original draft of the MS. E.C. and P.P. reviewed and edited the MS. F.M.J. visualized the study. F.M.J. and E.C. supervised the study. F.M.J. and E.C. administered the project. E.C. acquired funding.

## Competing interests

The authors declare no competing interests.

## Additional information

**Extended data** is available for this paper at https://doi.org/10.1038/s41588-023-01367-1.

**Correspondence and requests for materials** should be addressed to Francisco Martínez-Jiménez or Edwin Cuppen.

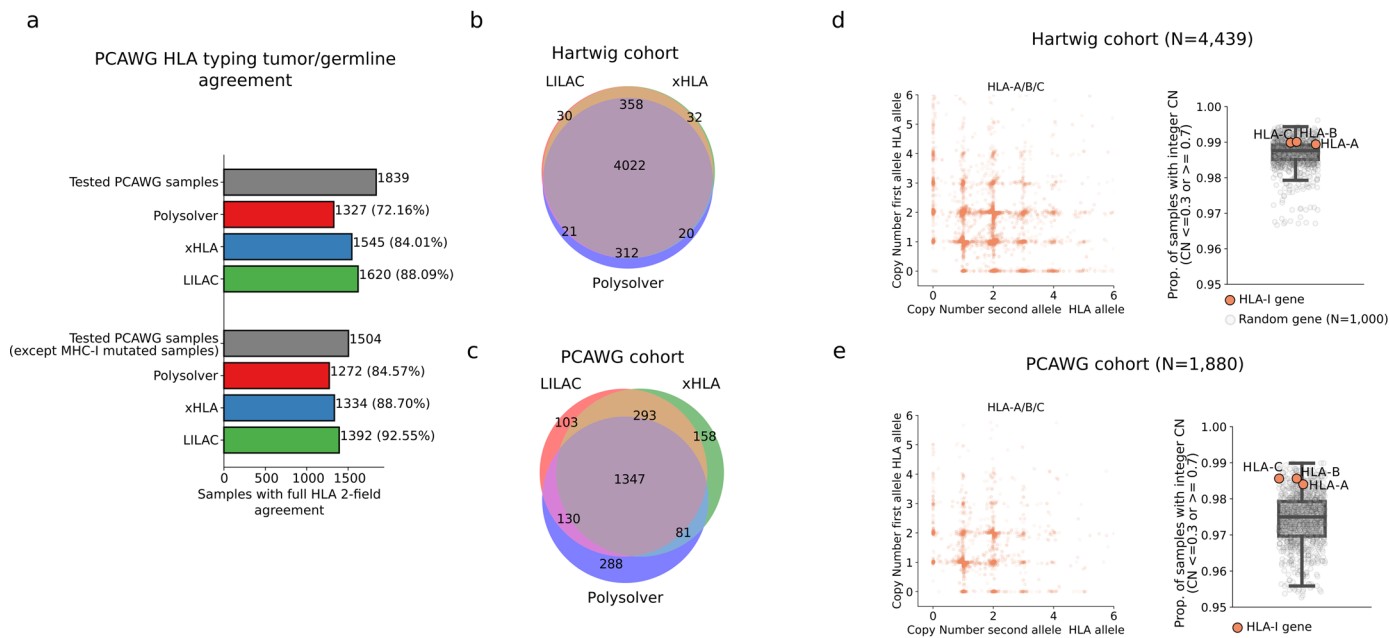

**Extended Data Fig. 1 | LILAC's validation. a)** *HLA-I* typing tumor and germline agreement in PCAWG. Venn diagrams representing the overlap of germline samples with perfect 2-field *HLA-I* allotype match in **b)** Hartwig and **c)** PCAWG. **d)** Left, copy number of minor and major alleles of *HLA-A*, *HLA-B* and *HLA-C* in Hartwig dataset. Right, proportion of samples with integer minor and major copy number (copy number <=0.3 or >=0.7) of *HLA-I* genes (orange) compared to the average of 1,000 randomly selected genes (gray) in the Hartwig cohort (N=4,439). **e)** analogous for PCAWG samples (N=1,880). Box-plots: center line, median; box limits, first and third quartiles; whiskers, lowest/highest data points at first quartile minus/plus 1.5× IQR. N, number of samples.

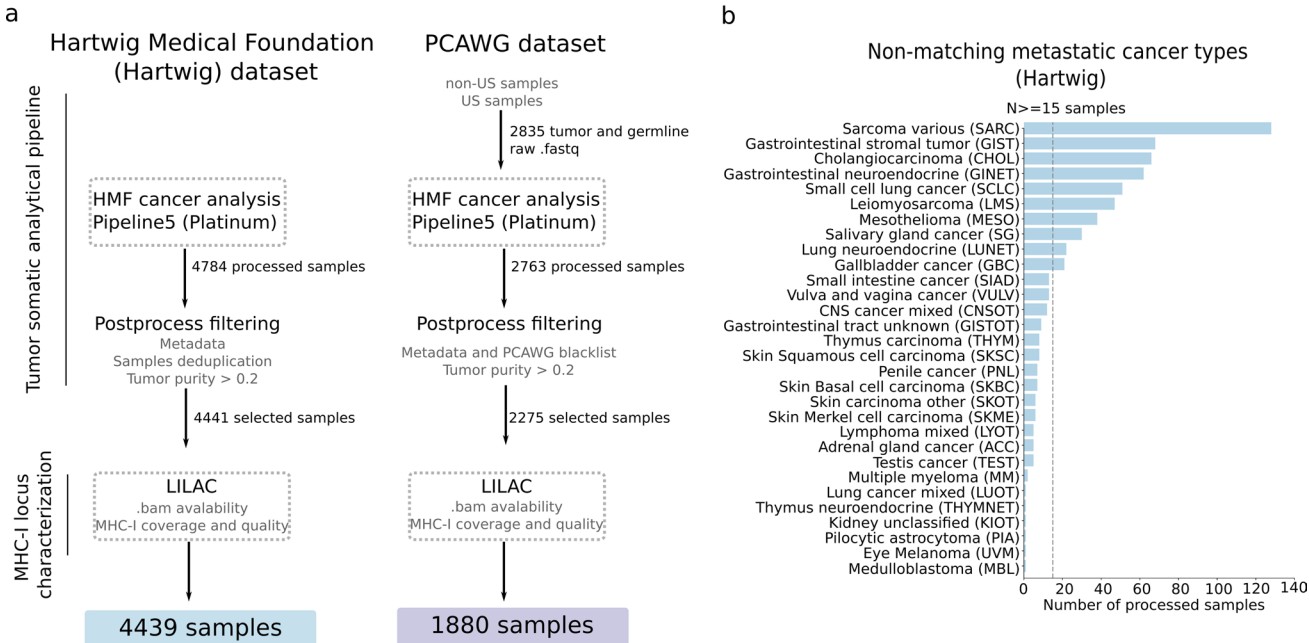

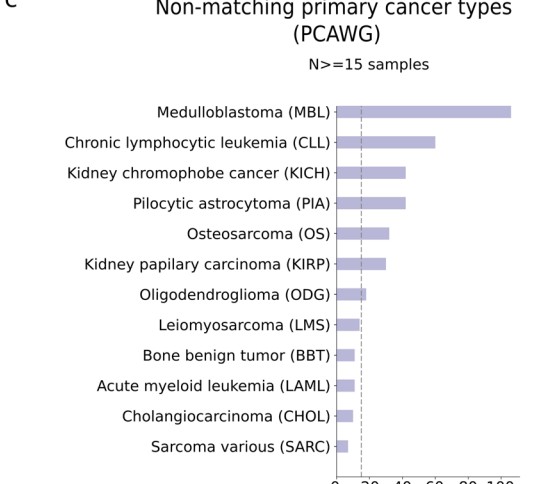

**Extended Data Fig. 2 | GIE in primary and metastatic tumors. a)** Workflow of the sample processing pipeline used in this study in Hartwig (left) and PCAWG (right). Each rectangle represents a processing step. The resulting number of selected samples for this study are displayed at the bottom. Sample-exclusion criteria for LILAC is described in Supp. Note 1. **b)** number of metastatic (Hartwig) samples across cancer types that lack sufficient representation in the primary (PCAWG) dataset. **c)** Analogous representation for primary (PCAWG) samples. Vertical dashed lines (N>=15 samples) represent the threshold of samples to consider a cohort as sufficiently populated.

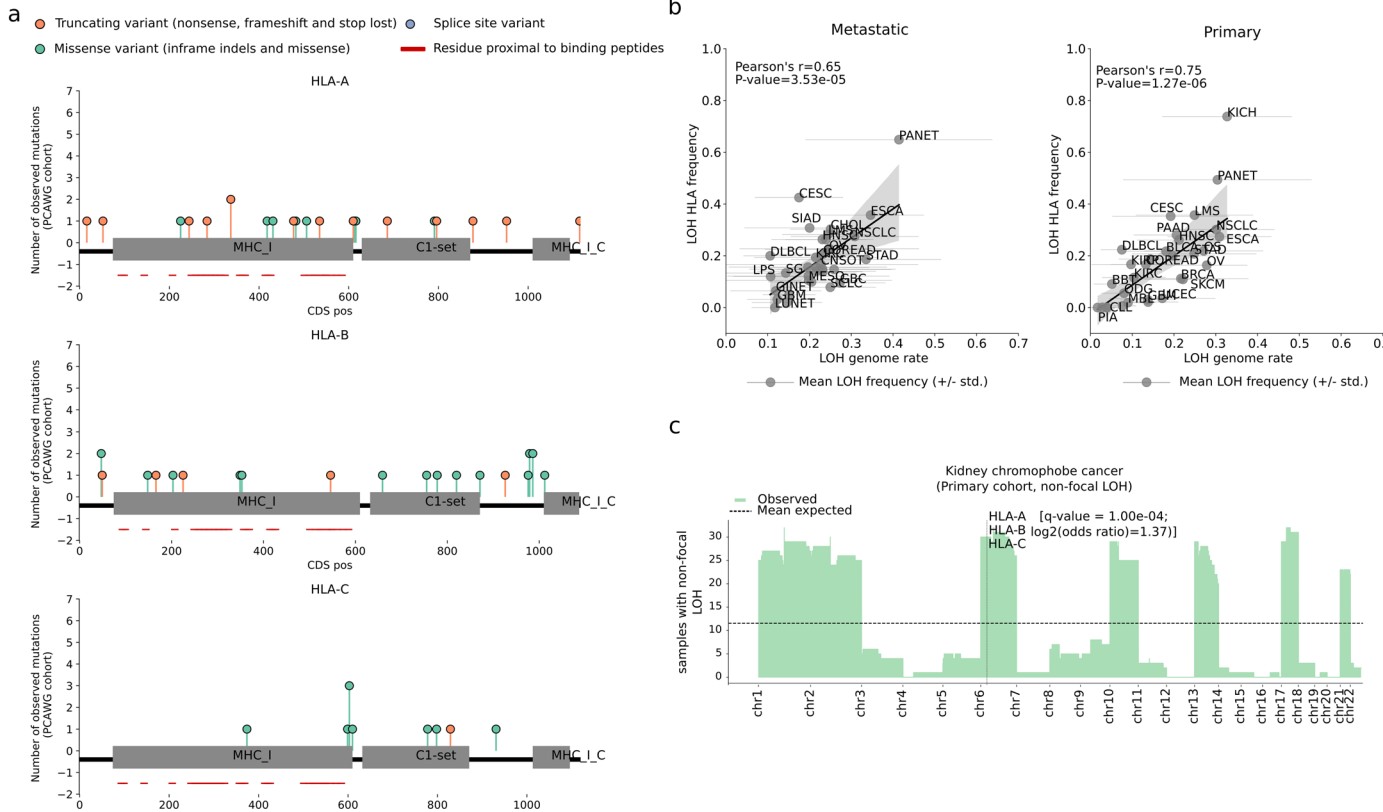

**Extended Data Fig. 3 | Positive selection of *HLA-I* genes. a)** Needle plots representing the pan-cancer distribution of somatic mutations along the *HLA-A*, *HLA-B* and *HLA-C* protein sequences in the primary (PCAWG) dataset. Mutations are colored according to the consequence type. Rectangles represent the Pfam domains. **b)** Correlation of LOH of *HLA-I* rates (y-axis) with the mean genome-wide LOH rates (x-axis) across the metastatic (left) and primary (right) datasets. Each dot represents a cancer type. Horizontal lines represent genome-wide LOH standard deviation across samples from each cancer type. The regression and 95% confidence intervals of the linear regression are represented as a solid line and the adjacent shaded area, respectively. Confidence intervals are calculated using 1,000 bootstraps. P-values are obtained from a two-sided test of no-correlation. **c)** Distribution of non-focal LOH events along the autosomes in the primary kidney chromophobe cancer cohort. X-ticks represent the chromosomal starting position. Dashed horizontal lines represent the expected mean after randomization. Vertical dashed lines highlight the *HLA-I* genomic location. CDS pos, coding sequence position. Std., standard deviation.

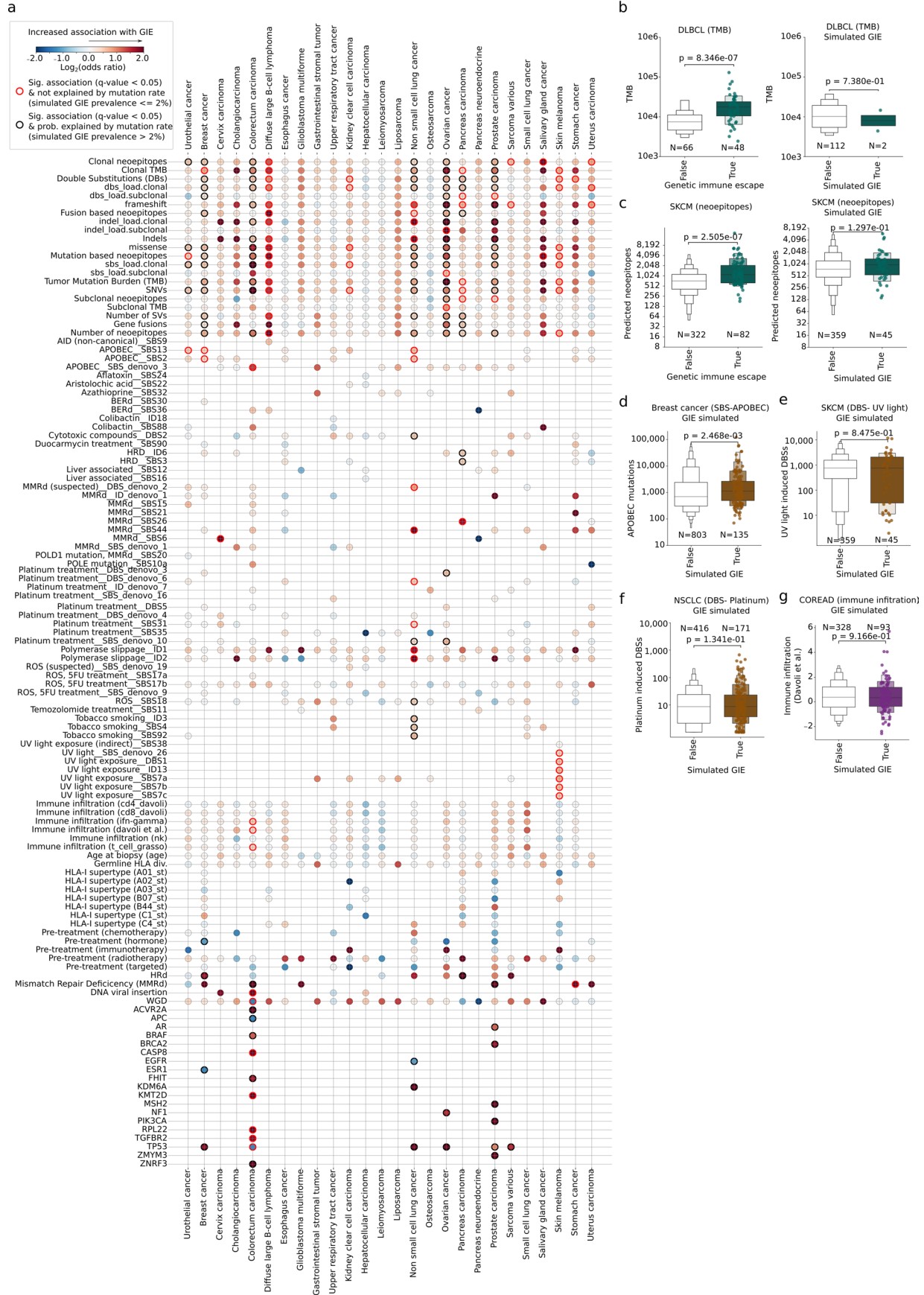

**Extended Data Fig. 4 | See next page for caption.**

**Extended Data Fig. 4 | GIE association with cancer genomic features.**
**a)** Heatmap displaying the association of genomic features with GIE frequency across cancer types. Significant associations that can not be explained by higher background mutation rate are highlighted by a red border line. Significant associations found in >2% of the GIE simulations are highlighted by a black border line. **b)** Comparison of the TMB between samples bearing GIE alterations and non-GIE samples in diffuse large B-cell lymphoma (DLBCL). Left boxplot, using observed GIE events. Right using simulated GIE events. **c)** Similar comparison for neoepitope burden in skin melanoma. **d)** Comparison of the APOBEC mutational exposure between samples bearing simulated GIE alterations and wild-type (no simulated GIE) in breast cancer in one randomly selected simulation.

**e)** Analogous for ultraviolet light (UV) associated double base substitutions (DBSs) in skin melanoma and **f)** for platinum treatment associated DBSs in non-small cell lung cancer (NSCLC). **g)** Comparison of immune infiltration estimates from Davoli et al.[37] between samples bearing simulated GIE alterations and wild-type (no simulated GIE) in colorectal cancer. Boxplots: center line, median; First section out from the centerline contains 50% of the data. The next sections contain half the remaining data until we are at the outlier level. Each level out is shaded lighter. N, number of samples. P-values of the boxplots are calculated using a two-sided Mann–Whitney U test. One of the 100 simulations was randomly selected for all the simulated GIE boxplots. SBS, single base substitution.

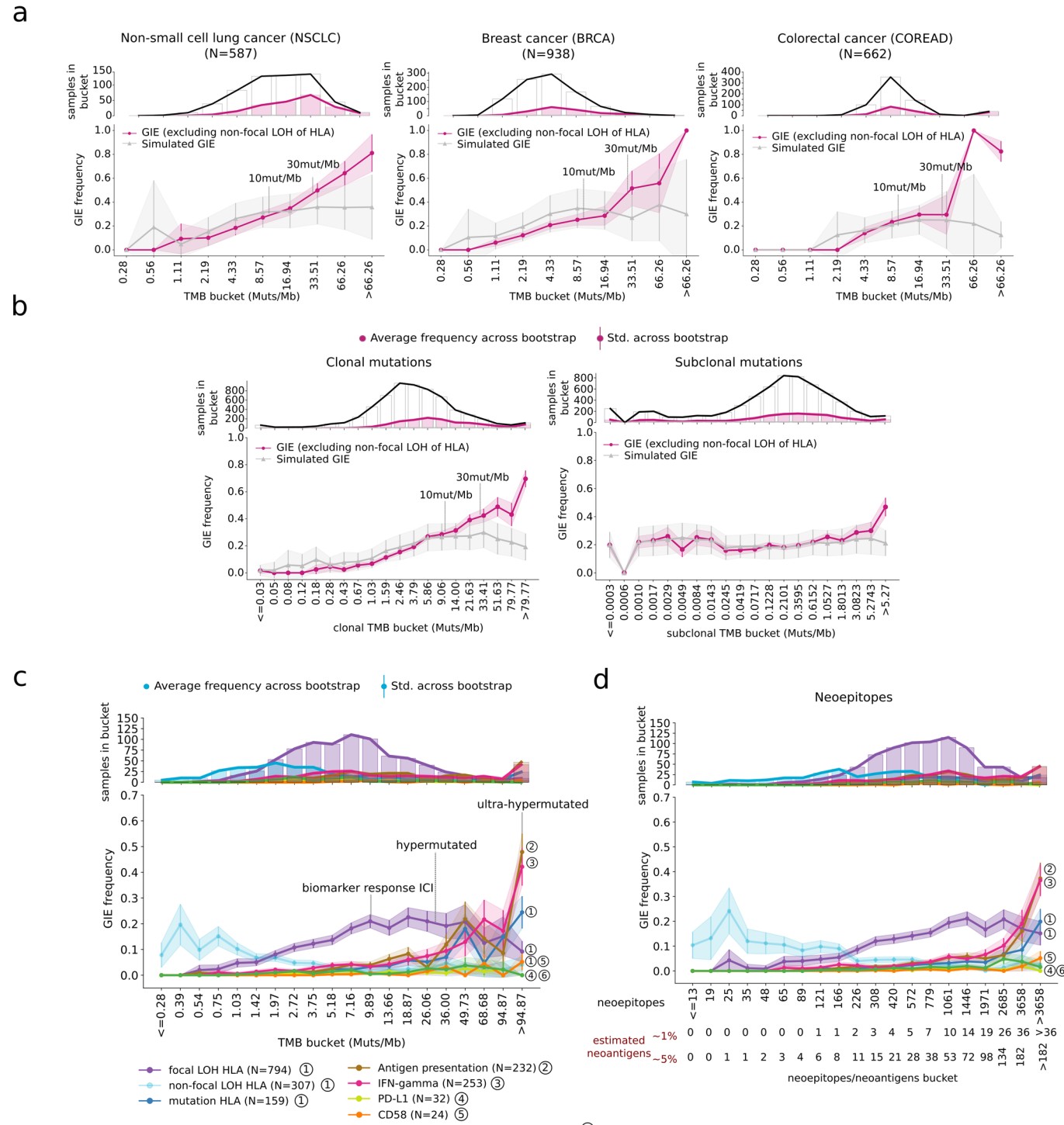

**Extended Data Fig. 5 | Immune evasion mechanisms and TMB. a)** Analogous to Fig. 7a, but restricted to (from left to right) non-small cell lung cancer, breast cancer and colorectal cancer patients. Top panels represent the total number of bucket-assigned (white bars with a black contouring line) and GIE-positive (pink bars with pink contouring lines) samples across the ten evenly arranged TMB buckets. bottom panel, representation of observed (pink) and simulated (gray) GIE frequency across these buckets. For the observed GIE values, average (represented as pink dots) and standard deviation (std, vertical error bars and shaded pink area) values are computed using 1,000 bootstraps from the total number of cancer type samples classified into each bucket (from the top panel). For the simulated GIE values, average (gray triangle) and standard deviation (std, vertical bars and shaded gray area) values are computed from 100 GIE

simulations using the total number of cancer type samples assigned into each bucket. **b)** Analogous to panel a) but using the entire cohort (N=6,319) and twenty evenly arranged buckets as well as clonal TMB (left) and subclonal TMB (right) as baseline. **c)** Equivalent to panel a) but using the entire cohort (N=6,319) and twenty evenly arranged buckets and grouping GIE events according to the assigned GIE pathway. Label number represents the assigned immune escape pathway (from Fig. 1a) **d)** Similar to panel c) but using the number of predicted neoepitopes as baseline for the buckets. Bottom labels, number of estimated neoantigens as a relative percentage (1% and 5%) of the number of predicted neoepitopes in the bucket. N, number of samples. Muts/Mb, mutations per megabase. ICI, immune checkpoint inhibitor.

# Reporting Summary

## Statistics

For all statistical analyses, confirm that the following items are present in the figure legend, table legend, main text, or Methods section.

| n/a | Confirmed | |
|---|---|---|
| ☐ | ☒ | The exact sample size (*n*) for each experimental group/condition, given as a discrete number and unit of measurement |
| ☐ | ☒ | A statement on whether measurements were taken from distinct samples or whether the same sample was measured repeatedly |
| ☐ | ☒ | The statistical test(s) used AND whether they are one- or two-sided *Only common tests should be described solely by name; describe more complex techniques in the Methods section.* |
| ☐ | ☒ | A description of all covariates tested |
| ☐ | ☒ | A description of any assumptions or corrections, such as tests of normality and adjustment for multiple comparisons |
| ☐ | ☒ | A full description of the statistical parameters including central tendency (e.g. means) or other basic estimates (e.g. regression coefficient) AND variation (e.g. standard deviation) or associated estimates of uncertainty (e.g. confidence intervals) |
| ☐ | ☒ | For null hypothesis testing, the test statistic (e.g. *F*, *t*, *r*) with confidence intervals, effect sizes, degrees of freedom and *P* value noted *Give P values as exact values whenever suitable.* |
| ☒ | ☐ | For Bayesian analysis, information on the choice of priors and Markov chain Monte Carlo settings |
| ☐ | ☒ | For hierarchical and complex designs, identification of the appropriate level for tests and full reporting of outcomes |
| ☐ | ☒ | Estimates of effect sizes (e.g. Cohen's *d*, Pearson's *r*), indicating how they were calculated |

*Our web collection on statistics for biologists contains articles on many of the points above.*

## Software and code

Policy information about availability of computer code

| | |
|---|---|
| Data collection | No software was used for data collection |
| Data analysis | Somatic mutation data of the CPCT, DRUP and WIDE projects were kindly shared by Hartwig on 6 February 2020 withan update received on 4 Februari 2022. |

The PCAWG samples were reanalyzed with the Hartwig somatic variant calling pipeline (https://github.com/hartwigmedical/pipeline5) which was hosted on the Google Cloud Platform using Platinum (v1.0) (https://github.com/hartwigmedical/platinum). This pipeline uses the following software packages:
BWA (v0.7.17): read mapping
GATK (v3.8.0) Haplotype Caller: calling germline variants in the reference sample
SAGE (v2.2): somatic SMNVs and indels calling
GRIDSS (v2.9.3): simple and complex structural variant calling
PURPLE (v2.53): combines B-allele frequency (BAF) from AMBER (v3.3), read depth ratios from COBALT (v1.7), and structural variants from GRIDSS to estimate copy number profiles, variant allele frequency (VAF) and variant clonality. PURPLE also determines sample gender based on sex chromosome ploidy.
LINX (v1.17): interpretation of simple mutations and structural variants
CHORD (v1.0): detection of Homologus Repair deficiency.

Unless otherwise specified the scipy (v.1.5.3) library from python v3.6.9 was used to carry out the statistical tests.
The Hartwig analytical processing pipeline is available at (https://github.com/hartwigmedical/pipeline5) and implemented in Platinum (https://github.com/hartwigmedical/platinum). LILAC's source code is available at (https://github.com/hartwigmedical/hmftools/tree/master/

lilac). The source code of the neoepitope prioritization pipeline is available at https://github.com/hartwigmedical/hmftools/tree/master/neo. The source code to reproduce the figures and analysis of the manuscript is available at https://github.com/UMCUGenetics/Genetic-Immune-Escape.

For manuscripts utilizing custom algorithms or software that are central to the research but not yet described in published literature, software must be made available to editors and reviewers. We strongly encourage code deposition in a community repository (e.g. GitHub). See the Nature Portfolio guidelines for submitting code & software for further information.

# Data

Policy information about availability of data

All manuscripts must include a data availability statement. This statement should provide the following information, where applicable:
- Accession codes, unique identifiers, or web links for publicly available datasets
- A description of any restrictions on data availability
- For clinical datasets or third party data, please ensure that the statement adheres to our policy

The Hartwig dataset used in this study are freely available for academic use from the Hartwig Medical Foundation through standardized procedures and request forms that can be found at https://www.hartwigmedicalfoundation.nl/en/applying-for-data/. This includes raw sequencing data (.bam files and unmapped reads, in hg19 reference genome) as well as the processed data through the latest version of the Hartwig tumor processing pipeline.

The re-processed PCAWG data using Hartwig Medical Foundation pipeline (for hg19 reference genome) have also been made available for academic purposes. The ICGC part of the PCAWG dataset can be accessed now through the ICGC platform (https://dcc.icgc.org/releases/PCAWG/Hartwig), following their standard access control mechanisms originally put in place. Similarly, users with authorized access can download the TCGA portion of the PCAWG dataset at https://icgc.bionimbus.org/files/5310a3ac-0344-458a-88ce-d55445540120.

Raw sequencing data of the high-resolution HLA typing performed by GenDx can also be downloaded via European Genome-phenome Archive (http://www.ebi.ac.uk/ega/) under accession number EGAD00001008643.

HLA-I typing, sample-specific GIE events and processed data is now shared in the supplementary data and suplementary tables.

# Human research participants

Policy information about studies involving human research participants and Sex and Gender in Research.

| Reporting on sex and gender | Consistent gender proportions were observed across all cancer types except for thyroid adenocarcinomas, which had higher male representation in the metastatic cohort (metastatic: 72% male, 28% female; primary: 25% male, 75% female). |
| Population characteristics | The Hartwig cohort includes late-stage adult (>18 years old) cancer patients recruited across Dutch hospitals. Patients had frequently recived pre-biopsy treatment. The PCAWG cohort primarily include adult and early-stage cancer patients that in most cases have not recived any treatment prior to tumor biopsy. We refer to the Hartwig (doi: 10.1038/s41586-019-1689-y) and PCAWG (doi: 10.1038/s41586-020-1969-6.) flagship papers for further description of patient's population, recruitment and ethics oversight. |
| Recruitment | Patient recruitment was originally performed by the clinical institutions and hospitals. This study did not play any role in patient recruitment. |
| Ethics oversight | N/A |

Note that full information on the approval of the study protocol must also be provided in the manuscript.

# Field-specific reporting

Please select the one below that is the best fit for your research. If you are not sure, read the appropriate sections before making your selection.

☒ Life sciences     ☐ Behavioural & social sciences     ☐ Ecological, evolutionary & environmental sciences

For a reference copy of the document with all sections, see nature.com/documents/nr-reporting-summary-flat.pdf

# Life sciences study design

All studies must disclose on these points even when the disclosure is negative.

| Sample size | We requested the data for all possible samples from the Hartwig and PCAWG cohorts.<br>The Hartwig cohort included 4782 metastatic tumor samples from 4572 patients.<br>The PCAWG cohort consisted of 2835 tumor samples from unique patients.<br>After substantial filtering based on quality control (see data exclusions) we used a total of 6,319 tumor samples, including 1,880 primary patients from PCAWG and 4,439 metastatic patients from Hartwig cohort. |

| Data exclusions | A selection of samples for all analyses was made based on several criteria. To exclude duplicate samples from the same patient for the Hartwig cohort, we selected the tumor sample with the most recent biopsy date, and if this information did not exist we selected the sample with the highest tumor purity. However, some patients had biopsies from different primary tumor locations (likely independent or secondary tumors). In these cases, we kept at least one sample from each primary tumor location, and when there were multiple samples from the same primary tumor location, we applied the aforementioned biopsy date and tumor purity filtering criteria. For the PCAWG cohort, we processed one tumor sample per donor and tumor sample IDs are included in Supp. Table 2 of the manuscript. As with Hartwig QC filter criteria, samples with a tumor purity lower than 20% were removed as somatic variant calling was less reliable for these samples. PCAWG samples that were gray- or blacklisted by the PCAWG consortium were also removed (see https://dcc.icgc.org/releases/PCAWG/donors_and_biospecimens). For both cohorts, we only kept samples with >=50 SNVs/indels (likely no tumor cells present in the sample), and removed an additional set of samples for several reasons including failed variant calling, insufficient informed consent for use of the WGS data, unnatural SV landscape, and one duplicate PCAWG patient (DO217844) that was also included in the Hartwig cohort. Finally, samples with insufficient coverage and quality of the HLA-I locus according to LILAC were also discarded. After strict QC filtering, the PCAWG whitelisted cohort includes 1,880 samples and this dataset will be made available for the cancer research community via the PCAWG resource page. The metadata for every sample including those selected for analyses is detailed in supplementary table 2.<br>See also the accompanying publication https://www.biorxiv.org/content/10.1101/2022.06.17.496528v1 for more details about the dataset. |
|---|---|
| Replication | The source data and the source code used in this study are publicly available for academic purposes to ensure the reproducibility of the analysis conducted in this stud |
| Randomization | Patients from both datasets (Hartwig and PCAWG) were independently recruited by clinical institutions and hospitals. Patients from the Hartwig Medical Foundation cohort represent late-stage cancer patients while PCAWG patients are primarily early-stage untreated cancer patients. This study did not play any role in patient's recruitment and randomization into experimental groups. |
| Blinding | This study did not play any role in patient's recruitment . |

# Behavioural & social sciences study design

All studies must disclose on these points even when the disclosure is negative.

| Study description | Briefly describe the study type including whether data are quantitative, qualitative, or mixed-methods (e.g. qualitative cross-sectional, quantitative experimental, mixed-methods case study). |
|---|---|
| Research sample | State the research sample (e.g. Harvard university undergraduates, villagers in rural India) and provide relevant demographic information (e.g. age, sex) and indicate whether the sample is representative. Provide a rationale for the study sample chosen. For studies involving existing datasets, please describe the dataset and source. |
| Sampling strategy | Describe the sampling procedure (e.g. random, snowball, stratified, convenience). Describe the statistical methods that were used to predetermine sample size OR if no sample-size calculation was performed, describe how sample sizes were chosen and provide a rationale for why these sample sizes are sufficient. For qualitative data, please indicate whether data saturation was considered, and what criteria were used to decide that no further sampling was needed. |
| Data collection | Provide details about the data collection procedure, including the instruments or devices used to record the data (e.g. pen and paper, computer, eye tracker, video or audio equipment) whether anyone was present besides the participant(s) and the researcher, and whether the researcher was blind to experimental condition and/or the study hypothesis during data collection. |
| Timing | Indicate the start and stop dates of data collection. If there is a gap between collection periods, state the dates for each sample cohort. |
| Data exclusions | If no data were excluded from the analyses, state so OR if data were excluded, provide the exact number of exclusions and the rationale behind them, indicating whether exclusion criteria were pre-established. |
| Non-participation | State how many participants dropped out/declined participation and the reason(s) given OR provide response rate OR state that no participants dropped out/declined participation. |
| Randomization | If participants were not allocated into experimental groups, state so OR describe how participants were allocated to groups, and if allocation was not random, describe how covariates were controlled. |

# Ecological, evolutionary & environmental sciences study design

All studies must disclose on these points even when the disclosure is negative.

| Study description | Briefly describe the study. For quantitative data include treatment factors and interactions, design structure (e.g. factorial, nested, hierarchical), nature and number of experimental units and replicates. |
|---|---|
| Research sample | Describe the research sample (e.g. a group of tagged Passer domesticus, all Stenocereus thurberi within Organ Pipe Cactus National Monument), and provide a rationale for the sample choice. When relevant, describe the organism taxa, source, sex, age range and any manipulations. State what population the sample is meant to represent when applicable. For studies involving existing datasets, describe the data and its source. |

| Sampling strategy | *Note the sampling procedure. Describe the statistical methods that were used to predetermine sample size OR if no sample-size calculation was performed, describe how sample sizes were chosen and provide a rationale for why these sample sizes are sufficient.* |
|---|---|
| Data collection | *Describe the data collection procedure, including who recorded the data and how.* |
| Timing and spatial scale | *Indicate the start and stop dates of data collection, noting the frequency and periodicity of sampling and providing a rationale for these choices. If there is a gap between collection periods, state the dates for each sample cohort. Specify the spatial scale from which the data are taken* |
| Data exclusions | *If no data were excluded from the analyses, state so OR if data were excluded, describe the exclusions and the rationale behind them, indicating whether exclusion criteria were pre-established.* |
| Reproducibility | *Describe the measures taken to verify the reproducibility of experimental findings. For each experiment, note whether any attempts to repeat the experiment failed OR state that all attempts to repeat the experiment were successful.* |
| Randomization | *Describe how samples/organisms/participants were allocated into groups. If allocation was not random, describe how covariates were controlled. If this is not relevant to your study, explain why.* |
| Blinding | *Describe the extent of blinding used during data acquisition and analysis. If blinding was not possible, describe why OR explain why blinding was not relevant to your study.* |

Did the study involve field work? ☐ Yes ☐ No

## Field work, collection and transport

| Field conditions | *Describe the study conditions for field work, providing relevant parameters (e.g. temperature, rainfall).* |
|---|---|
| Location | *State the location of the sampling or experiment, providing relevant parameters (e.g. latitude and longitude, elevation, water depth).* |
| Access & import/export | *Describe the efforts you have made to access habitats and to collect and import/export your samples in a responsible manner and in compliance with local, national and international laws, noting any permits that were obtained (give the name of the issuing authority, the date of issue, and any identifying information).* |
| Disturbance | *Describe any disturbance caused by the study and how it was minimized.* |

## Reporting for specific materials, systems and methods

We require information from authors about some types of materials, experimental systems and methods used in many studies. Here, indicate whether each material, system or method listed is relevant to your study. If you are not sure if a list item applies to your research, read the appropriate section before selecting a response.

### Materials & experimental systems

| n/a | Involved in the study |
|---|---|
| ☒ | ☐ Antibodies |
| ☒ | ☐ Eukaryotic cell lines |
| ☒ | ☐ Palaeontology and archaeology |
| ☒ | ☐ Animals and other organisms |
| ☒ | ☐ Clinical data |
| ☒ | ☐ Dual use research of concern |

### Methods

| n/a | Involved in the study |
|---|---|
| ☒ | ☐ ChIP-seq |
| ☒ | ☐ Flow cytometry |
| ☒ | ☐ MRI-based neuroimaging |

## Antibodies

| Antibodies used | *Describe all antibodies used in the study; as applicable, provide supplier name, catalog number, clone name, and lot number.* |
|---|---|
| Validation | *Describe the validation of each primary antibody for the species and application, noting any validation statements on the manufacturer's website, relevant citations, antibody profiles in online databases, or data provided in the manuscript.* |

# Eukaryotic cell lines

Policy information about cell lines and Sex and Gender in Research

| Cell line source(s) | State the source of each cell line used and the sex of all primary cell lines and cells derived from human participants or vertebrate models. |
| --- | --- |
| Authentication | Describe the authentication procedures for each cell line used OR declare that none of the cell lines used were authenticated. |
| Mycoplasma contamination | Confirm that all cell lines tested negative for mycoplasma contamination OR describe the results of the testing for mycoplasma contamination OR declare that the cell lines were not tested for mycoplasma contamination. |
| Commonly misidentified lines (See ICLAC register) | Name any commonly misidentified cell lines used in the study and provide a rationale for their use. |

# Palaeontology and Archaeology

| Specimen provenance | Provide provenance information for specimens and describe permits that were obtained for the work (including the name of the issuing authority, the date of issue, and any identifying information). Permits should encompass collection and, where applicable, export. |
| --- | --- |
| Specimen deposition | Indicate where the specimens have been deposited to permit free access by other researchers. |
| Dating methods | If new dates are provided, describe how they were obtained (e.g. collection, storage, sample pretreatment and measurement), where they were obtained (i.e. lab name), the calibration program and the protocol for quality assurance OR state that no new dates are provided. |

☐ Tick this box to confirm that the raw and calibrated dates are available in the paper or in Supplementary Information.

| Ethics oversight | Identify the organization(s) that approved or provided guidance on the study protocol, OR state that no ethical approval or guidance was required and explain why not. |
| --- | --- |

Note that full information on the approval of the study protocol must also be provided in the manuscript.

# Animals and other research organisms

Policy information about studies involving animals; ARRIVE guidelines recommended for reporting animal research, and Sex and Gender in Research

| Laboratory animals | For laboratory animals, report species, strain and age OR state that the study did not involve laboratory animals. |
| --- | --- |
| Wild animals | Provide details on animals observed in or captured in the field; report species and age where possible. Describe how animals were caught and transported and what happened to captive animals after the study (if killed, explain why and describe method; if released, say where and when) OR state that the study did not involve wild animals. |
| Reporting on sex | Indicate if findings apply to only one sex; describe whether sex was considered in study design, methods used for assigning sex. Provide data disaggregated for sex where this information has been collected in the source data as appropriate; provide overall numbers in this Reporting Summary. Please state if this information has not been collected. Report sex-based analyses where performed, justify reasons for lack of sex-based analysis. |
| Field-collected samples | For laboratory work with field-collected samples, describe all relevant parameters such as housing, maintenance, temperature, photoperiod and end-of-experiment protocol OR state that the study did not involve samples collected from the field. |
| Ethics oversight | Identify the organization(s) that approved or provided guidance on the study protocol, OR state that no ethical approval or guidance was required and explain why not. |

Note that full information on the approval of the study protocol must also be provided in the manuscript.

# Clinical data

Policy information about clinical studies
All manuscripts should comply with the ICMJE guidelines for publication of clinical research and a completed CONSORT checklist must be included with all submissions.

| Clinical trial registration | Provide the trial registration number from ClinicalTrials.gov or an equivalent agency. |
| --- | --- |
| Study protocol | Note where the full trial protocol can be accessed OR if not available, explain why. |
| Data collection | Describe the settings and locales of data collection, noting the time periods of recruitment and data collection. |

| Outcomes | *Describe how you pre-defined primary and secondary outcome measures and how you assessed these measures.* |

# Dual use research of concern

Policy information about dual use research of concern

## Hazards

Could the accidental, deliberate or reckless misuse of agents or technologies generated in the work, or the application of information presented in the manuscript, pose a threat to:

No | Yes
☐ | ☐ Public health
☐ | ☐ National security
☐ | ☐ Crops and/or livestock
☐ | ☐ Ecosystems
☐ | ☐ Any other significant area

## Experiments of concern

Does the work involve any of these experiments of concern:

No | Yes
☐ | ☐ Demonstrate how to render a vaccine ineffective
☐ | ☐ Confer resistance to therapeutically useful antibiotics or antiviral agents
☐ | ☐ Enhance the virulence of a pathogen or render a nonpathogen virulent
☐ | ☐ Increase transmissibility of a pathogen
☐ | ☐ Alter the host range of a pathogen
☐ | ☐ Enable evasion of diagnostic/detection modalities
☐ | ☐ Enable the weaponization of a biological agent or toxin
☐ | ☐ Any other potentially harmful combination of experiments and agents

# ChIP-seq

## Data deposition

☐ Confirm that both raw and final processed data have been deposited in a public database such as GEO.

☐ Confirm that you have deposited or provided access to graph files (e.g. BED files) for the called peaks.

| Data access links<br>*May remain private before publication.* | *For "Initial submission" or "Revised version" documents, provide reviewer access links. For your "Final submission" document, provide a link to the deposited data.* |
| Files in database submission | *Provide a list of all files available in the database submission.* |
| Genome browser session<br>(e.g. UCSC) | *Provide a link to an anonymized genome browser session for "Initial submission" and "Revised version" documents only, to enable peer review. Write "no longer applicable" for "Final submission" documents.* |

## Methodology

| Replicates | *Describe the experimental replicates, specifying number, type and replicate agreement.* |
| Sequencing depth | *Describe the sequencing depth for each experiment, providing the total number of reads, uniquely mapped reads, length of reads and whether they were paired- or single-end.* |
| Antibodies | *Describe the antibodies used for the ChIP-seq experiments; as applicable, provide supplier name, catalog number, clone name, and lot number.* |
| Peak calling parameters | *Specify the command line program and parameters used for read mapping and peak calling, including the ChIP, control and index files used.* |
| Data quality | *Describe the methods used to ensure data quality in full detail, including how many peaks are at FDR 5% and above 5-fold enrichment.* |
| Software | *Describe the software used to collect and analyze the ChIP-seq data. For custom code that has been deposited into a community repository, provide accession details.* |

# Flow Cytometry

## Plots

Confirm that:

☐ The axis labels state the marker and fluorochrome used (e.g. CD4-FITC).

☐ The axis scales are clearly visible. Include numbers along axes only for bottom left plot of group (a 'group' is an analysis of identical markers).

☐ All plots are contour plots with outliers or pseudocolor plots.

☐ A numerical value for number of cells or percentage (with statistics) is provided.

## Methodology

| | |
|---|---|
| Sample preparation | *Describe the sample preparation, detailing the biological source of the cells and any tissue processing steps used.* |
| Instrument | *Identify the instrument used for data collection, specifying make and model number.* |
| Software | *Describe the software used to collect and analyze the flow cytometry data. For custom code that has been deposited into a community repository, provide accession details.* |
| Cell population abundance | *Describe the abundance of the relevant cell populations within post-sort fractions, providing details on the purity of the samples and how it was determined.* |
| Gating strategy | *Describe the gating strategy used for all relevant experiments, specifying the preliminary FSC/SSC gates of the starting cell population, indicating where boundaries between "positive" and "negative" staining cell populations are defined.* |

☐ Tick this box to confirm that a figure exemplifying the gating strategy is provided in the Supplementary Information.

# Magnetic resonance imaging

## Experimental design

| | |
|---|---|
| Design type | *Indicate task or resting state; event-related or block design.* |
| Design specifications | *Specify the number of blocks, trials or experimental units per session and/or subject, and specify the length of each trial or block (if trials are blocked) and interval between trials.* |
| Behavioral performance measures | *State number and/or type of variables recorded (e.g. correct button press, response time) and what statistics were used to establish that the subjects were performing the task as expected (e.g. mean, range, and/or standard deviation across subjects).* |

## Acquisition

| | |
|---|---|
| Imaging type(s) | *Specify: functional, structural, diffusion, perfusion.* |
| Field strength | *Specify in Tesla* |
| Sequence & imaging parameters | *Specify the pulse sequence type (gradient echo, spin echo, etc.), imaging type (EPI, spiral, etc.), field of view, matrix size, slice thickness, orientation and TE/TR/flip angle.* |
| Area of acquisition | *State whether a whole brain scan was used OR define the area of acquisition, describing how the region was determined.* |

Diffusion MRI    ☐ Used    ☐ Not used

## Preprocessing

| | |
|---|---|
| Preprocessing software | *Provide detail on software version and revision number and on specific parameters (model/functions, brain extraction, segmentation, smoothing kernel size, etc.).* |
| Normalization | *If data were normalized/standardized, describe the approach(es): specify linear or non-linear and define image types used for transformation OR indicate that data were not normalized and explain rationale for lack of normalization.* |
| Normalization template | *Describe the template used for normalization/transformation, specifying subject space or group standardized space (e.g. original Talairach, MNI305, ICBM152) OR indicate that the data were not normalized.* |
| Noise and artifact removal | *Describe your procedure(s) for artifact and structured noise removal, specifying motion parameters, tissue signals and physiological signals (heart rate, respiration).* |

| Volume censoring | *Define your software and/or method and criteria for volume censoring, and state the extent of such censoring.* |

## Statistical modeling & inference

| Model type and settings | *Specify type (mass univariate, multivariate, RSA, predictive, etc.) and describe essential details of the model at the first and second levels (e.g. fixed, random or mixed effects; drift or auto-correlation).* |

| Effect(s) tested | *Define precise effect in terms of the task or stimulus conditions instead of psychological concepts and indicate whether ANOVA or factorial designs were used.* |

Specify type of analysis: ☐ Whole brain ☐ ROI-based ☐ Both

| Statistic type for inference (See Eklund et al. 2016) | *Specify voxel-wise or cluster-wise and report all relevant parameters for cluster-wise methods.* |

| Correction | *Describe the type of correction and how it is obtained for multiple comparisons (e.g. FWE, FDR, permutation or Monte Carlo).* |

## Models & analysis

| n/a | Involved in the study |
|-----|----------------------|
| ☐ | ☐ Functional and/or effective connectivity |
| ☐ | ☐ Graph analysis |
| ☐ | ☐ Multivariate modeling or predictive analysis |

| Functional and/or effective connectivity | *Report the measures of dependence used and the model details (e.g. Pearson correlation, partial correlation, mutual information).* |

| Graph analysis | *Report the dependent variable and connectivity measure, specifying weighted graph or binarized graph, subject- or group-level, and the global and/or node summaries used (e.g. clustering coefficient, efficiency, etc.).* |

| Multivariate modeling and predictive analysis | *Specify independent variables, features extraction and dimension reduction, model, training and evaluation metrics.* |

