## [Peer Review File · Nature Genetics]

Peer Review Information

Manuscript Title: Genetic immune escape landscape in primary and metastatic cancer

Corresponding author name(s): Professor Edwin Cuppen, Dr Francisco Martinez-Jimenez

Reviewer Comments & Decisions:

Decision Letter, initial version:

8th Aug 2022

Dear Professor Cuppen,

I'm sorry it's taken so long to return this decision to you. Thanks so much for bearing with me.

Your Article, "Genetic immune escape landscape in primary and metastatic cancer" has now been seen by 3 referees. You will see from their comments below that while they find your work of interest, some important points are raised. We are interested in the possibility of publishing your study in Nature Genetics, but would like to consider your response to these concerns in the form of a revised manuscript before we make a final decision on publication.

To guide the scope of the revisions, the editors discuss the referee reports in detail within the team, including with the chief editor, with a view to identifying key priorities that should be addressed in revision and sometimes overruling referee requests that are deemed beyond the scope of the current study. In this case, we'd like you to prioritise the comments regarding the benchmarking and validation of LILAC. But we would also expect you to address all the other comments in full, experimentally where possible, or textually.

You'll also see that Reviewer #2 has referred to data availability. As you may know, Nature Genetics mandates full and free access to all legitimate researchers at the point of acceptance. If there are any restrictions to data access, your data availability statement must clearly describe them, describe why they are in place, and indicate which datasets restrictions apply to. Please could you update your data availability statement (and reporting summary) with this in mind. If there are any issues, please feel free to contact me to discuss. You can read more about our data availability policy here:

<https://www.nature.com/nature-portfolio/editorial-policies/reporting-standards#availability-of-data>

We therefore invite you to revise your manuscript taking into account all reviewer and editor comments. Please highlight all changes in the manuscript text file. At this stage we will need you to upload a copy of the manuscript in MS Word .docx or similar editable format.

*2) If you have not done so already please begin to revise your manuscript so that it conforms to our Article format instructions, available

http://www.nature.com/ng/authors/article_types/index.html here.

*3) Include a revised version of any required Reporting Summary:

Please be aware of our <https://www.nature.com/nature-research/editorial-policies/image-integrity> guidelines on digital image standards.

[REDACTED]

We hope to receive your revised manuscript within four to eight weeks but we can be completely flexible on this point. If you cannot send it within this time, please let us know and we'd be happy to discuss alternative timeframes.

Nature Genetics is committed to improving transparency in authorship. As part of our efforts in this direction, we are now requesting that all authors identified as 'corresponding author' on published papers create and link their Open Researcher and Contributor Identifier (ORCID) with their account on the Manuscript Tracking System (MTS), prior to acceptance. ORCID helps the scientific community achieve unambiguous attribution of all scholarly contributions. You can create and link your ORCID

from the home page of the MTS by clicking on 'Modify my Springer Nature account'. For more information please visit www.springernature.com/orcid.

Sincerely,

Safia Danovi
Editor
Nature Genetics

Referee expertise:

Referee #1: cancer genomics, immune selection, evolution

Referee #2: immune escape

Referee #3: cancer genomics

Reviewers' Comments:

Reviewer #1:

Remarks to the Author:

The manuscript of Martinez-Jimenez et al. provides a comprehensive characterization of genetic immune escape (GIE) in primary and metastatic tumor samples. They find that +/- 25% of all tumor samples harbor GIE (mainly LOH of HLA) and that this is comparable between primary and metastatic samples, suggesting that GIE is a relatively early event during tumour evolution.

The manuscript is very well written and easy to follow. This is a timely study that has the potential to become a core reference for GIE in the coming years. While the high prevalence of GIE is not new, I particularly liked the comprehensive nature of the study and the insights in immune selection processes that arise (e.g., the analysis shown in figure 4 is very important and convincing i.m.o.).

My main concern is the lack of proper validation of LILAC, the newly developed method that was used to determine HLA LOH, the main form of GIE. This algorithm is capable of 1) calling MHC-I genotypes, 2) calling somatic mutations in MHC-I genes and 3) determining LOH in HLA genes. While other methods have been developed to accomplish these tasks individually (or 2 at most), LILAC is novel in its integrative nature.

The authors validate the genotyping accuracy in figs. 1c-1e in a rather indirect way. They compare the % agreement between normal and tumor samples with xHLA and Polysolver (fig. 1c-1d) and then perform an orthogonal validation on 95 samples (fig. 1e). However, the later samples were heavily filtered (methods section) and I mainly found the filter "disagreement of LILAC with either xHLA or Polysolver" concerning. Therefore, I think the authors should benchmark their tools on an independent

dataset and using an independent gold standard (e.g., 1000 genomes data could be well suited), before concluding "LILAC showed a near-perfect HLA typing performance on WGS data" (discussion section). Additionally, I recommend they also compare to other state-of-the-art genotyping tools such as Optitype. Similarly, I lacked convincing validation analyses for the HLA somatic mutation and the LOH detection property. The analysis shown in fig. 1f has limited value i.m.o.. How does the tool compare to LOHHLA (McGranahan 2017)? This comparison would be highly interesting because these authors demonstrated high prevalence of HLA LOH in lung cancer before. LOHHLA was later used to demonstrate similarly high prevalences in colorectal cancer, ... (e.g., Lakatos 2020).

One of the challenges in calling somatic mutations in HLA genes is the highly polymorphic nature and the risk of identifying germline variants as somatic variants. Is there any possibility that the hot spot identified in HLA-A is a germline variant? What is the variant allele frequency? Do these mutations occur in 1 or multiple cancers? Some additional (negative control) analyses seem necessary to make these results convincing. The authors could consider showing that these clusters do not exist for silent mutations, are restricted to 1 cancer type, show the absence on non-cancer datasets, ...

One of the limitations of LILAC is its restriction to MHC-I. As stated higher the 3 core functionalities (genotyping, HLA LOH, mutations) are not new as such, so I believe an extension to MHC-II genes would vastly increase the novelty.

Further, is LILAC also applicable on WES? If no, why not?

In fig. 5 and fig. 6, the authors demonstrate a correlation between TMB (and other, related variables) and GIE, first based on a logistic regression approach (fig. 5), then based on a TMB "bucket"/binning approach (fig. 6). Why not just plotting TMB on a continuous scale? More importantly, I was lacking a negative control in panels c and d. Isn't it rather trivial that the likeliness of somatic mutations in GIE pathways (or any other pathway) increase when TMB is high? I recommend the authors consider some random background control here or show the lack of these mutations in matched (similar coverage) non-immune pathways. Relatedly, if GIE becomes so prevalent in hypermutated cancers, wouldn't one expect that these are the least responsive tumors to immunotherapy (rather than the most responsive)?

If GIE is indeed an important mechanism during tumor evolution (as suggested by the positive selection analyses), one would expect some degree of mutual exclusivity between the different GIE events (i.e., one GIE mechanisms relieves the selection pressure on the other genes). Have the authors checked this?

I found it highly interesting that the authors conclude that GIE is an early event during tumour evolution. Interestingly, this conclusion is similar to other, orthogonal Nature Genetics studies such as Lakatos et al. 2020 (tumor modelling) and Van den Eynden et al., 2019 (lack of neoantigen depletion). It would be interesting to see a broader perspective on these findings in discussion section.

Minor comments:

- The authors state that "GIE may be further fueled by LOH of HLA-I in later tumorigenic stages". It's unclear for me where this conclusion comes from.
- High proportions of HLA LOH were found in PANET in KICH. The non-focality and lack of other GIE mechanisms suggest that the main driver is not HLA but another gene on chr. 6. Could the authors speculate which driver this could be?
- I find fig. 2h unnecessary complex with all the triangles etc. Why not just use a continuous scale and directly show the odds ratios?
- I understand that dN/dS was calculated using the dNdScv method? To avoid confusion for the reader

I would avoid using these terms interchangeably.

- Fig. 3f-h. The difference between "highly-focal LOH" and "(non-)focal LOH" is clear for me but between "focal LOH" and "non-focal LOH" is rather vague. Could the authors clarify how this distinction is made? Additionally, the y-axis label indicates "samples with ... LOH HLA". I understand the plot refers the genome-wide presence of LOH (and not just HLA)?
- Fig. 3a. Higher-than-1 dN/dS values are not found for missense mutations. However, some missense mutations could have a profound impact in a protein's functionality, similar like a nonsense mutation. A further distinction between high- and low impact mutations could be useful. Additionally, do dN/dS results remain the same if the "hot spots" mutations shown in panel i are excluded?
- Analysis shown in fig. 5a is limited to 16 genes (or 18 genes according to main text?). Positive selection in most of these genes is already known from previous studies. What about other genes involved in the 5 non-HLA pathways? Further, while selection of nonsense mutations make sense for genes with a tumor suppressor role, the opposite is expected for genes such as PDL1, CD28, ... (increased activity necessary for GIE). Could the authors comment on this?
- P8. Some references to Fig. 3e and 3f seem to be wrong (think it should be d and e).
- At the end of the first paragraph of p. 8 the authors conclude there is purifying selection against homozygous deletions. Could the authors clarify why this would be?
- The text on p8 is jumping from dNdS somatic mutations results, to LOH results and back to somatic mutations. I found this confusing
- Could it be that the association of most genomic features (mutational signatures, ...) shown in fig. 6 is secondary to the association with TMB?

Reviewer #2:

Remarks to the Author:

Summary:

This manuscript by Martinez-Jimenez represent a very interesting broad analysis of genetic alterations affecting immune evasion in primary and metastatic cancers, using published and new cohorts. Overall the findings are new and significant; the methods are overall well explained. The manuscript would benefit from some edits and additional analyses to further strengthen the main conclusions. Furthermore, all the data must be made available as de-identified data to the academic community priori to publication.

Main points:

1)

One of the strengths of the paper is the concomitant analysis of primary AND metastatic tumors also thanks to a new dataset of metastatic tumors from the Hartwig foundation.

One very important point before publication is that de-identified data from the Hartwig foundation used for this publication have to be available immediately to the academic community through platforms such as dbGap to allow reproducibility of the results.

2)

The authors should state more clearly which are the somatic alterations that are included for the GIE genes to classify a tumor as having a GIE event: Deletions? Loss of function mutations? Any missense mutations? Silent mutations? How was this corrected for the total N of mutations or the total N of copy number alterations?

3)

In general, the analyses involving the GIE alterations should all involve a comparison to a random set of 'neutral' genes through bootstrapping or other type of randomization.

This is important for example in the context of the analysis shown in Fig. 6a.

4)

Regarding the alterations in GIE, in the supplemental table the authors should specify the frequency of mutations or alteration for each of the 21 genes in each tumor type, without pooling the genes by pathways but explicitly reporting the data for each of the GIE gene.

5)

In the GIE evaluation LOH of HLA is the top pathway altered and mainly driving the frequency of GIE alteration. LOH is considered only for HLA, not for other GIE genes — and this makes sense — but the authors should consider whether HLA LOH is a marker of general chromosomal instability and control for the overall rate of LOH across the genome.

6)

Given the fact that chr6 loss could be driven by other genes different than HLA as the authors point out, for the GIE only the focal events (<3Mb) should be considered. This is also because the authors showed that 'Focal LOH of HLA-I preferentially targets the alleles that presents the highest neoepitope repertoire', suggesting that these focal events are the truly important ones for immune evasion. Therefore, in all analyses involving GIE, the authors should calculate a GIE score and a GIE score based on focal-only-HLA LOH (excluding non focal LOH).

7)

Can the authors distinguish and study the different effect of LOH due to deletion of one chromosome and copy-number neutral LOH? This could tell whether the main effect of LOH is LOH itself or decrease dosage of HLA genes/molecules.

8)

Last section of the paper:

This section is summarized in the abstract by the sentence:

"Finally, there is a strong tendency for mid and high tumor mutation burden (TMB) tumors to preferentially select LOH of HLA-I for GIE whereas hypermutated samples favor global immune evasion strategies."

The authors should make clear in this sentence of the abstract that they refer to FOCAL LOH of HLA-I (they should add the word 'focal'. Furthermore, it is not clear from this sentence what does GLOBAL immune evasion strategies mean; perhaps 'global' should be substituted by 'other'.

The authors should show the analysis done in Fig. 6a excluding HLA LOH to see if the associations hold true or not and in which tumor types.

The analysis shown in fig 7c is very interesting, but would benefit of some clarifications:

- can the authors show the same analysis just including the non clonal TMB?
- is this analysis a pan-cancer analysis? what happens if the authors control for tumor type? (I would restrict the analysis to tumor types showing a large spectrum/variance of TMB)
- the increase in GIE alterations correlating with TMB could just be due to the fact that high TMB tumors have a lot of mutations; does this correlation hold true also if the authors disregard point mutations and just consider deletion and amplifications?

9)

The Discussion would benefit from more insights and thoughts on the underlying mechanisms of the new findings of the paper, such as the one described in Fig, 7.

Reviewer #3:

Remarks to the Author:

The manuscript from Martinez-Jimenez, Cuppen and colleagues presents their analyses of over 6000 cancer genomes evaluated for evidence of genetic mechanisms of presumptive immune escape (GIE). They introduce a new tool, LILAC to improve the complicated business of mutation calling in the HLA-I locus. They find that genetic mechanisms of immune evasion are prevalent pan-cancer and that, importantly, these are present before metastases. In addition, they present analyses which provide insight into the selective drive imparted by mutation burden, mutational processes, and other facets of the genomic landscape.

I found this to be a fascinating and well-executed study. The sample size is substantial and truly drives the strength of the analyses and the, I think, important conclusions - particularly around assumptions around metastatic disease and immune evasion. There has been ample thought given to the vagaries of genome-scale statistics, the algorithmic considerations are clearly laid out, and the manuscript strikes a good balance of detail and 'big-picture' that conveys the overall intent without overstating and claiming easy translational impact. I have few concerns over the approach, the analyses, and the current conclusions as presented.

I have several suggestions, given the findings presented and considering the power of the dataset assembled that could strengthen and expand the insights.

Given the fact that there are over 6000 genomes in play in nearly 2000 primary and 4000 metastatic/advanced buckets, there is an opportunity to delve more definitely into the timing of genetic/genomic events.

Firstly, can the authors estimate the clonal or subclonal nature of the HLA-I events themselves? This becomes a rather interesting facet of cancer genome evolution to get to grips with. Further, placing the GIE events in context of the acquisition of the known cancer gene mutations that drive the various cancer types under study is needed. Is there an order that emerges by tissue of origin/tumour type?

Is there an order associated with particular driver mutations?

The same could be asked of genome doubling, and more grossly, aneuploidy as defined by breakpoint prevalence. Putting the GIE into this estimated temporal framework would further strengthen and indeed augment the important primary versus metastases observation.

Further, are cancer driver gene mutations correlated with type/extent of GIE? This is explored to an extent in the mismatch repair gene data but needs to be extended to other cancer genes – those associated with driving proliferation, as an example, or those associated with deregulated chromatin. Again, having a notion of order of events here further enriches these possible insights. This should be extended to copy number/rearrangement driver events as well.

Following on from this, and again exploiting the fact of having a large number of cancer genomes in play, for the mutational signature associations with GIE, can the authors explore relative timing of the signature processes becoming active (or going dark) relative to the GIE events? Do the GIE events driven at the point mutation level overall bear the signatures of the most prevalent mutational processes?

Whilst treatment is given some consideration in the supplemental figures, it is not particularly clearly called out in the main body of the manuscript. I would consider making the apparent lack of statistical correlation with chemotherapy/immunotherapy in the post treatment metastatic cases a point in the main text. It further highlights the importance of the wheels being set in motion in GIE regardless of subsequent treatment.

I assume not, but were there any paired primary/metastatic pairs in the dataset? If so, was there sufficient opportunity to explore immune editing correlation with extent of GIE?

Whilst the correction for molecular age with respect to mutational signatures is appreciated, a different question that arises is if GIE presence in and of itself is correlated with chronological age of the patient. This, of course, ties in with the consideration of GIE timing and clonality overall. One might hypothesize it being more prevalent in generally later onset cancers given the general notion of decreased immune 'tone' in advancing age, for instance.

Is it possible to infer TCR/BCR clonality metrics in the sample series, particularly from those having matched RNAseq? This would potentially augment the nice data re immune infiltrates and GIE presence/absence.

Did the authors consider an assessment of the tumor microbiome and correlation with GIE in this dataset? Given the growing body of evidence (much still somewhat associative in nature, to be fair) of the tumor microbiome in the biology of the cancer, having genomes presents a rather unique opportunity to extend some of these studies in this immune escape context. Further, the association with particular tumor types and chemotherapy efficacy/exposures might be another avenue to walk down if the data allow.

Finally, given the uniqueness of the dataset, have the authors considered assessment of matching germline (those from PBMC) for any evidence of CHIP GIE events? Is this a feature of aging in accompanying CHIP point mutations?

There is a tendency in the manuscript to use language of active involvement of the tumor in shaping its genome. "Tumors tailor", "tumors select", 'tumors leveraged' – this, of course, is all selection and fitness. It would better to avoid this sort of language in my view and rephrase as things being selected for, increased fitness – the parlance of tumor evolution. The former fuels the tumor as sentient agent of patient demise that is quite problematic for many patients and their advocates.

Point-by-point response to reviewer comments

Color text code:

Response to reviewers

Changes in the main text

Referee expertise:

Referee #1: cancer genomics, immune selection, evolution

Referee #2: immune escape

Referee #3: cancer genomics

Reviewers' Comments:

Reviewer #1:

Remarks to the Author:

The manuscript of Martinez-Jimenez et al. provides a comprehensive characterization of genetic immune escape (GIE) in primary and metastatic tumor samples. They find that +/- 25% of all tumor samples harbor GIE (mainly LOH of HLA) and that this is comparable between primary and metastatic samples, suggesting that GIE is a relatively early event during tumour evolution.

The manuscript is very well written and easy to follow. This is a timely study that has the potential to become a core reference for GIE in the coming years. While the high prevalence of GIE is not new, I particularly liked the comprehensive nature of the study and the insights in

immune selection processes that arise (e.g., the analysis shown in figure 4 is very important and convincing i.m.o.).

We are very glad that the reviewer appreciates the importance of our work.

My main concern is the lack of proper validation of LILAC, the newly developed method that was used to determine HLA LOH, the main form of GIE. This algorithm is capable of 1) calling MHC-I genotypes, 2) calling somatic mutations in MHC-I genes and 3) determining LOH in HLA genes. While other methods have been developed to accomplish these tasks individually (or 2 at most), LILAC is novel in its integrative nature.

The authors validate the genotyping accuracy in figs. 1c-1e in a rather indirect way. They compare the % agreement between normal and tumor samples with xHLA and Polysolver (fig. 1c-1d) and then perform an orthogonal validation on 95 samples (fig. 1e). However, the later samples were heavily filtered (methods section) and I mainly found the filter “disagreement of LILAC with either xHLA or Polysolver” concerning. Therefore, I think the authors should benchmark their tools on an independent dataset and using an independent gold standard (e.g., 1000 genomes data could be well suited), before concluding “LILAC showed a near-perfect HLA typing performance on WGS data” (discussion section). Additionally, I recommend they also compare to other state-of-the-art genotyping tools such as Optitype.

We agree with the reviewer that extending our benchmarking to other datasets would further strengthen our claims. However, we would also like to clarify that our experimental validation was explicitly designed to address the most challenging HLA-I typing cases with sample availability (i.e., those cases that there is no agreement across the three tools and those cases with rare alleles that are usually undercalled by most of HLA-I typing tools). Moreover, although we did not include an explicit comparison with Optitype (among other HLA-I typing tools), xHLA carried out a performance comparison with Optitype in their publication (DOI:10.1073/pnas.1707945114), where xHLA showed a slightly better performance. That is the reason we decided to use xHLA as a reference tool for comparison in the first place.

In any case, following the reviewer suggestion, we have improved the benchmarking of LILAC HLA-typing performance by:

1. **Clarifying the sample selection criteria.** The underlying criteria for selection for the experimental validation was **i)** the material availability of samples for the experiments, **ii)** prioritization of samples with disagreement between the three tools (i.e., LILAC, Polysolver and xHLA) and **iii)** samples with rare predicted alleles as they may represent a challenging scenario compared to more prevailing HLA-I alleles. As mentioned above, these samples likely present the most challenging HLA allotypes as indicated by the lack of consensus

HLA-I types. We have clarified the criteria filtering used to select the samples to be orthogonally validated as we believe it was not properly explained in the manuscript.

2. **Validating LILAC's HLA-typing performance using an independent dataset of family trios with divergent genetic ancestries.** We have assessed LILAC's performance using three family trios with diverse genomic ancestries and with available WGS. Specifically, we tested LILAC's HLA-I typing accuracy in the Illumina Platinum trios and the Yoruban trio with african ancestry. In total 54 alleles were tested and all of them were correctly matched by LILAC's inferred HLA-I types.

Next, we evaluated LILAC's HLA-I typing performance on three family trios with diverse genetic ancestries (see methods), where it displayed a perfect agreement with previously reported HLA-I types (Fig. 1d).

3. **Evaluating LILAC's agreement with the HLA-I typing performed in TRACERx lung dataset.** We have assessed LILAC's HLA typing agreement with the HLA typing originally used by LOHHLA on the TRACERx 100 lung cohort (McGranahan N, et al. Cell. 2017). On the one hand, this analysis demonstrates LILAC's applicability to whole-exome sequencing samples (one of the questions of the reviewer, see below). On the other hand, it provides an estimation of LILAC's agreement with Optitype as, according to their methods, a large fraction of the HLA-I typing was originally performed by Optitype.

We then processed 100 tumor samples from the TRACERx lung cohort and compared LILAC's HLA-I typing with the HLA-I types used in the LOHHLA publication (data provided by the authors via personal communication). Our comparison revealed a nearly perfect agreement between LILAC and the HLA-I typing provided by the authors (2-field perfect match in 481 out of the 490 tested alleles (98.16%)). Of note, we could not test all 600 alleles because homozygous HLA-I calls were not shared by the authors (i.e., 110 homozygous alleles were missing from their data).

LILAC's agreement with TRACERx Lung 100

Furthermore, we demonstrated whole-exome sequencing (WES) applicability by running LILAC on the TRACERx100 lung cohort, where it showed a 98.16% agreement with the HLA-I types originally reported in the publication (Fig. 1e).

Altogether, we believe that these results further support the robustness of LILAC to perform HLA-I typing based on high-quality WGS or WES data. These analyses have been integrated into the results section (Inference of HLA-I tumor status with LILAC) as well as in the accompanying figures and supplementary tables (Figure 1, Supp. Figure 1 and Supp. Table 2).

Similarly, I lacked convincing validation analyses for the HLA somatic mutation and the LOH detection property. The analysis shown in fig. 1f has limited value i.m.o.. How does the tool compare to LOHHLA (McGranahan 2017)? This comparison would be highly interesting because these authors demonstrated high prevalence of HLA LOH in lung cancer before. LOHHLA was later used to demonstrate similarly high prevalences in colorectal cancer, ... (e.g., Lakatos 2020).

We also agree that given the prevalence of LOH of HLA-I as a mechanism of immune escape it is highly relevant to illustrate how LILAC compares to other state-of-the-art tools for LOH of HLA-I. To address this, we then aimed to compare LILAC and LOHHLA agreement. Initially, we tried to run LOHHLA on our cohort, but we were unable to properly run without major adjustments of its original source code. This is likely caused because LOHHLA was originally designed and tested in a high-coverage tumor sequencing setting (>300x sequencing depth) which is clearly not the case for PCAWG (varying between 30x and 90x tumor sequencing depth) nor for Hartwig (~109 x tumor sequencing depth).

Instead, we decided to run LILAC on the TRACERx lung cohort (EGAS00001002247). Since identification of LOH of HLA-I by LILAC requires tumor purity and copy number estimations (and the HLA-I typing already computed by LILAC) we entirely re-processed 100 tumor-normal paired WES samples from TRACERx cohort using our tumor analytical pipeline

(<https://github.com/hartwigmedical/hmftools>). Several adjustments were made to adapt our pipeline, originally developed to work with WGS data, to the characteristics of this WES dataset. Finally, the comparison was made by comparing the LOHHLA calls provided by the authors (via personal communication) with LILAC's output.

As illustrated by the image above (now Fig. 1g) LOHHLA and LILAC showed an overall agreement of 90% on their LOH of HLA-I calling in the WES TRACERx lung dataset. Importantly, this agreement was substantially higher in high-tumor purity samples (i.e., tumor purity ≥ 0.3 with a 96% agreement) than in low-tumor purity samples (i.e., tumor purity < 0.3 , agreement of 75.51%). This clearly illustrates the challenges associated with performing a univocal copy number estimation in WES samples with low tumor content. In this regard, it is important to mention that all samples considered in our dataset have a minimum tumor purity of 0.20 (which is not the case in the TRACERx dataset) and 88% of them have a tumor purity higher than 0.3 (see Supp. Table 3), where the copy number estimations are highly reliable.

See results from main text:

Moreover, we quantified LILAC's LOH of HLA-I agreement with LOHHLA¹⁴ in the TRACERx lung WES cohort. LILAC and LOHHLA LOH estimations displayed a global 90% agreement (Fig. 1g and Supp. Table 2). Importantly, high tumor purity samples showed considerably better concordance than low purity samples (96.08% in samples with tumor purity ≥ 0.3 , 75.51% when tumor purity < 0.3), reflecting increased challenges for genome-wide copy number loss calling in low purity (WES) samples.

One of the challenges in calling somatic mutations in HLA genes is the highly polymorphic nature and the risk of identifying germline variants as somatic variants. Is there any possibility that the hot spot identified in HLA-A is a germline variant? What is the variant allele frequency? Do these mutations occur in 1 or multiple cancers? Some additional (negative control) analyses seem necessary to make these results convincing. The authors could consider showing that these clusters do not exist for silent mutations, are restricted to 1 cancer type, show the absence on non-cancer datasets, ...

We agree with the reviewer that performing somatic calling in HLA genes requires exceptional attention. To further inquiry about the somatic nature of HLA-I gene mutations in general, and the hotspot mutations of HLA-A, in particular, we performed the following analyses:

1. Comprehensive characterization of the HLA-A hotspot mutations.

There were 6 metastatic patients from the Hartwig cohort that harbored 7 recurrent frameshift indels at the Lys210 of HLA-A (i.e., hg19 genomic coordinates chr6:29911899) (see Fig. 4f and table below). This genomic region overlaps with a homopolymer of 7 cytosines "ACCCCC" (chr6:29911899-chr6:29911906). Interestingly, all these tumors present microsatellite instability (MSI), which explains its susceptibility for one base insertion/deletion at the start of the homopolymer.

gene	csq	protein_pos	cds_pos	purple_vaf	raw_vaf	position	ref	alt	type_mut	sample_id	msi_status	cancer_type
HLA-A	frameshift_variant&splice_region_variant	207	621	0.3927	0.160	29911899	A	AC	truncating_variant	HMF004509A	MSI	CUP
HLA-A	frameshift_variant&splice_region_variant	207	621	0.5163	0.264	29911899	AC	A	truncating_variant	HMF004249A	MSI	Small intestine cancer
HLA-A	frameshift_variant&splice_region_variant	207	621	0.8210	0.452	29911899	AC	A	truncating_variant	HMF001099A	MSI	Colorectum carcinoma
HLA-A	frameshift_variant&splice_region_variant	207	621	0.6004	0.306	29911899	A	AC	truncating_variant	HMF000644A	MSI	Colorectum carcinoma
HLA-A	frameshift_variant&splice_region_variant	207	621	0.3355	0.171	29911899	AC	A	truncating_variant	HMF000644A	MSI	Colorectum carcinoma
HLA-A	frameshift_variant&splice_region_variant	207	621	0.4049	0.264	29911899	A	AC	truncating_variant	HMF002548A	MSI	Non small cell lung cancer
HLA-A	frameshift_variant&splice_region_variant	207	621	0.4021	0.176	29911899	AC	A	truncating_variant	HMF004149A	MSI	Non small cell lung cancer

Moreover, the variant allele frequency (VAF) of these mutations are far below 0.5 (the expected VAF for heterozygous germline polymorphisms), supporting its somatic nature.

“The main exception was the recurrent HLA-A Lys210 frameshift indel (chr. 6 at 29911899), which was observed in six MSI metastatic patients. This genomic region overlaps with a homopolymer repeat of “CCCCCC”, which likely explains its susceptibility for one base insertion/deletion at the start of the homopolymer.”

2. Analysis of genomic distribution of synonymous mutations overlapping to HLA-I genes.

It is conceived that individual synonymous mutations do not generally provide a functional advantage. As a consequence, one would expect a near-random accumulation of somatic synonymous mutations along the HLA-I genes, which would be mainly shaped by the background mutation rate. To confirm this we examined the distribution of somatic synonymous mutations in HLA-I genes in our primary and metastatic datasets. In both cases, the observed distribution revealed a uniform distribution that did not show any kind of clustering of mutations (see below). This results shows that the clustering of mutations was exclusive to frameshift indels in MSI samples (which are possibly selected because of loss-of-function capacity).

Hartwig cohort

PCAWG cohort

These analyses, alongside the orthogonal validation performed for nine somatic mutations (see results Inference of HLA-I tumor status with LILAC) demonstrate the robustness of our tumor analytical pipeline to identify genuine somatic mutations in the HLA-I locus.

One of the limitations of LILAC is its restriction to MHC-I. As stated higher the 3 core functionalities (genotyping, HLA LOH, mutations) are not new as such, so I believe an extension to MHC-II genes would vastly increase the novelty.

We thank the reviewer for this interesting suggestion. However, extending LILAC to HLA-II would also require an extensive and dedicated validation of LILAC's performance on HLA-II (including HLA-II typing and LOH/somatic mutation identification in this locus). Moreover, we still lack sufficient knowledge of the role of this locus in tumor immune surveillance escape and we prefer to focus this -already extensive- analysis on pathways which have known reported influence in immune escape. It is nevertheless a valuable suggestion for a follow-up study.

However, in our dataset, three of four patients did not harbor GIE events, highlighting the need to characterize other mechanisms of immune evasion. These may involve not only alternative molecular pathways such as the HLA-II⁴⁰ and the killer immunoglobulin-like receptor³⁹ (KIR); but also other types of alterations such as germline variants⁴¹ and epigenetic modifications^{2,42}.

Further, is LILAC also applicable on WES? If no, why not?

Yes, LILAC is capable of performing HLA-I typing in WES samples with sufficient coverage of the exonic regions of HLA-I genes. We demonstrated its applicability by processing 100 tumor-normal paired WES samples from TRACERx lung cohort and showing high agreement between our analytical framework HLA-I types used in their publication.

Moreover, our tumor analytical pipeline (including LILAC) has been used to detect LOH of HLA-I in the aforementioned WES dataset, where it also displayed a good agreement with the LOHHLA LOH calls. Nevertheless, we would like to emphasize that our tumor analytical pipeline (including LILAC) has been primarily optimized and tested for high-quality WGS data.

In fig. 5 and fig. 6, the authors demonstrate a correlation between TMB (and other, related variables) and GIE, first based on a logistic regression approach (fig. 5), then based on a TMB "bucket"/binning approach (fig. 6). Why not just plotting TMB on a continuous scale?

We guess the reviewer refers to Fig. 7 (new Figure 8). This figure was meant to illustrate the relationship (or lack of) between the TMB (or neoepitope burden) and the likelihood of GIE. Hence,

we decided to group tumor samples with similar TMB (or neoepitopes burden, respectively) so one could compute the GIE frequency in samples within the same TMB range. Plotting TMB on a continuous scale would not enable a frequency/likelihood estimation (only presence/absence of GIE at sample specific level) and therefore would hamper the study of the association between GIE frequency and TMB on a global scale.

More importantly, I was lacking a negative control in panels c and d. Isn't it rather trivial that the likeliness of somatic mutations in GIE pathways (or any other pathway) increase when TMB is high? I recommend the authors consider some random background control here or show the lack of these mutations in matched (similar coverage) non-immune pathways.

This is an important point also raised by reviewer #2. To address this issue we have performed 100 simulations of GIE alterations (see methods background simulation of GIE alterations) and used these simulations as background control (hereafter referred to as simulated GIE rates) in the analyses performed in Figure 7 and Figure 8.

Specifically, in the association between genomic features and GIE displayed in Figure 7 (old figure 6), we now filtered out associations that could be explained by higher background mutation rate (including structural variant and copy number variant rates) and kept those exclusively associated with GIE (i.e., significant association [q-value < 0.05] and background control simulations not showing a prevalent association [$\leq 2\%$ of GIE simulations showing a significant association]). As expected, some significant GIE-feature associations were also associated with higher simulated GIE rates (e.g., colorectal and breast cancer TMB association, ovarian and pancreatic cancer SV load, among others; see Supp. Fig. 4a), whereas other features showed an exclusive association with GIE (e.g., APOBEC mutation load in breast, urothelial and NSCLC; MMRd in NSCLC and stomach cancer, immune infiltration in colorectal cancer, HPV+ in colorectal cancer, etc.) (see Fig. 7 and Supp. Fig. 4). The revised version of the manuscript (results GIE association with cancer genomic features) and figures (Figure 7 and Supp. Fig. 4) have been updated to accommodate for the background control.

We believe this issue is also important in the context of analyzing the relationship between the TMB and GIE prevalence (i.e., last results section, analysis displayed in Figure 8). Therefore, we have also included the background simulated GIE rates as a surrogate of the expected GIE frequency.

Our results revealed that the observed GIE rates can not be fully explained by the increased background/CNV rates across the TMB buckets (Figure 8a-b, see below). Particularly, as the TMB increases the observed GIE frequency deviates from the expected frequency given by the GIE simulations. This is particularly evident in hypermutated and ultra-hypermutated tumors,

which show a GIE incidence 2-3 fold higher than the simulations. Therefore, our results suggest that although the background alteration rate of tumors may play an important role in the frequency of GIE events in low and mid TMB tumors (mostly driven by the background LOH rates, see Supp. Fig. 3b and response to reviewer #2 below), the increased GIE frequency in high TMB tumors can not fully be attributed to the background rate of these tumors.

We have now amended the results section to consider this:

This trend was not fully explained by an increased background mutation and CNV rate (Fig. 8a simulated GIE). More specifically, as the TMB increases the observed GIE frequency deviates from the expected frequency given by the GIE simulations. This is particularly noticeable for (ultra)hypermuted tumors, which show a GIE incidence 2-3 fold higher than the simulations. This trend was still consistent after controlling for the cancer type (Supp. Fig. 5a). Furthermore, we also observed an association between increased clonal TMB and GIE prevalence, whereas the relationship with subclonal TMB burden was less evident (Supp. Fig. 5b). Using the burden of predicted neoepitopes based on the germline HLA-I profile as baseline also revealed a near-uniformly increasing distribution across the neoepitope buckets, which becomes sharper -and higher than the expected given the randomizations- after the 17th bucket (i.e., number of neoepitopes greater than 1,146 and lower than 1,971, see Fig. 8b). Altogether, our results indicated that GIE incidence increases with TMB and that selection for GIE events seem to play an important role primarily in high TMB tumors.

Figure 7

Relatedly, if GIE becomes so prevalent in hypermutated cancers, wouldn't one expect that these are the least responsive tumors to immunotherapy (rather than the most responsive)?

This is a very relevant point too. We believe that GIE events are required to elude the selective pressure imposed by the immune system, specifically in cancer types bearing a significant number of non-self antigens. These alterations would enable these tumors to maintain the extremely high mutation rates while rendering them invisible to the immune system. However, this delicate balance may be broken when the immune system's pressure is magnified after, for example, inhibiting tumor checkpoint blockade. Supporting this notion, in our cohort, patients pre-treated with ICI did not show a significant enrichment in GIE compared to untreated patients (see

results GIE association with cancer genomic features and response to reviewer #3). Also in line with this, recent work by Giovanni Germano et al. (DOI: 10.1158/2159-8290.CD-20-0987) showed that MMRd colorectal tumors may respond to ICI even in the absence of B2M. Finally, two reference pan-cancer studies showed that LOH of HLA-I is not negatively associated with ICB responses (DOI:10.1016/j.annonc.2020.04.004 and 10.1016/j.cell.2021.01.002). We now mention this matter in the discussion:

We observed a lack of association between prior exposure to cancer therapies, including immunotherapy, and higher GIE frequency. Consequently, the efficacy of immune escape alterations, such as LOH or HLA-I³¹, may be compromised when dealing with the strong immune pressure released by ICI. Dedicated studies are thus required to further understand the role of the different GIE alterations in immunotherapy responses.

If GIE is indeed an important mechanism during tumor evolution (as suggested by the positive selection analyses), one would expect some degree of mutual exclusivity between the different GIE events (i.e., one GIE mechanisms relieves the selection pressure on the other genes). Have the authors checked this?

This is indeed an interesting question. To perform a mutual exclusivity (ME) test, it is necessary to have a reasonable representation of the two groups for ME evaluation. Therefore, we restricted the analysis to evaluate whether, in a cancer type specific manner, LOH of HLA-I (the most frequent mechanism of immune evasion) was ME with other alterations beyond the HLA-I pathway (i.e., pathways from 2-6).

We did not find any evidence of mutual exclusivity between LOH of HLA-I and other GIE alterations (see Supp. Table 4 sheet pasted below and methods for mutual exclusivity test).

cancer_type	loh_and_other_gie	loh_not_other_gie	not_loh_and_other_gie	not_loh_not_other_gie	odds_ratio	pvalue_fisher	expected_mean_overlap	pvalue_simulations
Ovarian cancer	1	36	15	113	0.209259	0.085566	3.5980	0.1170
Prostate carcinoma	0	20	39	342	0.000000	0.122554	1.9324	0.1414
CUP	0	16	4	88	0.000000	0.521386	0.5970	0.5520
Colorectum carcinoma	11	117	39	453	1.092045	0.674741	10.2859	0.6569
Cervix carcinoma	3	14	3	20	1.428571	0.802823	2.5495	0.7483
Pancreas carcinoma	0	26	1	72	0.000000	0.737374	0.2610	0.7690
Non small cell lung cancer	21	117	46	320	1.248606	0.823992	18.3263	0.7837
Breast cancer	11	98	53	615	1.302464	0.829720	8.9783	0.8047
Esophagus cancer	4	46	4	86	1.869565	0.892127	2.8866	0.8333
Pancreas neuroendocrine	1	23	0	13	inf	1.000000	0.6654	0.8575
Kidney clear cell carcinoma	2	23	4	100	2.173913	0.913453	1.1465	0.8906
Skin melanoma	6	24	33	240	1.818182	0.928429	3.8702	0.9067
Sarcoma various	1	19	2	106	2.789474	0.936551	0.4747	0.9181
Cholangiocarcinoma	5	15	5	41	2.733333	0.963788	3.0147	0.9190
Urothelial cancer	5	19	15	134	2.350877	0.961649	2.7806	0.9389

Results of mutual exclusivity between LOH of HLA-I and non-HLA alterations for the Hartwig cohort.

It is important to remark that a genetic immune escape study conducted in DLBCL (DOI: 10.1073/pnas.2104504118) found that only biallelic inactivation of HLA-I genes was mutually exclusive with B2M loss. However, other GIE alterations, including monoallelic inactivation of HLA-I and CD58 loss, may co-occur with B2M loss. We believe that our results suggest that in certain tumors, multiple complementary GIE alterations that target different immune escape pathways may be required to satisfactorily avoid immune system control. An example of this rationale has been reported in DLBCL, where biallelic disruption B2M is strongly associated with CD58 loss in order to avoid NK-mediated immune responses (DOI: 10.1016/j.ccr.2011.11.006). We now mention this result in the manuscript:

Results:

Of note, we did not observe a significant mutual exclusivity between LOH of HLA-I and other GIE events in cancer types with sufficient representation of multiple GIE mechanisms (Supp. Table 4). This suggests that certain tumors may require complementary GIE alterations, such as concurrent alterations that disrupt HLA-I mediated neopeptide presentation and CD58 loss (Challa-Malladi et al. 2011), to effectively escape immune surveillance.

Discussion

Importantly, the fact that we did not observe mutual exclusivity between GIE alterations targeting different pathways suggest that in some cases multiple GIE alterations may concur to completely avoid immune surveillance.

I found it highly interesting that the authors conclude that GIE is an early event during tumour evolution. Interestingly, this conclusion is similar to other, orthogonal Nature Genetics studies such as Lakatos et al. 2020 (tumor modelling) and Van den Eynden et al., 2019 (lack of neoantigen depletion). It would be interesting to see a broader perspective on these findings in discussion section.

We have now discussed how our findings fits into the current knowledge about immune escape in tumor evolution. See discussion:

Remarkably, our results also showed that the frequency of genetic immune escape alterations in metastatic patients are comparable to their primary counterparts across most of the cancer types. This result is also supported by independent studies relying on orthogonal approaches^{3,32}, which

denote that early stages of tumorigenesis have already acquired the capacity to escape from immune system recognition. A natural follow-up question is whether one of main differences between macroscopic malignant lesions and microscopic clonal expansions in non-malignant tissue may be related to the immune escape capacity of the former, particularly considering that non-cancer tissues can display comparable rates of certain cancer driver alterations³⁶.

Minor comments:

- The authors state that “GIE may be further fueled by LOH of HLA-I in later tumorigenic stages”. It’s unclear for me where this conclusion comes from.

This sentence comes from the fact that in 7/9 of cancer types with a trend towards GIE enrichment in the metastatic cohort ($\text{Log}_2(\text{odds_ratio}) > 0.5$ in Figure 3b) show a trend towards increased LOH of HLA-I. Additionally, this also refers to the fact that late-stage aggressive tumors tend to harbor higher rates of aneuploidy and LOH rates (DOI: 10.1101/2022.06.17.496528), which in turn correlate with global LOH of HLA-I rates (see Supp. Fig. 3b and response to reviewer #2). In any case we now clarify this further to avoid confusion.

- High proportions of HLA LOH were found in PANET in KICH. The non-focality and lack of other GIE mechanisms suggest that the main driver is not HLA but another gene on chr. 6. Could the authors speculate which driver this could be?

Recurrent patterns of whole chromosomal losses (RPCL) have been broadly reported in both pancreatic neuroendocrine (PANET) (see DOI:10.1038/nature21063 and DOI:10.1158/2159-8290.CD-21-0669) and kidney chromophobe (KICH) (see 10.1016/j.ccr.2014.07.014 and 10.1172/jci.insight.92688.) tumors. However, it is still unclear whether these losses, which seem to target specific chromosomes and are not randomly scattered, are selected because they target a specific gene/locus or due to lack of purifying selection.

Speculating about the potential gene targets in chromosome 6, a conspicuous choice could be DAXX, a known repressor of alternative lengthening of telomeres (ALT) located in chromosome 6 arm p. DAXX show high rates of inactivating mutations in PANET tumors (DOI: 10.1126/science.1200609) and it thus plausible that DAXX mutations combined with loss of the wild-type allele may confer a stronger selective advantage. However this is purely hypothetical and dedicated analyses are required to provide further insights.

We now speculate about this in the discussion:

On the contrary, we did not observe such allelic preference for non-focal LOH of HLA-I, indicating the HLA-I locus does not seem to be the main target of these events. Alternatively, recurrent

patterns of whole chromosome 6 loss^{22,23} might be the result of selection for haploinsufficient tumor-suppressor genes located in this chromosome (such as for instance DAXX, a known regulator of alternative lengthening of telomeres in pancreatic neuroendocrine cancer³⁸).

- I find fig. 2h unnecessary complex with all the triangles etc. Why not just use a continuous scale and directly show the odds ratios?

Following the reviewer's suggestion we have simplified Fig. 2h.

- I understand that dN/dS was calculated using the dNdScv method? To avoid confusion for the reader I would avoid using these terms interchangeably.

Corrected, thanks for the suggestion.

- Fig. 3f-h. The difference between "highly-focal LOH" and "(non-)focal LOH" is clear for me but between "focal LOH" and "non-focal LOH" is rather vague. Could the authors clarify how this distinction is made? Additionally, the y-axis label indicates "samples with ... LOH HLA". I understand the plot refers the genome-wide presence of LOH (and not just HLA)?

The distinction between the three types of LOH events was based on prior thresholds used for CNV positive selection analysis. Specifically, in the seminal study by Nicholas McGranahan and colleagues (PMC5720478) they used a definition of focal LOH of HLA-I events as those LOH events shorter than <75% of the chromosome arm length. However, we also believe that there could be other CNV events more targeted towards a specific loci, in this case the HLA-I locus, which in total spans around to 2 Megabases. Therefore we decided to include a third category of highly-focal LOH representing those events shorter than 3 Megabases. We have now clarified this in the results and methods sections of the manuscript.

We also have corrected the y-axis labels of Fig. 4g-i and Supp. Fig. 3a.

- Fig. 3a. Higher-than-1 dN/dS values are not found for missense mutations. However, some missense mutations could have a profound impact in a protein's functionality, similar like a nonsense mutation. A further distinction between high- and low impact mutations could be useful.

We agree with the reviewer that certain HLA-I missense may have a strong functional impact. However, we did not observe an enrichment of missense variants in the peptide binding domain, which would be the most natural distinction between high and low impact mutations. Alternatively, relying on functional scores based on evolutionary conservation, such as CADD or SIFT, could

render noisy results due to the limited capacity to identify oncogenic mutations (DOI: 10.1038/s41586-021-03771-1) and because of the lack of standardized cut-off thresholds. Considering all these aspects (and the fact that we did not observe higher than-1 dN/dS ratios for missense mutations), we are not sufficiently confident about the reliability of the results precluding us from including this analysis in the study.

Additionally, do dN/dS results remain the same if the “hot spots” mutations shown in panel i are excluded?

In order to evaluate this, without artificially modifying the set of input mutations, we decided to conduct the dN/dS analysis excluding MSI samples (all hotspot mutations come from MSI samples, see above).

We observed very consistent dN/dS patterns, including an enrichment in nonsense and truncating variants and lack of positive selection evidence for missense mutations. This further supports our observations and highlights the role of HLA-I truncating variants beyond MSI tumors.

- Analysis shown in fig. 5a is limited to 16 genes (or 18 genes according to main text?). Positive selection in most of these genes is already known from previous studies. What about other genes involved in the 5 non-HLA pathways? Further, while selection of nonsense mutations make sense for genes with a tumor suppressor role, the opposite is expected for genes such as PDL1, CD28, ... (increased activity necessary for GIE). Could the authors comment on this?

There are 18 genes involved in non-HLA pathways. However, the dN/dS ratios are only calculated for genes whose mechanism of dysregulation includes mutations. This implies excluding SETDB1 and PDL1 (CD274) because their GIE mechanism is high-level amplification. Consequently, the high dN/dS ratios observed for truncating variants only refer to genes with a suspected tumor suppressor role. This is now clarified in the manuscript:

Grouped pan-cancer analysis of the dN/dS ratio in these pathways (covering a total of 16 genes, excluding those whose oncogenic mechanism is based on copy number amplification, see methods Positive selection somatic mutations and indels) revealed a greater than one ratio for nonsense, splice site and truncating variants in both the metastatic and primary datasets (Fig. 6a), which was indicative of positive selection. Moreover, the strong tendency towards loss-of-function mutations is in agreement with the suspected tumor suppressor role of these genes.

- P8. Some references to Fig. 3e and 3f seem to be wrong (think it should be d and e). This is now corrected. Thanks.

- At the end of the first paragraph of p. 8 the authors conclude there is purifying selection against homozygous deletions. Could the authors clarify why this would be?

In our opinion this is a very interesting observation. In spite of the higher-than-expected frequency of HLA-I truncating variants and LOH of HLA-I rates, which clearly indicates that loss-of-function of certain alleles confer a selective advantage to the tumor cell, we only observed 7 patients bearing partial homozygous deletions of HLA-I genes and none of them bearing full homozygous deletions of the entire HLA-I locus.

This suggests that tumors need to express certain levels of HLA-I in the cell surface, likely, to avoid NK mediated immune responses (see DOI:10.1038/onc.2008.267). Specifically, we believe that HLA-C and HLA-E are instrumental to avoid a “kill-me” signal to the infiltrated NK cells (see DOI:10.1111/j.1365-2567.2011.03422.x).

We discuss this observation in the manuscript:

Finally, despite the high frequency of LOH of HLA-I, biallelic deletion of both HLA alleles was an extremely unusual event, featuring the importance of expressing a minimal amount of HLA-I molecules to avoid immune-alterer signals³⁹.

- The text on p8 is jumping from dNdS somatic mutations results, to LOH results and back to somatic mutations. I found this confusing

We have now re-organized this section. We first focus on point mutations and indels and then describe our LOH and homozygous deletions analyses. We agree with the reviewer that this change improves readability.

- Could it be that the association of most genomic features (mutational signatures, ...) shown in fig. 6 is secondary to the association with TMB?

As we mentioned above, we agree with the reviewer that this is an important issue. Therefore we have now included a background control of genes for all features-GIE associations. Moreover, for the associations with mutational signatures we specifically include the clock-like mutation burden as covariate in the linear regression (see methods).

Reviewer #2:

Remarks to the Author:

Summary:

This manuscript by Martinez-Jimenez represent a very interesting broad analysis of genetic alterations affecting immune evasion in primary and metastatic cancers, using published and new cohorts. Overall the findings are new and significant; the methods are overall well explained. The manuscript would benefit from some edits and additional analyses to further strengthen the main conclusions. Furthermore, all the data must be made available as de-identified data to the academic community priori to publication.

We thank the reviewer for these appreciative comments of our work. We have now addressed the comments raised by the reviewers and we believe the manuscript has significantly improved.

Main points:

1)

One of the strengths of the paper is the concomitant analysis of primary AND metastatic tumors also thanks to a new dataset of metastatic tumors from the Hartwig foundation.

One very important point before publication is that de-identified data from the Hartwig foundation used for this publication have to be available immediately to the academic community through platforms such as dbGap to allow reproducibility of the results.

We thank the reviewer for raising this point because it illustrates it was not properly explained in the previous data availability section. All the underlying data included in this study is freely available for the community for academic purposes. However, since both primary and metastatic datasets contain patient's sensitive data, access to genome-wide sequencing data is controlled to ensure the patient's privacy.

Specifically, the Hartwig Medical Foundation metastatic dataset is available upon request for academic use through standardized procedures. Request forms can be found at <https://www.hartwigmedicalfoundation.nl/en/data/data-acces-request/>. To date, more than 200 international different projects and 50 peer-reviewed publications have made use (<https://www.hartwigmedicalfoundation.nl/en/research-and-science/scientific-publications/>) of this cohort.

Concerning the re-processed data from PCAWG with the Hartwig pipeline, the ICGC part can now be accessed through the ICGC platform (<https://dcc.icgc.org/releases/PCAWG/Hartwig>), following their standard access control mechanisms originally put in place. Similarly, the TCGA

portion of the PCAWG dataset can be downloaded at <https://icgc.bionimbus.org/files/5310a3ac-0344-458a-88ce-d55445540120> for users with authorized TCGA access.

Finally, raw sequencing data of the experimental HLA-I typing performed by GenDX has been deposited in EGA under accession number EGAD00001008643.

We have now clarified this in the data availability section.

2)

The authors should state more clearly which are the somatic alterations that are included for the GIE genes to classify a tumor as having a GIE event: Deletions? Loss of function mutations? Any missense mutations? Silent mutations? How was this corrected for the total N of mutations or the total N of copy number alterations?

The precise definition of the somatic alterations considered for each of the 21 genes included in this analysis can be found in the methods section (see GIE alterations, definition).

Briefly, for genes whose mechanism of immune evasion entails the loss of the canonical activity, such as B2M, JAK1/JAK2 or NLRC5, we only considered clonal loss-of-function mutations (e.g., nonsense, frameshift and annotated splice site variants), clonal bi-allelic missense mutations and homozygous deletions. Additionally, for HLA-I genes we also considered LOH of HLA-I as a mechanism of immune evasion.

For CD274 (PD-L1) and SETDB1, whose mechanism of immune evasion involves a gain of function activity, we only considered high-level copy number gains after correcting by total genome ploidy.

To detect positively selected GIE events, we identified GIE alterations with higher-than-expected observed frequency after correcting for the background mutation rate (using dNdScv) and the global copy-number burden (using our own framework, see methods). Moreover, in the revised version of the manuscript we have included simulated control for GIE (simulated GIE, see response to reviewer #1) that is now used in Figure 7 (genomic features and GIE) and Figure 8 (tumor mutation burden and GIE) to correct for the likelihood of our observations given the background alteration rate of the tumors.

3)

In general, the analyses involving the GIE alterations should all involve a comparison to a random set of 'neutral' genes through bootstrapping or other type of randomization.

This is important for example in the context of the analysis shown in Fig. 6a.

We agree with the reviewer that this is an important point that was also raised by reviewer #1. To address these comments, we have now included a background control for GIE, named simulated GIE, for the analyses displayed in Fig. 7 (genomic features and GIE) and Fig. 8. See above response to reviewer#1 for further details.

4)

Regarding the alterations in GIE, in the supplemental table the authors should specify the frequency of mutations or alteration for each of the 21 genes in each tumor type, without pooling the genes by pathways but explicitly reporting the data for each of the GIE gene.

We have now included an additional sheet in Supp. Table 4 that explicitly displays the cancer type-specific alteration frequency for each gene included in the study.

5)

In the GIE evaluation LOH of HLA is the top pathway altered and mainly driving the frequency of GIE alteration. LOH is considered only for HLA, not for other GIE genes — and this makes sense — but the authors should consider whether HLA LOH is a marker of general chromosomal instability and control for the overall rate of LOH across the genome.

As mentioned in the manuscript the high polymorphic nature of the HLA-I locus undoubtedly renders LOH of HLA-I a unique case of LOH that may provide a fitness gain due to the shrinkage of the presentable neopeptidome in HLA-I heterozygous patients. However, we agree with the reviewer that understanding LOH of HLA-I dynamics across cancer types can provide important insights about its role in tumor evolution.

Therefore, for each cancer type we matched the observed LOH of HLA-I frequency with the mean genome-wide LOH rates. This comparison revealed a positive significant association between these two measurements (see below and now included as a panel in Supp. Fig. 3b), suggesting that, to some extent, LOH of HLA-I frequency correlates with global chromosomal instability rates.

This result further justifies our approach of controlling for the background LOH rates in the LOH of HLA-I positive selection analysis (see methods). Importantly, our results revealed that in certain cancer types, the frequency of LOH of HLA-I is higher than the expected given the LOH background rate of these tumors (see Fig. 4), which is likely indicative of positive selection.

We now mention this as part of the results:

“LOH of HLA-I trims the repertoire of HLA-I presented epitopes in HLA-I heterozygous individuals. Consequently, this genomic event may provide a selective advantage to tumor cells often harboring a considerable load of mutations that could ultimately be part of a HLA-I presented neoepitope^{14,12}. Nevertheless, the observed LOH of HLA-I frequency across cancer types showed a global correlation with the average background genome-wide LOH rates (Pearson’s R=0.65 in metastatic and R=0.75 in primary; p-value < 0.01; Supp. Fig. 3b). To further shed light on the tumorigenic role of LOH of HLA-I, we developed a randomization strategy that pinpoints cancer types where the LOH of HLA-I rates were significantly higher than the expected given their background LOH rates using three genomic resolutions...”

And discussion:

Despite the expetable association between genome-wide LOH and LOH of HLA-I rates, our data also showed higher-than-expected LOH of HLA-I frequency across multiple cancer types.

6)

Given the fact that chr6 loss could be driven by other genes different than HLA as the authors point out, for the GIE only the focal events (<3Mb) should be considered. This is also because the authors showed that 'Focal LOH of HLA-I preferentially targets the alleles that presents the highest neopeptide repertoire', suggesting that these focal events are the truly important ones for immune evasion. Therefore, in all analyses involving GIE, the authors should calculate a GIE score and a GIE score based on focal-only-HLA LOH (excluding non focal LOH).

We agree with the reviewer that based on the analysis displayed in Fig. 5 (selection of allele with highest neopeptide ratio in focal LOH of HLA-I) we lack evidence supporting an immune escape role for non-focal LOH of HLA-I. Therefore, we excluded this event for the identification of between genomic features and GIE (Fig. 7) as well as for the analysis investigating the relationship between GIE and tumor mutation burden (Fig. 8).

7)

Can the authors distinguish and study the different effects of LOH due to deletion of one chromosome and copy-number neutral LOH? This could tell whether the main effect of LOH is LOH itself or decrease dosage of HLA genes/molecules.

We appreciate this comment because alongside point 5) (see above) may bring further insights into the role of LOH of HLA-I in tumorigenesis.

Interestingly, the majority of focal LOH of HLA-I cases were copy-number neutral (81% and 70% in metastatic and primary, respectively). This result suggests that in focal LOH of HLA-I gene-dosage decrease does not seem to be the main contributing factor. Importantly, this ratio was considerably lower for non-focal LOH of HLA-I (65% and 35% for metastatic and primary, respectively), supporting the notion that the operative mechanistic forces are different between focal and non-focal HLA-I events.

In our opinion these results also strengthen the observation that focal LOH of HLA-I selects the HLA-I allele with the highest neopeptide repertoire and that this type of GIE events are not primarily aiming at decreasing the number of gene copies in the tumor's DNA.

Results:

"Furthermore, the majority of focal LOH of HLA-I events were copy number neutral (81% in metastatic tumors and 70% in primary), which was considerably higher than for non-focal events (65% in metastatic and 35% in primary), providing further support to the notion that the loss of neopeptide repertoire, and not HLA gene dosage, is the main driving force behind focal LOH of HLA-I."

8)

Last section of the paper:

This section is summarized in the abstract by the sentence:

"Finally, there is a strong tendency for mid and high tumor mutation burden (TMB) tumors to preferentially select LOH of HLA-I for GIE whereas hypermutated samples favor global immune evasion strategies."

The authors should make clear in this sentence of the abstract that they refer to FOCAL LOH of HLA-I (they should add the word 'focal'. Furthermore, it is not clear from this sentence what does GLOBAL immune evasion strategies mean; perhaps 'global' should be substituted by 'other'.

These suggestions have been considered in the revised form of the manuscript. Thanks.

The authors should show the analysis done in Fig. 6a excluding HLA LOH to see if the associations hold true or not and in which tumor types.

Repeating the analysis performed in Fig. 7a -including the control by the simulated GIE- excluding LOH of HLA-I revealed consistent results. Specifically, ~70% (45 out of 65) of the original significant associations between genomic features and GIE were kept, highlighting the robustness of the associations (see Supp. Table 6). It is also important to mention that excluding the most frequent mechanism of immune evasion confines the statistical power of our analyses, which may partially explain the lost significant associations.

Results:

"Importantly, the majority of significant associations (45 out of 65, 69%) were retained after excluding patients harboring LOH of HLA-I, which highlights the consistency of the observed associations."

The analysis shown in fig 7c is very interesting, but would benefit of some clarifications:

-can the authors show the same analysis just including the non clonal TMB?

As expected, when selecting for non-clonal TMB the association between GIE incidence and subclonal TMB (or subclonal neoepitope burden) was less apparent (see below). This is in line with independent observations and emphasizes the importance of mutation cellularity in triggering immune responses.

Furthermore, we also observed an association between increased clonal TMB and GIE prevalence, whereas the relationship with subclonal TMB burden was less evident (Supp. Fig. 5b).

-is this analysis a pan-cancer analysis? what happens if the authors control for tumor type? (I would restrict the analysis to tumor types showing a large spectrum/variance of TMB)

This is indeed an interesting point. We repeated the analysis in three cancer types with high sample size and ample TMB representation (breast, colorectal and non-small cell lung cancers, see below) and observed a consistent pattern of association between GIE incidence and TMB that can not be fully explained by the increased background mutation rate in high-TMB tumors. These results illustrate that the effects are non-tumor type specific.

This trend was still consistent after controlling for the cancer type (Supp. Fig. 5a).

-the increase in GIE alterations correlating with TMB could just be due to the fact that high TMB tumors have a lot of mutations; does this correlation hold true also if the authors disregard point mutations and just consider deletion and amplifications?

It is important to remark that after including a GIE background control (i.e., simulated GIE) our results revealed that the increase in GIE incidence cannot be solely explained by the background mutation rate. Moreover, our positive selection analyses revealed that there is a higher-than-expected frequency of mutation of HLA-I and non-HLA genes across multiple cancer types. Finally, when only regarding deletion and amplification driven GIE events, we observed a modest -but still- positive association with TMB (see below). We believe that the deletion/amplification GIE only result should be taken with caution because of the limited number of patients bearing deletion/amplification driven GIE.

9)

The Discussion would benefit from more insights and thoughts on the underlying mechanisms of the new findings of the paper, such as the one described in Fig, 7.

Following this and reviewer#1 suggestions we now provide a broader perspective of our findings and thoughts in the manuscript discussion.

Reviewer #3:

Remarks to the Author:

The manuscript from Martinez-Jimenez, Cuppen and colleagues presents their analyses of over 6000 cancer genomes evaluated for evidence of genetic mechanisms of presumptive immune escape (GIE). They introduce a new tool, LILAC to improve the complicated business of mutation calling in the HLA-I locus. They find that genetic mechanisms of immune evasion are prevalent pan-cancer and that, importantly, these are present before metastases. In addition, they present analyses which provide insight into the selective drive imparted by mutation burden, mutational processes, and other facets of the genomic landscape.

I found this to be a fascinating and well-executed study. The sample size is substantial and truly drives the strength of the analyses and the, I think, important conclusions - particularly around assumptions around metastatic disease and immune evasion. There has been ample thought given to the vagaries of genome-scale statistics, the algorithmic considerations are clearly laid out, and the manuscript strikes a good balance of detail and 'big-picture' that conveys the overall intent without overstating and claiming easy translational impact. I have few concerns over the approach, the analyses, and the current conclusions as presented.

We thank the reviewer for the very appreciative comments of our work.

I have several suggestions, given the findings presented and considering the power of the dataset assembled that could strengthen and expand the insights.

Given the fact that there are over 6000 genomes in play in nearly 2000 primary and 4000 metastatic/advanced buckets, there is an opportunity to delve more definitely into the timing of genetic/genomic events.

It is important to mention that the analyses conducted here include a large (perhaps the largest) cohort of primary and metastatic unpaired WGS tumors. This means that our approach relies on comprehensively analyzing the global genomic differences in cancer types with sufficient representativeness at the two stages. As illustrated, this is already very useful to identify global patterns of immune escape as well as its relationship with tumor genomic features. However, we believe it is not the optimal scenario to study the precise timing of GIE events relative to other genomic events during tumor evolution. To address that question, studies that include multiple longitudinal biopsies from the same patient such as the ones performed by TRACERx in Lung (10.1038/s41586-019-1032-7) and Kidney (10.1016/j.cell.2018.03.057) or, more recently, by Marco Gerlinger in colorectal MRRd (10.1101/2022.02.16.479224) are more suitable.

Firstly, can the authors estimate the clonal or subclonal nature of the HLA-I events themselves?

By definition, all GIE events considered in our study are clonal (see methods). The reason for this is because it is difficult to foresee the impact of subclonal alterations for the entire tumor entity. However, it is still a reasonable question which is the prevalence of subclonal GIE alterations, the differences across cancer types and whether there are significant differences between primary and metastatic tumors.

Our results revealed that mutation-driven subclonal GIE is a rare event both in primary and metastatic cancer (see below). We did not observe any significant trend towards a specific pathway in subclonal GIE. Overall, there were not remarkable differences between primary and metastatic tumors, although the limited number of subclonal GIE events precluded a statistically robust examination.

Altogether, these results reinforce the notion that GIE is generally an early and clonal event in tumor evolution.

This becomes a rather interesting facet of cancer genome evolution to get to grips with. Further, placing the GIE events in context of the acquisition of the known cancer gene mutations that drive the various cancer types under study is needed. Is there an order that emerges by tissue of origin/tumour type? Is there an order associated with particular driver mutations? The same could be asked of genome doubling, and more grossly, aneuploidy as defined by breakpoint prevalence. Putting the GIE into this estimated temporal framework would further strengthen and indeed augment the important primary versus metastases observation.

As we mentioned above, although very relevant questions, we believe that our dataset does not provide sufficient sensitivity to properly address these evolutionary inquiries. It is extremely challenging to assess the relative timing of individual alterations compared to each other based

on a single tumor biopsy. Thus, we are concerned about the reliability of the aggregated results and prefer to focus on questions that can confidently be addressed with our current data. We mention this limitation in the manuscript's discussion.

Further analyses, ideally relying on longitudinal biopsies, are required to unravel the evolutionary trajectories of these genomic features relative to GIE events.

Further, are cancer driver gene mutations correlated with type/extent of GIE? This is explored to an extent in the mismatch repair gene data but needs to be extended to other cancer genes – those associated with driving proliferation, as an example, or those associated with deregulated chromatin. Again, having a notion of order of events here further enriches these possible insights. This should be extended to copy number/rearrangement driver events as well.

This is a very compelling point that can be addressed with our data. Hence, we explored the association between driver alterations (including those from mutations and copy-number gain/losses) and the presence/absence of GIE (see methods). Interestingly, after correcting by the background control of genes (simulated GIE, see responses to reviewer#1 and #2), several driver alterations were significantly associated with GIE events in colorectal cancer.

Specifically, CASP8, KMT2D, RPL22 and TGFBR2 alterations tended to co-occur with GIE events in colorectal cancer patients (see above). CASP8 and TGFBR2 loss-of-function had been previously associated with immune evasion (see PMID: 26372948 and PMID: 35290801, respectively), which may indicate that these alterations may target independent immune surveillance routes in highly infiltrated tumors (see response to reviewer #1 about mutual exclusivity). However, it is also worth mentioning that co-occurrence of GIE with these driver alterations were highly enriched in MMRd samples (CASP8 50%, KMT2D 37%, RPL22 85% and TGFBR2 50%), which may indicate that these genes are recurrent mutation targets in MMRd tumors. Follow-up studies are thus required to understand the interplay of these driver alterations with GIE events as well as their impact on the tumor microenvironment.

This analysis is now integrated into the genomic features section (Fig. 7 and Supp. Table 6):

Certain driver alterations, beyond the GIE pathways considered in this study, also showed a strong association with GIE events. Specifically, our results showed that CASP8, KMT2D, RPL22 and TGFBR2 alterations tended to co-occur with GIE in colorectal cancer patients. Of note, CASP8¹⁵ and TGFBR2³¹ alterations have previously been linked to immune surveillance escape. Nevertheless, it is also important to consider that in our colorectal cancer cohort co-occurrence of GIE with these driver alterations were highly enriched in MMRd samples (CASP8 50%, KMT2D 37%, RPL22 85% and TGFBR2 50%), which may indicate that these genes are recurrent mutation targets in microsatellite unstable tumors. On the contrary, TP53 alterations were depleted in samples bearing GIE (Fig. 7a). Further studies are thus needed to understand the interplay of these driver alterations with known GIE events as well as their impact on the tumor microenvironment.

Finally, as discussed above, we believe that estimating the precise relative timing of these driver alterations compared to -other- GIE events is hardly possible with our current data.

Following on from this, and again exploiting the fact of having a large number of cancer genomes in play, for the mutational signature associations with GIE, can the authors explore relative timing of the signature processes becoming active (or going dark) relative to the GIE events? Do the GIE events driven at the point mutation level overall bear the signatures of the most prevalent mutational processes?

In line with previous responses, we believe our dataset is not suitable to address these evolutionary questions that would require multiple longitudinal samples at different time points of tumor evolution.

Whilst treatment is given some consideration in the supplemental figures, it is not particularly clearly called out in the main body of the manuscript. I would consider making the apparent lack of statistical correlation with chemotherapy/immunotherapy in the post treatment metastatic cases a point in the main text. It further highlights the importance of the wheels being set in motion in GIE regardless of subsequent treatment.

We thank the reviewer for bringing up this point as we agree this is one of the most important clinical implications of our study. Therefore we now explicitly mention this in the manuscript body where we also discuss its clinical implications.

Results:

Finally, other factors, such as the HLA-I supertype, the germline HLA-I divergence, patient chronological age or exposure to previous treatments, including immunotherapy; failed to attain significant association with GIE (or the association was also observed in the simulated GIE)

Discussion:

We observed a lack of association between prior exposure to cancer therapies, including immunotherapy, and higher GIE frequency. Consequently, the efficacy of immune escape alterations, such as LOH or HLA-I³¹, may be compromised when dealing with the strong immune pressure released by ICI. Dedicated studies are thus required to further understand the role of the different GIE alterations in immunotherapy responses.

I assume not, but were there any paired primary/metastatic pairs in the dataset? If so, was there sufficient opportunity to explore immune editing correlation with extent of GIE?

Unfortunately, our dataset does not include primary and metastatic paired biopsies.

Whilst the correction for molecular age with respect to mutational signatures is appreciated, a different question that arises is if GIE presence in and of itself is correlated with chronological age of the patient. This, of course, ties in with the consideration of GIE timing and clonality overall. One might hypothesize it being more prevalent in generally later onset cancers given the general notion of decreased immune ‘tone’ in advancing age, for instance.

This is also an intriguing point. We assessed, in a cancer type specific manner, whether the patient's age at biopsy correlates with GIE prevalence using a logistic regression (see methods). The majority of cancer types did not show a clear association between age at biopsy and GIE prevalence. The only exception was Diffuse large B-cell lymphoma (DLBCL) that showed a non-significant (after FDR correction) trend towards higher GIE prevalence in older patients, which suggest that late onset DLBCL patients may harbor higher GIE incidence. However, we prefer not to elaborate on this in the main body in the manuscript due to the lack of statistical significance after FDR correction. In any case, this analysis has been integrated into the genomic features and GIE association (see Fig. 7 and Supp. Table 6).

column	ttype	5%	95%	Odds Ratio	pvalue	qvalue
age	DLBCL	0.104090	0.955329	0.529709	0.014716	0.367899
age	SARC	-0.005624	1.741218	0.867797	0.051494	0.413597
age	PRAD	-0.023446	0.518994	0.247774	0.073368	0.413597
age	CESC	-0.060618	1.150111	0.544746	0.077782	0.413597
age	PAAD	-0.038044	0.554267	0.258112	0.087601	0.413597
age	LIHC	-0.769148	0.076010	-0.346569	0.107961	0.413597
age	OV	-0.080763	0.591226	0.255231	0.136528	0.413597
age	SKCM	-0.065507	0.458213	0.196353	0.141654	0.413597
age	GBM	-1.217181	0.200227	-0.508477	0.159657	0.413597
age	STAD	-0.139892	0.817194	0.338651	0.165439	0.413597
age	SG	-0.522049	2.042877	0.760414	0.245184	0.519974
age	COREAD	-0.083355	0.307023	0.111834	0.261453	0.519974
age	UCEC	-0.317153	1.132170	0.407509	0.270386	0.519974
age	NSCLC	-0.303107	0.092064	-0.105521	0.295227	0.527191
age	SCLC	-0.981883	0.337731	-0.322076	0.338703	0.564505
age	CHOL	-0.298730	0.748369	0.224819	0.399992	0.624987
age	ESCA	-0.172127	0.394943	0.111408	0.441228	0.634267
age	GIST	-1.208897	0.543418	-0.332739	0.456672	0.634267
age	CUP	-0.497180	0.905632	0.204226	0.568220	0.735007
age	PANET	-0.601858	1.061646	0.229894	0.588005	0.735007
age	KIRC	-0.343867	0.534731	0.095432	0.670270	0.780518
age	LMS	-0.693696	1.030783	0.168543	0.701633	0.780518
age	BRCA	-0.142456	0.206787	0.032166	0.718077	0.780518
age	HNSC	-0.496661	0.405183	-0.045739	0.842413	0.843909
age	BLCA	-0.307333	0.375978	0.034322	0.843909	0.843909

Results of the logistic regression of age at biopsy with GIE incidence across cancer types.

Is it possible to infer TCR/BCR clonality metrics in the sample series, particularly from those having matched RNAseq? This would potentially augment the nice data re immune infiltrates and GIE presence/absence.

This is a good suggestion. Nevertheless, we currently lack a sufficiently robust tool to efficiently perform TCR/BRC deconvolution in a dataset of this magnitude. Existing open-source tools are highly demanding in terms of time and resources restricting their application to a dataset of >12,000 WGS samples (including paired germline and tumor). For these reasons, we believe that exploration of the interplay of the TCR repertoire is an interesting follow-up question that is beyond the scope of this already extensive analysis.

Did the authors consider an assessment of the tumor microbiome and correlation with GIE in this dataset? Given the growing body of evidence (much still somewhat associative in nature, to be fair) of the tumor microbiome in the biology of the cancer, having genomes presents a rather unique opportunity to extend some of these studies in this immune escape context. Further, the

association with particular tumor types and chemotherapy efficacy/exposures might be another avenue to walk down if the data allow.

We agree with the reviewer that the study of the mechanistic interplay between the tumor microbiome and the immune system is one of the attractive open questions in the tumor immunology field. However, we also feel that the study of the tumor microbiome is in early days compared to other immunogenetic analysis and its inclusion would inevitably entail extensive analyses and subsequent controls. For that reason we believe this question is beyond the framework of the current study.

Finally, given the uniqueness of the dataset, have the authors considered assessment of matching germline (those from PBMC) for any evidence of CHIP GIE events? Is this a feature of aging in accompanying CHIP point mutations?

This is another of the most relevant questions in the tumor immunology field. However, as for the microbiome, we are beginning to grasp the role of CHIP in the initiation and development of solid tumors (DOI: 10.1158/0008-5472.CAN-22-0985). In fact, systematic identification of the (epi)genomic events underlying CHIP is still a challenging question (DOI: 10.1038/s41467-022-31878-0), which automatically hampers the integration with any downstream analysis. We believe that this question needs dedicated tools and analyses that are currently far from the scope of this manuscript.

We mention these two interesting follow-up questions in the manuscript discussion.

Finally, tumor extrinsic factors such as clonal hematopoiesis, tumor associated microbiome or the tissue architecture may also play an important role in tumor immune evasion. We thus hope that the combination of cancer genomics with high-resolution characterization of the tumor microenvironment will aid in further understanding of the interplay between tumor evolution and the immune system.

There is a tendency in the manuscript to use language of active involvement of the tumor in shaping its genome. “Tumors tailor”, “tumors select”, ‘tumors leveraged’ – this, of course, is all selection and fitness. It would better to avoid this sort of language in my view and rephrase as things being selected for, increased fitness – the parlance of tumor evolution. The former fuels the tumor as sentient agent of patient demise that is quite problematic for many patients and their advocates.

We thank the reviewer for raising this point. We have now reworded the manuscript to frame the language in the context of passive tumor evolution.

Decision Letter, first revision:

5th Jan 2023

Dear Dr. Cuppen,

Thank you for submitting your revised manuscript "Genetic immune escape landscape in primary and metastatic cancer" (NG-A60471R). It has now been seen by the original referees and their comments are below. The reviewers find that the paper has improved in revision, and therefore we'll be happy in principle to publish it in Nature Genetics, pending minor revisions to satisfy the referees' final requests and to comply with our editorial and formatting guidelines.

Sincerely,

Safia Danovi
Editor
Nature Genetics

Reviewer #1 (Remarks to the Author):

The authors provided an extensive rebuttal and added several additional analyses, addressing most concerns that were risen by the reviewers.

I mainly found the additional analysis on the TRACERx study useful. We also had issues with running LOHHLA in the past and look forward to using LILAC for future analyses.

The only point I'm still not entirely convinced about is the background control analysis that was run on the TMB-GIE prevalence. As expected, there is a background correlation, and the authors conclude that it does not fully explain the observed trend. However, one must realize that these background results are strongly dependent on the precise methodology and assumptions that are used. In this regard, I found it strange that the background plot seems to flatten and even go down for higher TMBs. I was missing the limitations of this background control analysis as a critical note in the discussion section.

Note. I agree with reviewer #2 that data availability is still a major problem in our field, and while most studies now make their data formally available upon publication, in practice it is sometimes hard (not to say impossible) to get access to data. In this regard, our experience with getting access to HMF data is also not that great.

Finally, I wish to congratulate the authors with this important work.

Reviewer #2 (Remarks to the Author):

My comments have been successfully addressed.

Reviewer #3 (Remarks to the Author):

The author's response to the various points of critique falls somewhat short of what I still think could be done in this unique dataset. Whilst the point is taken that these are non-paired tumor/met samples, analyses of the primary to begin to get insight into relative timing (subclonality) of events in whole genome data is possible. The response of "it is extremely challenging ... based on a single tumor biopsy", whilst being of course true in any and every analyses of human cancer, negates the fact that the majority of any type of longitudinal series at this scale will only have single biopsies and that, in clinical practice, this is nearly always the case. Thus, an exploration to the extent possible seemed useful here.

Likewise, even sampling within the dataset (either randomly or by level of GEI evidence) would be an approach to TCR/BCR association - rather than saying it is too computationally intense. The uniqueness of the main insights (that I fully recognize and applaud) would be further enriched and help direct others to the most potentially pertinent next steps in much smaller, longitudinally, clinically annotated series.

The responses to the various critiques take a somewhat counsel of perfection tack that is understandable but I think slightly overdone given the opportunity to further delineate some of the key additional features.

Overall, the additional validation and the limited exploration of additional correlatives strengthens what is very nice piece of work that has import for understanding the how's and why's de-novo resistance to immunotherapies my arise and how to think about strategies to overcome them for more patients.

Author Rebuttal, first revision:

Reviewer #1:

Remarks to the Author:

The authors provided an extensive rebuttal and added several additional analyses, addressing most concerns that were risen by the reviewers.

I mainly found the additional analysis on the TRACERx study useful. We also had issues with running LOHHLA in the past and look forward to using LILAC for future analyses.

The only point I'm still not entirely convinced about is the background control analysis that was run on the TMB-GIE prevalence. As expected, there is a background correlation, and the authors conclude that it does not fully explain the observed trend. However, one must realize that these background results are strongly dependent on the precise methodology and assumptions that are used. In this regard, I found it strange that the background plot seems to flatten and even go down for higher TMBs. I was missing the limitations of this background control analysis as a critical note in the discussion section.

We agree that the background "simulated GIE" is sensitive to the precise methodology used for randomization. However, the conclusions drawn from this analysis (i.e., higher than expected GIE frequency, particularly in high-TMB tumors) are also supported by the positive selection analysis (e.g., enrichment of HLA-I mutations in MSI tumors) as well as by independent studies using orthogonal simulation strategies (e.g., Lakatos E, et al. Nat. Genetics 2020).

The fact that the background TMB seems to decline at hypermutated tumors is likely caused by the lower background LOH rates of these tumors, which is the main underlying source of simulated GIE.

Any case we acknowledge this limitation in the manuscript discussion:

"It is important to mention that the GIE escalation as the TMB increases was not entirely ascribed to the underlying increase in background mutation rate, particularly in hypermutated tumors. Although the modeling of background GIE rates could be sensitive to the selected randomization strategy, our results are supported by independent studies based on orthogonal analytical approaches²⁹, evidencing the robustness of our conclusions."

Note. I agree with reviewer #2 that data availability is still a major problem in our field, and while most studies now make their data formally available upon publication, in practice it is sometimes hard (not to say impossible) to get access to data. In this regard, our experience with getting access to HMF data is also not that great.

At Hartwig we are fully committed to ensure data accessibility for academic purposes while preserving patient's rights when it comes to their privacy. To date more than 200 independent researchers have accessed the Hartwig cohort.

Finally, I wish to congratulate the authors with this important work.

Thanks.

Reviewer #2:

Remarks to the Author:

My comments have been successfully addressed.

Reviewer #3:

Remarks to the Author:

The author's response to the various points of critique falls somewhat short of what I still think could be done in this unique dataset. Whilst the point is taken that these are non-paired tumor/met samples, analyses of the primary to begin to get insight into relative timing (subclonality) of events in whole genome data is possible. The response of "it is extremely

challenging ... based on a single tumor biopsy", whilst being of course true in any and every analyses of human cancer, negates the fact that the majority of any type of longitudinal series at this scale will only have single biopsies and that, in clinical practice, this is nearly always the case. Thus, an exploration to the extent possible seemed useful here.

Likewise, even sampling within the dataset (either randomly or by level of GEI evidence) would be an approach to TCR/BCR association - rather than saying it is too computationally intense. The uniqueness

of the main insights (that I fully recognize and applaud) would be further enriched and help direct others to the most potentially pertinent next steps in much smaller, longitudinally, clinically annotated series.

The responses to the various critiques take a somewhat counsel of perfection tack that is understandable but I think slightly overdone given the opportunity to further delineate some of the key additional features.

Overall, the additional validation and the limited exploration of additional correlatives strengthens what is very nice piece of work that has import for understanding the how's and why's de-novo resistance to immunotherapies my arise and how to think about strategies to overcome them for more patients.

We thank the reviewer for the positive comments and for the relevant analyses suggested. We are confident that follow-up studies -leveraging this dataset- will address some of the suggestions raised by the reviewer.

Final Decision Letter:

10th Mar 2023

Dear Dr. Cuppen,

I am delighted to say that your manuscript "Genetic immune escape landscape in primary and metastatic cancer" has been accepted for publication in an upcoming issue of Nature Genetics.

Your paper will be published online after we receive your corrections and will appear in print in the next available issue. You can find out your date of online publication by contacting the Nature Press Office (press@nature.com) after sending your e-proof corrections. Now is the time to inform your Public Relations or Press Office about your paper, as they might be interested in promoting its publication. This will allow them time to prepare an accurate and satisfactory press release. Include your manuscript tracking number (NG-A60471R1) and the name of the journal, which they will need when they contact our Press Office.

Please note that *Nature Genetics* is a Transformative Journal (TJ). Authors may publish their research with us through the traditional subscription access route or make their paper immediately open access through payment of an article-processing charge (APC). Authors will not be required to make a final decision about access to their article until it has been accepted. [Find out more about Transformative Journals](https://www.springernature.com/gp/open-research/transformative-journals)

Authors may need to take specific actions to achieve [compliance with funder and institutional open access mandates](https://www.springernature.com/gp/open-research/funding/policy-compliance-faqs). If your research is supported by a funder that requires immediate open access (e.g. according to [Plan S principles](https://www.springernature.com/gp/open-research/plan-s-compliance)) then you should select the gold OA route, and we will direct you to the compliant route where possible. For authors selecting the subscription publication route, the journal's standard licensing terms will need to be accepted, including [self-archiving-and-license-to-publish](https://www.nature.com/nature-portfolio/editorial-policies/self-archiving-and-license-to-publish). Those licensing terms will supersede any other terms that the author or any third party may assert apply to any version of the manuscript.

Please note that Nature Portfolio offers an immediate open access option only for papers that were first submitted after 1 January, 2021.

If you have not already done so, we invite you to upload the step-by-step protocols used in this manuscript to the Protocols Exchange, part of our on-line web resource, natureprotocols.com. If you complete the upload by the time you receive your manuscript proofs, we can insert links in your article that lead directly to the protocol details. Your protocol will be made freely available upon publication of your paper. By participating in natureprotocols.com, you are enabling researchers to more readily reproduce or adapt the methodology you use. [Natureprotocols.com](http://natureprotocols.com) is fully searchable, providing your protocols and paper with increased utility and visibility. Please submit your protocol to <https://protocolexchange.researchsquare.com/>. After entering your nature.com username and password you will need to enter your manuscript number (NG-A60471R1). Further information can be found at <https://www.nature.com/nature-portfolio/editorial-policies/reporting-standards#protocols>

Sincerely,

Safia Danovi
Editor
Nature Genetics